# Persistent warm and cold spells in the Northern Hemisphere extratropics: regionalisation, synoptic-scale dynamics, and temperature budget.

Alexandre Tuel[1] and Olivia Martius[1,2]

[1]Institute of Geography and Oeschger Centre for Climate Change Research, University of Bern, Switzerland
[2]Mobiliar Lab for Natural Risks, University of Bern, Switzerland

**Correspondence:** Alexandre Tuel (alexandre.tuel@unibe.ch)

**Abstract.** Persistent warm and cold spells are often high-impact events that may lead to significant increases in mortality and crop damage, and can put substantial pressure on the power grid. Taking their spatial dependence into account is critical to understand the associated risks, whether in present-day or future climates. Here, we present a novel regionalisation approach of 3-week warm and cold spells in winter and summer across the Northern Hemisphere extratropics based on the association of the warm and cold spells with the large-scale circulation. We identify spatially coherent but not necessarily connected regions where spells tend to co-occur over 3-week timescales and are associated with similar large-scale circulation patterns. We discuss the physical drivers responsible for persistent extreme temperature anomalies. Cold spells systematically result from northerly cold advection, whereas warm spells are caused either by adiabatic warming (in summer) or warm advection (in winter). We also discuss some key mechanisms contributing to the persistence in temperature extremes. Blocks are important upper-level features associated with such events – co-localized blocks for persistent summer warm spells in the northern latitudes, downstream blocks for winter cold spells in the eastern edges of continental landmasses and upstream blocks for winter cold spells in Europe, north-western North America and east Asia. Recurrent Rossby wave patterns are also relevant for cold and warm spell persistence in many mid-latitude regions, in particular in central and southern Europe. Additionally, summer warm spells are often accompanied by negative precipitation anomalies that likely play an important role through land-atmosphere feedbacks.

## 1   Introduction

Warm and cold spells often have damaging consequences for agriculture, power demand, human health or infrastructure (e.g., Jendritzky, 1999; Añel et al., 2017; Buras et al., 2020). Just the last few years have witnessed several high-impact events associated with unusually persistent surface temperature anomalies. In February 2021, large parts of North America experienced a prolonged, record-breaking cold wave that led to unprecedented power failures, more than 240 casualties, and upwards of $200 billion in damages (ASCE, 2022; NPR, 2022). Throughout June and July of that same year, persistent high temperatures were reported in parts of Russia and Eastern Europe (Tuel et al., 2022) and in Northwestern America (McKinnon and Simpson, 2022), where they caused massive wildfires and thousands of excess deaths (White et al., 2022). In 2022, temperatures across Western Europe remained largely above normal for much of the year (InfoClimat, 2022), while China experienced its longest

heatwave on record, causing crop failures, power shortages, and high water stress (China Meteorological Administration, 2022). Finally, in 2023, the southwestern US and Canada experienced persistent above-average temperatures (temperatures in Phoenix for instance exceeded 43°C for 31 days straight; New York Times, 2023), causing major wildfires (Reuters, 2023).

The impacts of warm and cold spells are modulated not only by their magnitude, but also by their temporal persistence and their spatial extent. For instance, long summer or winter cold spells can be especially detrimental to vegetation (Chapman et al., 2020). von Buttlar et al. (2018) found that the duration of heat extremes was key in modulating the impacts on vegetation, with multi-week spells being more harmful than short-term events. Long summer warm spells can also lead to droughts or make droughts worse, notably in water-scarce regions (García-Herrera et al., 2010; Vogel et al., 2021). Finally, Polt et al. (2023) argued in the case of Germany that heatwaves were the most impact-relevant at time scales between 2 weeks and 2 months. While our knowledge of the relationship between the persistence of warm and cold spells and their impacts remains incomplete and further studies are needed to improve it, there is therefore some quantitative evidence that persistent warm spells can lead to much stronger impacts on human and natural systems.

To accurately estimate impacts associated with persistent warm and cold spells in the extratropics, and to improve our ability to forecast them, it is crucial to understand their spatio-temporal characteristics and their physical drivers (van Straaten et al., 2022). Of particular interest are the processes responsible for the persistence of temperature anomalies over sub-seasonal to seasonal timescales, especially as they may be impacted by climate change (Hoskins and Woollings, 2015; Pfleiderer et al., 2019; Hoffmann et al., 2021).

There already is substantial literature on extratropical warm and cold extremes and their driving factors (e.g., Domeisen et al., 2023). Among these, atmospheric blocking has long been recognized as a key driver of temperature extremes in the mid- to high-latitudes, for both summer warm spells and winter cold spells (Buehler et al., 2011; Pfahl and Wernli, 2012; Schaller et al., 2018; Kautz et al., 2022). Topography (Jiménez-Esteve and Domeisen, 2022) and land-atmosphere interactions (Bieli et al., 2015; Miralles et al., 2019; Wehrli et al., 2019) also play important roles, as does polar vortex variability for cold air outbreaks in mid- and high-latitudes (Kolstad et al., 2010; Biernat et al., 2021; Huang et al., 2021). However, previous studies have generally used short time windows to define warm or cold spells, on the order of 3-5 days (e.g., Schaller et al., 2018; Plavcová and Kyselý, 2019; Jeong et al., 2021; Jiménez-Esteve and Domeisen, 2022). Heatwaves, for instance, are frequently defined as sequences of at least 3 or 5 extreme warm days, with the longest events usually lasting about a week (e.g., Perkins and Alexander, 2013; Plavcová and Kyselý, 2019). While such windows may also identify multi-week spells, they focus on the part of the spells with the most extreme temperatures. In addition, a sizeable fraction of persistent spells do not include short periods of extreme temperature anomalies, especially in summer (Figure A1). Finally, choosing a longer window is useful to identify regions where spells tend to co-occur, as it allows to capture the trajectory of the synoptic-scale systems responsible for the temperature anomalies across their lifetime.

A handful of studies have looked specifically at prolonged periods of heat and cold, especially over sub-seasonal timescales. Galfi and Lucarini (2021) modeled the marginal probabilities of persistent warm and cold spells of various lengths to describe their climatology across several regions with large deviation theory. Carril et al. (2008) analysed the circulation patterns associated with extreme warm summer months in Europe and the Mediterranean, and Li et al. (2017) did the same for persistent cold

spells of at least 10 days in North America. More recently, van Straaten et al. (2022) identified potential sources of predictability for monthly warm anomalies in Central Europe. Additionally, (Röthlisberger and Martius, 2019) showed that blocking impacted summertime warm spell persistence in the mid-latitudes. Double jet structures Rousi et al. (2022) and recurrent Rossby wave packets (Röthlisberger et al., 2019; Ali et al., 2022; Jiménez-Esteve and Domeisen, 2022) are also thought to be important drivers of persistent summer heatwaves.

Still, much remains unknown about the geography of persistent warm and cold spells, their spatial dependence, and how the role of various identified drivers varies in space. Such information is essential for risk assessment and to improve forecasts (Perkins, 2015; van Straaten et al., 2022). Persistent spells indeed typically occur over large spatial scales, from hundreds to thousands of square kilometers, with potentially complex spatial footprints (Lyon et al., 2019; Vogel et al., 2020). Yet, most previous work on heat waves, for instance, has focused at the grid-point scale or over regions based on impacts, administrative boundaries, data availability or past observed events (e.g., Hirschi et al., 2011; Bieli et al., 2015; Lowe et al., 2015; Plavcová and Kyselý, 2019; Zschenderlein et al., 2019; Hartig et al., 2022). Case studies provide useful insights into the dynamics of persistent warm and cold spells, but their results cannot necessarily be generalised. There is consequently value in a more systematic regionalisation of persistent warm and cold spells, as has been attempted in a few studies at the regional scale (e.g. Carril et al., 2008; Stefanon et al., 2012; Sousa et al., 2018).

Here, we introduce a simple, meteorologically-driven regionalisation method for sub-seasonal warm and cold spells. Regionalisation is a common approach to explore spatial dependence in weather and climate (e.g., Bernard et al., 2013; Yu et al., 2018; Tuel and Martius, 2022). Among its advantages are more tractable analyses and potentially more robust and physically meaningful results (Saunders et al., 2021). Carril et al. (2008), Xie et al. (2017) and Yu et al. (2018) all attempted a regionalisation of warm and cold spells based on EOF analysis, but at regional scales only (respectively for Europe, North America and China). We take here instead a hemispheric perspective. Our proposed method uses quantile regression to group together areas where warm and cold spells share the same dependence to the large-scale circulation. We apply the method to the Northern Hemisphere for winter (December-February, DJF) and summer (June-August, JJA) separately, and analyse circulation patterns and temperature budget anomalies during persistent warm and cold spells. Our primary goal here is not to discuss individual regions in detail, but instead to compare and contrast the main processes associated with persistent temperature extremes across space and time.

## 2 Data and methods

### 2.1 Data

Data for this study come from the ERA5 reanalysis (Hersbach et al., 2020). All data, unless specified, extend over the Northern Hemisphere (0-85°N) on a $1 \times 1°$ grid and cover the 1979-2020 period at a daily resolution. In addition to 2-meter temperature (°C), we consider precipitation (mm), 500 hPa geopotential height (Z500) (m) and 200 hPa wind speed (m/s). We linearly detrend the daily temperature and Z500 data at each grid point and for each season (DJF/JJA) separately to remove long-term warming-induced trends. In addition, we use the binary cyclone detection indices computed by Rohrer et al. (2020) (which we

extended up to 2020) in which cyclones are identified as closed sea level pressure contours lasting at least 4 days (Wernli and Schwierz, 2006). We also analyse atmospheric blocks with the blocking detection method implemented by Steinfeld (2021) and adapted from the original index proposed by Schwierz et al. (2004). Blocks are identified as regions of persistent negative anomalies (70% contour overlap between consecutive 6-hourly time steps for at least 5 days) of 500–150 hPa vertically-integrated potential vorticity (PV) that are below the 10th percentile of the daily climatological PV anomaly distribution (1979–2020). PV fields are first obtained from ERA5 6-hourly model-level wind, temperature, and pressure. Finally, we calculate daily-mean temperature budget terms (see section 2.2.4) from 6-hourly temperature, zonal and meridional winds, and pressure velocity at the 850 hPa level and on the bottommost 15 ERA5 sigma levels (approximately corresponding to a 50-60 hPa-thick layer above the surface).

For the statistical modeling and regionalisation of warm and cold spells, we further normalise daily temperature data on each day and at each grid point by subtracting the mean and dividing by the standard deviation, both being estimated on a 30-day, 7-year moving window (as in Pfleiderer et al. (2019)). This allows us to remove the seasonality and long-term trends so as to focus exclusively on sub-seasonal variability. We only use this normalisation in the quantile regression (section 2.2.1). For anomaly maps during warm and cold spells (section 2.2.3), we use the non-normalised temperature data.

### 2.1.1 R-metric

To assess the potential role of recurrent synoptic-scale Rossby wave patterns, we use the R-metric developed by Röthlisberger et al. (2019). The R-metric is calculated from conventional Hovmoller diagrams of 6-hourly 35–65°N averaged 250 hPa meridional wind. We first apply apply a 14-day running mean to the series, and remove contributions outside the synoptic wavenumber range $k = 4 - 15$. The R-metric is then defined as the absolute value of the time- and wavenumber-filtered signal. Large R-metric values therefore tend to indicate the presence of several successive synoptic-scale wave packets, in other words recurrent Rossby waves.

### 2.2 Methods

Our proposed regionalisation method brings together grid points where persistent warm or cold spells share a similar dependence on the large-scale circulation. There are many ways to characterise a time series, such as temperature, as "persistent" (Tuel and Martius, 2023). What we consider as "persistent" spells here are periods of time that stretch over sub-seasonal timescales (week to month) and during which temperature anomalies tend to remain either much above or much below zero. This approach corresponds to the "episodic" persistence in the typology introduced by Tuel and Martius (2023). To identify such periods, we consider a fixed sub-seasonal timescale of 3 weeks, and define sub-seasonal warm and cold spells as 3-week periods during which the average temperature is above its 3-week 95[th] percentile or below its 5[th] percentile. The regionalisation relies on a quantile regression approach which models extreme temperature percentiles as a function of covariates, here the principal component time series of the Northern Hemisphere Z500 field. The approach is conceptually similar to that of Tuel and Martius (2022) for temporal clustering of precipitation extremes.

### 2.2.1 Statistical modeling of persistent temperature anomalies

125 We begin by calculating at each grid point time series of normalised temperature anomalies averaged over non-overlapping 3-week intervals. We select a 3-week timescale to specifically focus on warm and cold spells with sub-seasonal persistence. Our results are not especially sensitive to the exact value of this timescale within about 2-4 weeks. Taking non-overlapping intervals is however important to ensure successive values are reasonably independent. Similarly, we average the detrended Northern Hemisphere Z500 fields over the same 3-week intervals, and calculate, separately for DJF and JJA, time series of their first 25

130 principal components (accounting for at least 85% of variability). We do not consider the physical relevance of these principal component, but simply use them as a way to reduce the dimensionality of the Z500 fields.

We then implement at each grid point a quantile regression with the principal component series $\{X_i(t)\}_{i=1..25}$ as covariates:

$$Q_\tau(t) = \beta_0(\tau) + \sum_{i=1}^{25} \beta_i(\tau) X_i(t) + \epsilon(\tau) \tag{1}$$

where $Q_\tau(t)$ is the $\tau$-th percentile of the 3-week averaged temperature series and $\epsilon(\tau)$ is a Gaussian-distributed error term.

For warm spells, we select $\tau = 0.95$, and for cold spells $\tau = 0.05$. We perform the regression for DJF and JJA separately with the R package `quantreg` (Koenker, 2022). Model goodness-of-fit is estimated with the deviation ratio $DR = 1 - \frac{S}{S_0}$, where $S$ is the sum of absolute deviations in the fitted model, and $S_0$ is the sum of absolute deviations in the null model (where $Q_\tau(t) = \beta_0$). Like the standard coefficient of determination ($r^2$), $DR$ is between 0 and 1, with higher values indicating a better fit. To only retain locations with meaningful $\beta_i$ coefficients – *i.e.*, locations where large-scale dynamics exert a strong control

on extreme temperature percentiles that is well-captured by our selected covariates – we remove grid points with $DR < 0.4$ from the analysis. This threshold is subjective, but it is high enough to ensure a reasonable goodness-of-fit while retaining large parts of the extratropics (section 3).

We then implement a Partitioning Around Medoids (PAM) clustering algorithm on the $\{\beta_i\}_{i>0} \in \mathbb{R}^{25}$ vectors using the $L^2$ norm as distance metric. We therefore do not explicitly use location data for the clustering. PAM divides a set of $N$ points

between $K$ clusters (which we refer to here as "regions"), where $K$ must be specified beforehand. It begins by randomly selecting $K$ region centers (medoids) and assigning all other points to the closest medoid. It then updates each region medoid which minimises total intra-region distance and redistributes the other $N - K$ points, until region medoids stop changing. We choose $K$ based on the silhouette coefficient (Rousseeuw, 1987). The silhouette coefficient is a common metric used to identify an "optimal" cluster number. It compares the mean intra- and inter-cluster distances, with higher values indicating a better

clustering (high inter-cluster distance and low intra-cluster distance). We calculate the average silhouette coefficient over all points for $3 \le K \le 50$ from 100 realizations of the PAM algorithm for each $K$. The optimal number of regions, $K^*$, is that which maximises the silhouette coefficient. Only in one case do we choose $K^*$ differently (see section 3 for details).

Each region is screened for points far away from all others in the region. This step helps to better visualise regions because the $DR$ maps can be noisy, especially after removing points with $DR > 0.4$. We use the `dbscan` algorithm (Ester et al., 1996)

so that each region point has at least 10 neighboring points in the same region within 500 km. These values were chosen to

eliminate outlying points without affecting the regions' spatial coherence or the analysis results (at most 5% of the points end up being removed).

### 2.2.2 Cold and warm spell identification

Once regions are formed, we calculate region-averaged daily temperature anomaly series, which we then average with a three-week moving window. For each season, we identify persistent warm spells as the 3-week periods during which the three-week averaged temperature is above its 95[th] percentile. Similarly, cold spells are the periods when the series is below its 5[th] percentile. We make sure that spells do not overlap by first selecting the warmest/coldest three-week period, removing the corresponding data and recalculating the 3-week moving averages, until no more values exceed or go below the initial extreme percentiles. This yields an average of 16-19 events (*i.e.*, three-week periods) for each region and season, *i.e.* one every 2-3 years. We therefore select reasonably uncommon events, but numerous enough to obtain robust results from the limited, 42-year ERA5 record.

For each event, we define an analysis window (during which to calculate circulation and temperature budget anomalies) as the period between the last day before the event on which the region-mean daily temperature anomaly was negative (for warm spells) or positive (for cold spells) and the day during the event on which the daily-mean temperature anomaly peaks (maximum for warm spells, minimum for cold spells; Figure 1). This definition follows Röthlisberger and Papritz (2023a) and Röthlisberger and Papritz (2023b). The reason why we do this, instead of using the original three-week periods that define the spells, is to capture the onset of the persistent warm or cold spells at a daily resolution, and avoid the decay part of the spells when synoptic drivers are likely disappearing. Note that with this definition the analysis window length is different for each spell.

By looking back at the last day when temperature anomalies changed signs, we can capture the onset and build-up period of the spells, not just their peak. Admittedly, this choice may not always be the most relevant for individual events. However, on average, it allows to capture the period during which temperatures consistently depart from the climatology. One downside is that this definition may favour unrealistically long build-up times and weaken the significance of the detected synoptic anomalies. Another limitation of this choice is that it excludes the days following the peak temperature anomaly, which can still be relevant, for instance in terms of impacts. Still, we show in Figure A2 that spells do tend to develop over multi-week timescales, slowly and persistently deviating from their climatology, which supports our choice of analysis window.

### 2.2.3 Synoptic anomalies during warm and cold spells

We then calculate anomaly maps of selected fields (section 2.1) averaged over the analysis windows. Anomalies are calculated for each event within each season and each region separately. We average the different fields during the corresponding event time window, and subtract the long-term average calculated over the same calendar days across the whole ERA5 record. Anomaly maps for each season and region are then averaged across events for warm and cold spells separately. We assess the statistical significance of the anomalies by calculating at each grid point their rank among a sample of 1000 anomalies obtained from randomly generated sets of events with the same distribution of duration and timing of occurrence (calendar days) as observed events. From this rank an empirical p-value is obtained, which we correct for the false discovery rate (Wilks, 2016). As mentioned in the introduction, our primary goal is not to analyse each region separately, but to highlight common features

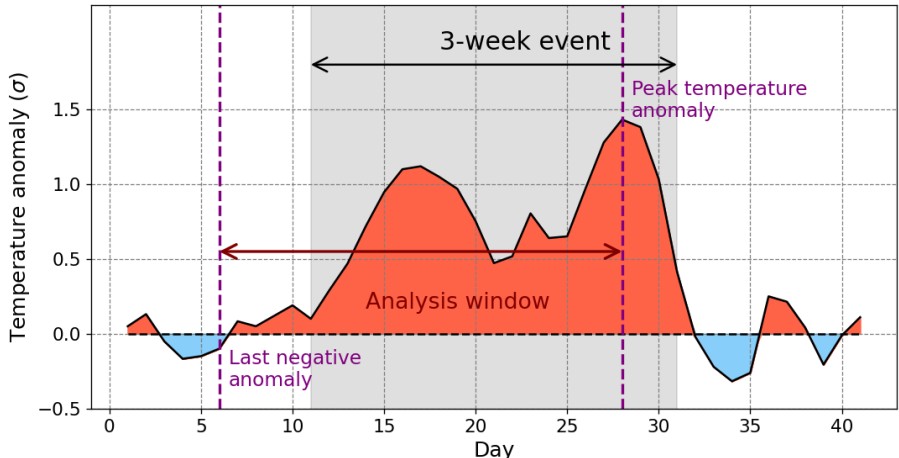

**Figure 1. Persistent warm and cold spell identification.** From the region-averaged normalised (*i.e.*, unitless; see section 2.1) daily temperature series, a warm spell (gray shading) is identified as an extreme 3-week average anomaly (here, from days 11-31). We then identify the corresponding analysis window as the period between the last day before the event on which the temperature anomaly was negative (here day 6) and the day during the event on which the temperature anomaly peaked (here day 28). Cold spells are identified in a symmetrical way. Note that the reference day value for this example was artificially set at 0.

between regions. To do so, we centre each anomaly map around the (geographical) median point of the corresponding region, so as to be able to visually identify sets of regions with similar characteristics.

### 2.2.4 Temperature budget analysis

To understand the processes contributing to warm and cold spell onset and persistence, we rely on the Eulerian temperature tendency equation near the surface (on the bottommost model levels; section 2.1). Let $T$ be the air temperature and $p$ the air pressure at a given grid point. The potential temperature is defined as $\theta = T \left( \frac{p_0}{p} \right)^{\kappa}$ where $\kappa = R/C_p$, $R$ is the gas constant for air and $c_p$ the specific heat capacity at constant pressure. The temperature tendency $\frac{\partial T}{\partial t}$ on sigma levels is obtained by taking the Lagrangian derivative of $\theta$ (Schielicke and Pfahl, 2022):

$$\frac{\partial T}{\partial t}\bigg|_{\sigma} = -\underbrace{\left( u \frac{\partial T}{\partial x}\bigg|_{\sigma} + v \frac{\partial T}{\partial y}\bigg|_{\sigma} \right)}_{\substack{\text{horizontal advection} \\ \text{on sigma level}}} + \underbrace{\frac{\kappa T \omega}{p}}_{\text{adiabatic term}} + \underbrace{\frac{D\theta}{Dt} \left( \frac{p_0}{p} \right)^{-\kappa}}_{\text{diabatic term}} \tag{2}$$

where the subscript $|_{\sigma}$ makes explicit that partial derivatives are taken along the sigma level. $\omega = \frac{Dp}{Dt}$ is the vertical velocity (in Pa/s) and $(u, v)$ the component of the horizontal (*i.e.*, constant-sigma) wind. Equation (2) expands the temperature tendency into three common terms: the horizontal advection of temperature (at constant sigma), the adiabatic warming/cooling term linked to vertical motion, and the diabatic term, which includes contributions from sensible and latent heat fluxes. The temperature

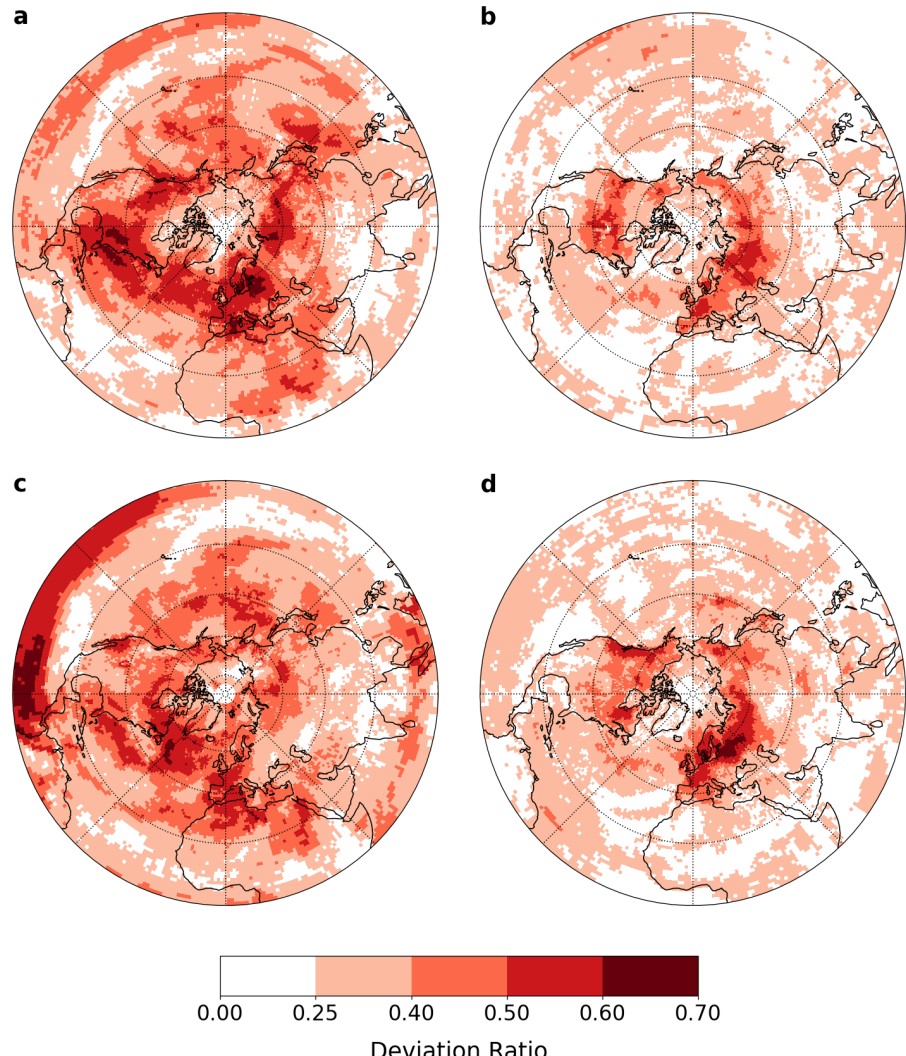

**Figure 2.** Goodnes-of-fit for the quantile regression model: (a) DJF cold spells, (b) JJA cold spells, (c) DJF warm spells and (d) JJA warm spells.

tendency, the horizontal advection and the adiabatic term can all be directly computed from the data using their analytical expressions. The diabatic term is then calculated as the residual in (2).

As an additional analysis, we also consider the temperature tendency equation in the low troposphere (at 850 hPa), for which
the expression is the same as in equation 2, with $T, u, v, \omega, p, \frac{D\theta}{Dt}$ however defined on the 850 hPa level.

## 3 Regionalisation results

We begin with the results of the quantile regression model and regionalisation approach. Deviation ratio $DR$ values are overall higher in DJF than in JJA (Fig.2), especially over oceans where the skill of the regression during JJA is very limited. This points to a weak link between the large-scale circulation and surface temperature variability over oceans during summer. This is consistent with a stable surface boundary layer, since SSTs are often colder than the overlying air during summer. Selecting Z500 variability as covariate makes our approach less applicable for the tropics and subtropics, because Z500 variability there is low compared to the extratropics. $DR$ values are thus mostly low ($< 0.4$) below 40°N, except in the equatorial Pacific during DJF. The high values in the Pacific are likely a result of the strong influence of ENSO-related SST variability on the tropical large-scale circulation.

For the regionalisation, we only keep grid points where the $DR$ is $\geq 0.4$ (about 45% of the total area in DJF, and 11-14% in JJA). The distributions of silhouette coefficients show a peak at $K^* = 16$ (JJA cold spells), $K^* = 21$ (DJF warm spells) and $K^* = 26$ (JJA warm spells) (Figure A4). The peaks are not particularly sharp, but our results do not change substantially, when considering slightly different $K^*$ values. For DJF cold spells, the maximum silhouette coefficient is at $K = 44$, but the silhouette coefficients are largely constant beyond $K = 23$, a value we choose for $K^*$ to make the regionalisation as parsimonious as possible.

Consistent with the $DR$ maps, most regions are located in the mid- to high-latitudes, and over continents in JJA (Figure 3). Regions are generally highly coherent in space (also true without applying the dbscan-based filtering; not shown), except for three regions (12 on Figure 3-b, 15 on Figure 3-d and 10 on Figure 3-d). This may point either to teleconnections or to uncertainties in the regionalisation. In the tropics and subtropics, there are similarities in the regionalisation between warm and cold spells in DJF. In both cases, the algorithm detects a region along the equatorial Pacific Ocean, most likely related to ENSO and its large imprint on the tropical circulation. The region even extends to the equatorial Indian Ocean for persistent warm spells. We also find similar regions in the subtropics (3, 4, 8 and 9 on Figure 3-a and 17, 13, 8 and 11 on Figure 3-a).

Regions are generally bigger in DJF than in JJA, for both cold and warm spells, especially in terms of zonal extent. They cover an average of 4.1-4.3 $10^6$ km$^2$ in DJF (excluding the handful of tropical regions) against 1.3-1.6 $10^6$ km$^2$ in JJA, with a mean zonal extent of 3000-3500 km in DJF against 1500-1700 km in JJA. These figures are naturally influenced by the choice of optimal region number, but nevertheless indicate a tendency for larger spatial footprints of persistent temperature extremes in DJF compared to JJA.

## 4 Synoptic conditions during persistent warm and cold spells

With regions now identified, we calculate the lengths of the analysis windows as described in Figure 1 and section 2 to assess the timescale(s) associated with the build-up and persistence of warm and cold spells. Results in Table 1 (see also Figure A3) show that in all cases, persistent warm and cold spells develop over a 1-4 week period, with an average window length of 2-2.5 weeks. These numbers do not noticeably change for a 4-week window, and are only slightly lower for a 2-week window (average of 1.5-2 weeks; see Tables A1 and A2). There is however a large variability in the analysis window length from one event to the

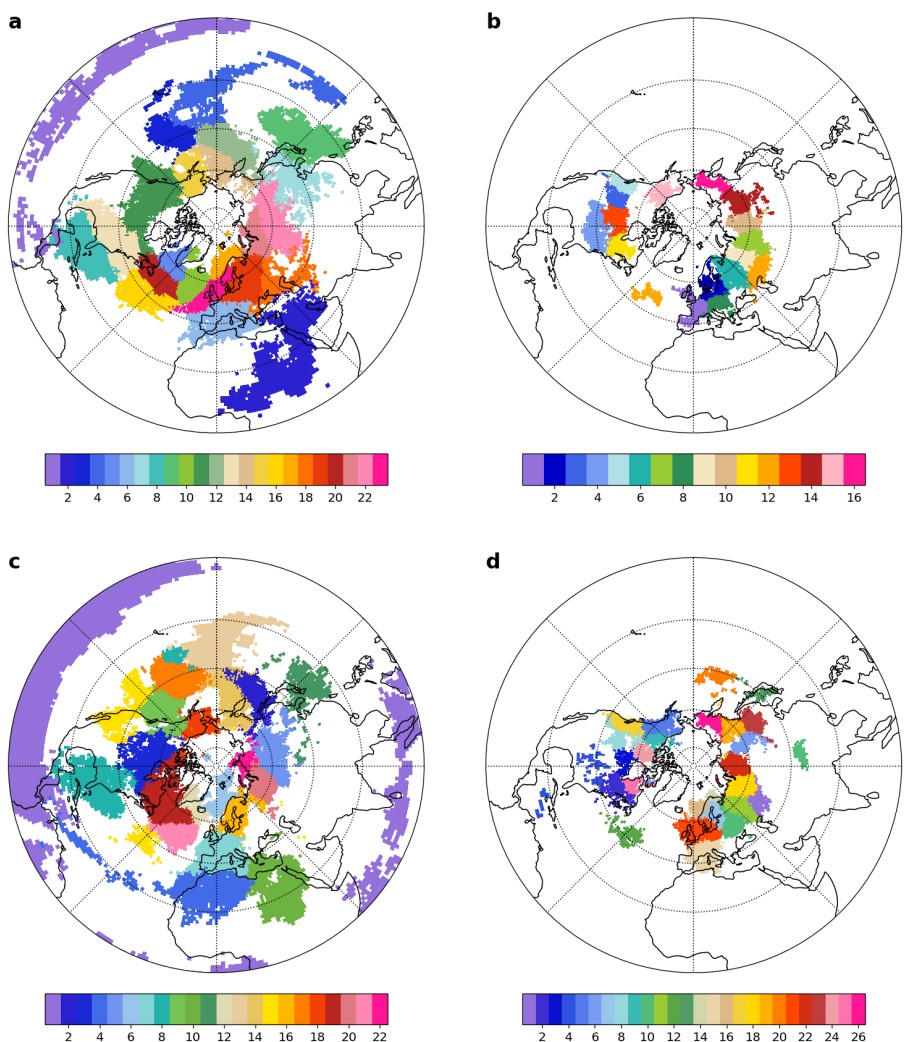

**Figure 3.** Regionalisation results: (a) DJF cold spells, (b) JJA cold spells, (c) DJF warm spells and (d) JJA warm spells.

other (coefficient of variation ≈ 0.5 for most regions). Note that because we calculate synoptic anomaly maps for each event
separately (section 2.2.3), longer events do not get more weight than shorter ones in the results.

In the following, we analyse the circulation and the temperature budget anomalies during DJF and JJA warm and cold spells
separately. We refrain from discussing the particulars of any one of the regions specifically, instead focusing on the robust
features that can be found across all regions or sub-groups of regions.

**Table 1.** Analysis window length (in days) for warm and cold spells: inter-region range and median (between brackets; see definition in Figure 1). For DJF, tropical regions 1 and 4 (Figure 3-a) and 1 (Figure 3-c) are excluded.

|       | Cold        | Warm       |
|-------|-------------|------------|
| **DJF** | 12-20 (16.5) | 11-25 (18) |
| **JJA** | 8-17 (13)   | 12-27 (17) |

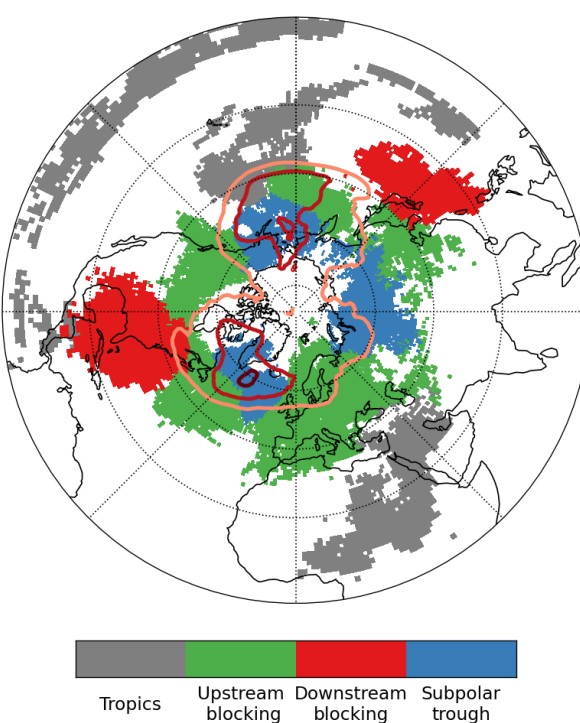

**Figure 4.** Classification of extratropical DJF cold spell regions according to their associated concurrent main circulation anomalies: upstream blocing (green), downstream blocking (red) and subpolar lows (blue). Tropical regions (excluded from the discussion) are shown in grey. DJF-mean blocking frequency is also indicated by the thick contours (2.5, 5 and 7.5%).

## 4.1  DJF cold spells

Comparing circulation anomaly maps across regions, we visually identify three main groups of regions with similar circulation anomalies during persistent DJF cold spells (tropical and sub-tropical regions being excluded from the analysis; Figure 4). The first group consists of 11 regions where cold spells are linked to upstream blocking, either to the west or to the northwest. For the three regions of the second group, blocking is also present, but downstream to the northeast. Finally, the third group consists of five high-latitude regions where cold spells are associated with a southward migration of the polar jet.

### 4.1.1 Upstream blocking

Regions where cold spells occur alongside upstream blocking are mainly located north of 40°N on each side of the North Atlantic and Pacific oceans (Figure 4). Most of them are located south and southwest of the two main blocking regions (Greenland and the Bering Strait). The distribution of average blocking anomalies for this group of regions shows enhanced blocking frequency and anomalously high Z500 to the west and northwest of the area with colder-than-average temperatures, which is located under a pronounced trough (Figure 5-a,b). The mid-latitude jet is displaced equatorwards of the cold region (Figure 5-b). Consistent with the circulation anomalies, cyclone activity and precipitation are suppressed downstream of the block, over the western half of the cold region, but enhanced downstream of the trough, especially in the left exit region of the jet streak (Figure 5-c). The near-surface temperature budget anomalies show that cold anomalies arise exclusively from cold air advection. The advection is partly compensated by near-surface diabatic warming, likely in the form of surface sensible heat flux and enhanced radiative warming from clouds in the eastern half of the region. At 850 hPa, cold air advection is partly compensated by adiabatic warming from the descent in the western part of the region (Figure A5-b,c).

### 4.1.2 Downstream blocking

Persistent cold spells can also occur upstream of a block, as in the three regions located on the eastern margins of North America and Asia, between 20°N and 40°N (see Figures 3-a and 4). Though the average anomalies are quite noisy due to the small number of regions, some robust features emerge. Cold anomalies occur below a zonally elongated trough that stretches to the southwest and south of a strong atmospheric block (Figure 6-a,b). A jet streak is present along the southern edge of the trough (Figure 6-b). As in the previous case, precipitation and cyclone frequency are below average in the western half of the trough and below the downstream block, but higher in the eastern half of the domain (Figure 6-c). Cold surface anomalies are again caused by substantial near-surface cold advection of several Kelvin per day (Figure 6-d), large values almost surely related to the steep zonal temperature gradient between the cold continents to the west and the relatively warm oceans to the east. The advection of cold and dry continental air masses over the ocean is then partly balanced by large positive surface heat fluxes found in the diabatic term (Figure 6-f). This picture is consistent with the precipitation anomalies. Dry conditions prevail over the western, continental half, due to large-scale descent and advection of cold and dry air. Over the eastern, oceanic half, wet conditions prevail, as cold air advection over the warm ocean surface increases surface sensible and latent heat fluxes along with the overall baroclinicity, which combines with quasi-geostrophic ascent to create an environment conducive to cyclone formation.

### 4.1.3 Subpolar troughs

The final group of regions is restricted to the high latitudes primarily below the main blocking centers in the North Atlantic and Pacific oceans, but also in northern Siberia, between 45°E and 135°E. There, persistent cold spells occur in conjunction with suppressed blocking activity (Figure 7-a). A high is sometimes present to the west of the cold region, which is located under a deep trough, while the polar jet is shifted equatorwards (Figure 7-b). Although the regions are of a similar size to ones from

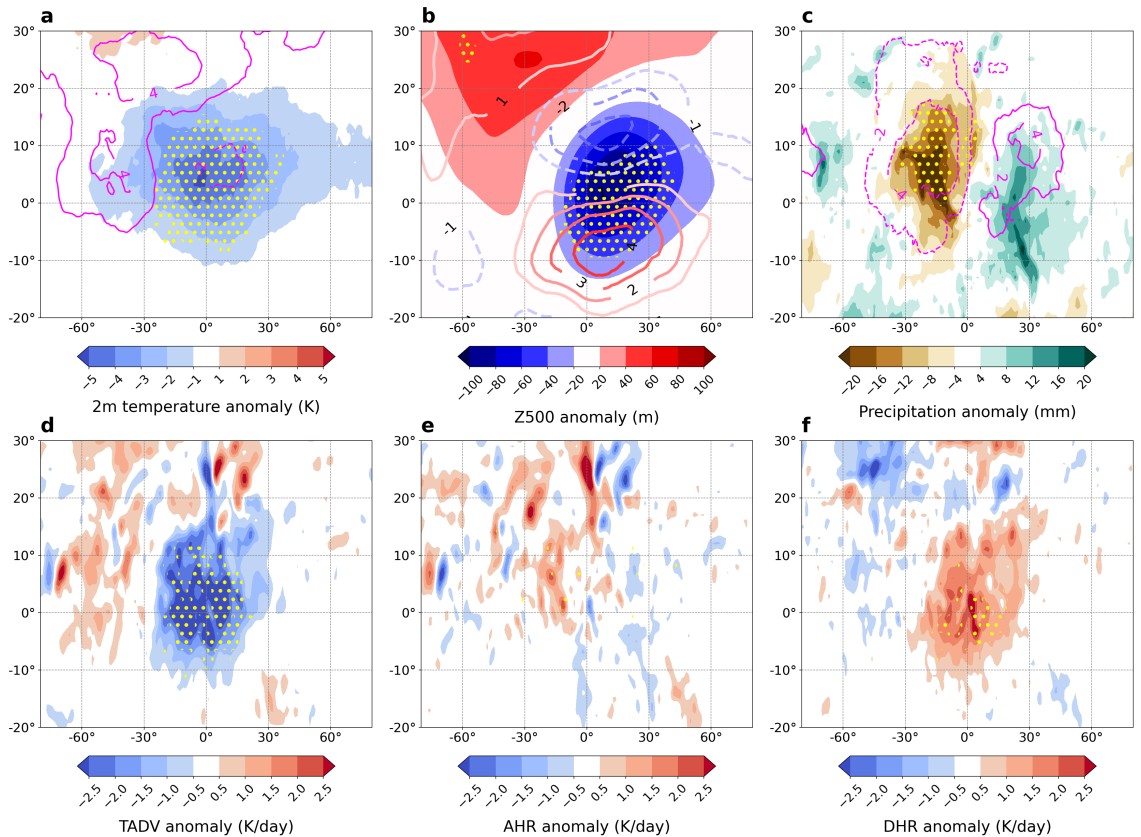

**Figure 5.** Average anomalies for regions where DJF cold spells are linked to upstream blocking (shown in green on Figure 4): (a) 2-meter temperature (shaded contours) and blocking frequency (contour lines, in %), (b) Z500 (shaded contours) and 200 hPa wind speed (contour lines, in m/s), (c) total precipitation (shaded contours) and cyclone frequency (contour lines, in %), (d) near-surface temperature advection, (e) near-surface adiabatic heating rate and (f) near-surface diabatic heating rate. Yellow hatching in all panels indicates areas where anomalies of shaded fields are of the same sign and statistically significant at the 10% level for more than two-thirds of the regions.

other groups, statistically significant cold anomalies extend over a much larger range of longitudes, and their magnitude tends to be larger as well (Figure 7-a). In keeping with the low specific moisture levels at high latitudes in winter, precipitation and cyclone frequency anomalies are quite small, though their pattern is similar to previous cases (Figure 7-c). The anomalies of temperature budget terms are noisy, probably because of orography, but the temperature balance remains similar to the other two regions groups. Cold advection dominates the temperature tendency equation, partly balanced by diabatic warming, likely resulting from surface-to-atmosphere heat transfers and enhanced radiative warming below clouds in the east of the region. Note that the subpolar regions on Figure 4 are far enough from the pole that cold advection from higher latitudes is possible. Farther north, advection plays a smaller role and radiative cooling drives the negative temperature tendencies (Messori et al., 2018).

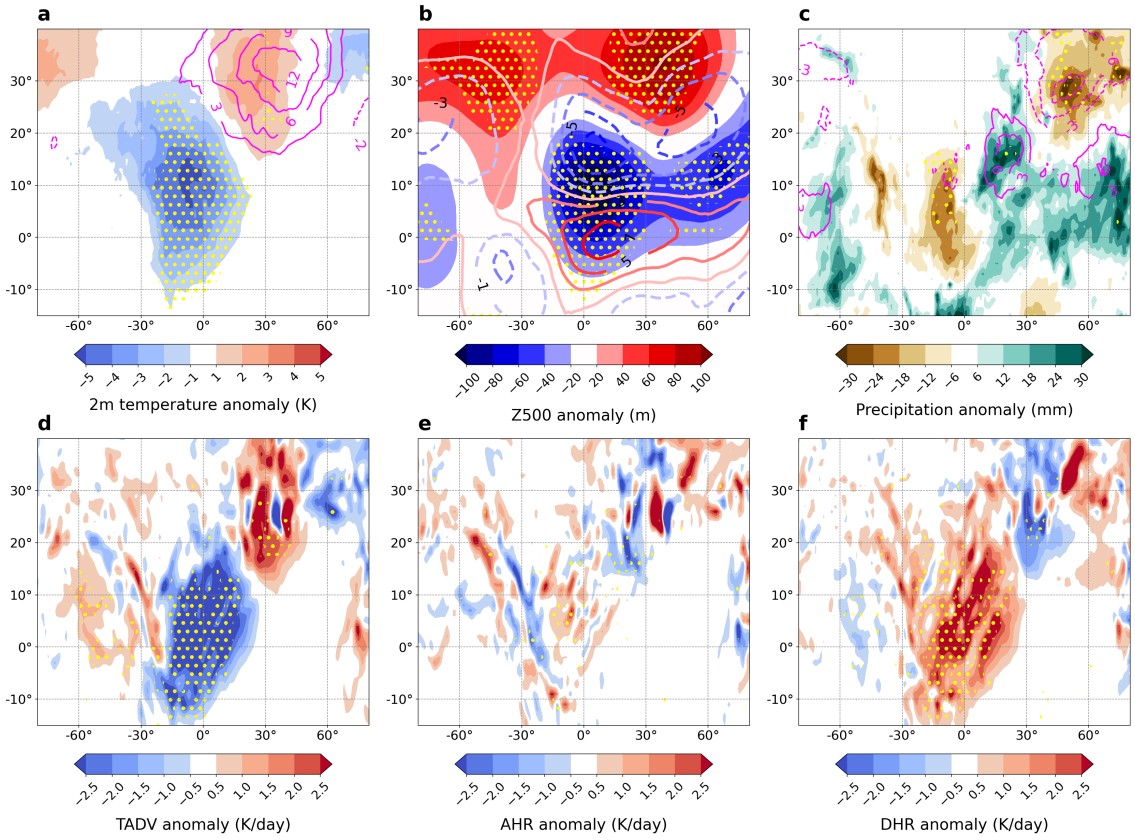

**Figure 6.** Same as Figure 5, but for regions where cold spells are linked to downstream blocking (shown in red on Figure 4).

## 4.2 JJA cold spells

Persistent cold spells in summer show very consistent circulation features across all regions, similar to the "upstream blocking" group for DJF cold spells (Figures 5 and 8). An anomalous ridge (often flagged as a block) is present directly to the west of the cold region. The cold region is under a trough (Figure 8-b). The jet is again meridionally amplified and displaced equatorwards to the south of the cold anomalies (Figure 8-b). Precipitation and cyclone frequency anomalies are shifted westward compared to those from Figure 5. Wet anomalies are located directly over the eastern half of the cold region, and dry anomalies are centred between the ridge and the trough, hardly overlapping with the cold region itself (Figure 8-c). Thus, while cold/dry conditions dominate in DJF, we find that in JJA cold/wet conditions are more prevalent.

Temperature budget anomalies show an important difference with DJF cold spells. While temperature advection is still the main contributor to negative temperature tendencies, adiabatic cooling also plays a significant role directly below the trough (Figure 8-d,e). At 850 hPa the contribution from adiabatic cooling is even more important (though not as statistically significant) than cold advection in the eastern half of the cold region (Figure A5-d,e). Negative temperature tendencies are partly balanced by

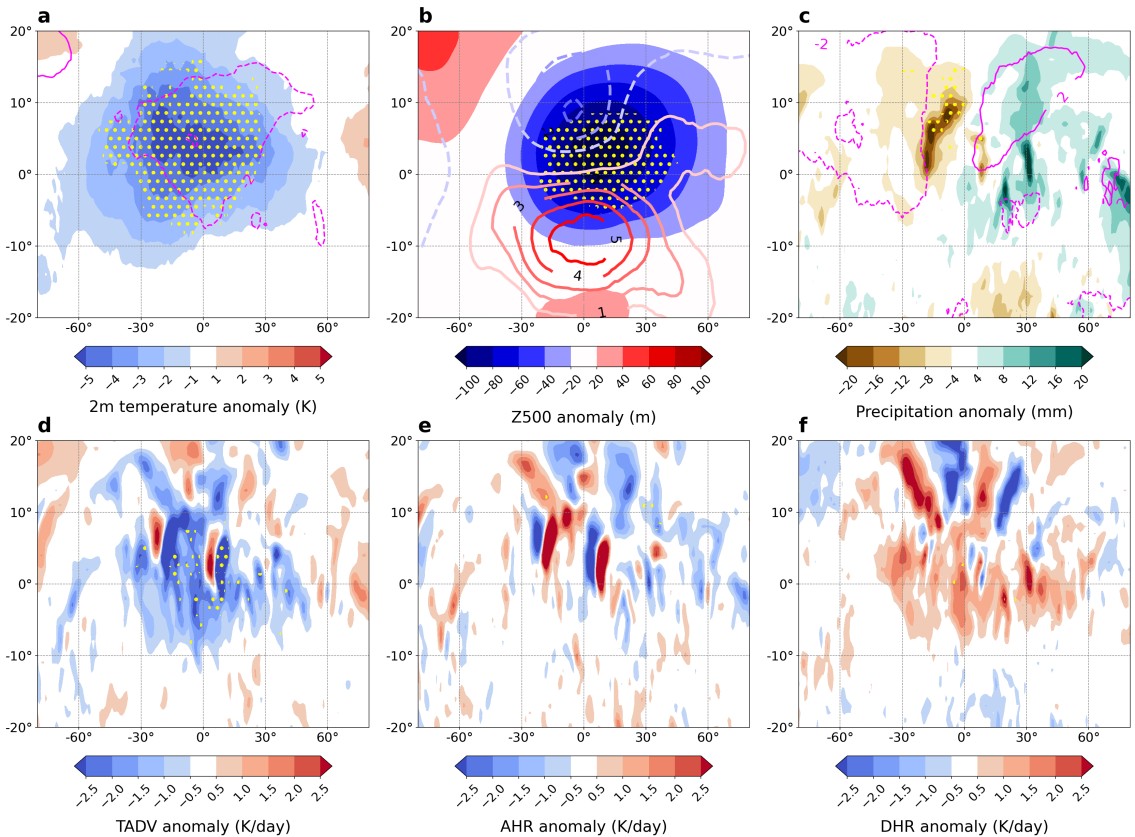

**Figure 7.** Same as Figure 5, but for regions where cold spells are linked to polar lows (shown in blue on Figure 4).

the diabatic term, in which surface heat fluxes probably dominate, though a contribution from the longwave budget (decreased nighttime cooling below clouds) is also possible (Figure 8-f).

### 4.3 DJF warm spells

Synoptic conditions associated with persistent DJF warm spells in the extratropics are all characterised by significant positive Z500 anomalies (partly) above the warm region (Figure 9-a,b). This anomalous ridge provides for significant warm advection at the surface, which is only partly balanced by the diabatic term (Figure 9-d,f). Cyclone frequency anomalies are opposite to those found during cold spells. The western and central parts of the warm region experience wetter-than-average conditions with more frequent cyclones, while dry anomalies occur mostly in the eastern third, on the downstream side of the ridge (Figure 9-c).

The exact location of the ridge relative to the warm region varies quite a lot, in keeping with the direction of the background temperature gradient. The direction of the temperature gradient is indeed not always in the meridional direction, depending e.g. on land/ocean geometry the strongest temperature gradient might also be tilted or even zonally oriented (see Figure 10-a). In

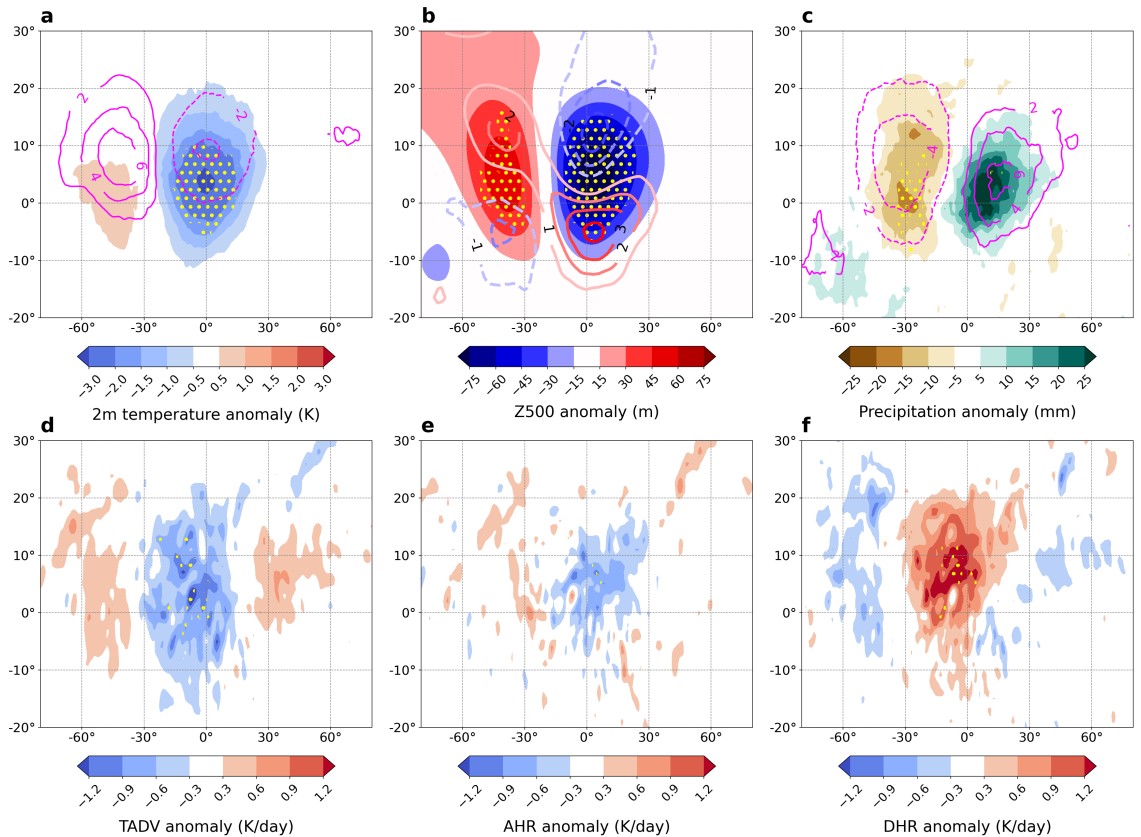

**Figure 8.** Same as Figure 5, but for all the JJA cold spell regions (shown on Figure 3-b).

most cost coastal regions (eastern and western Eurasia, Eastern North America), the anomalous warm advection comes from the neighbouring ocean. This means that the persistent ridge is centred to the south/southeast of the warm region in Western Europe/Scandinavia/coastal Siberia (regions 7, 16 and 22; Figure 3-c), but to the north/northeast in Eastern Asia (regions 14 and 19) and Eastern North America (regions 2 and 8). For all other regions, warm advection primarily comes from the south (Figure 10-a), and the ridge is located directly east of the warm region.

In many mid-to-high latitude regions, the anomalous ridge is often flagged as a block (see hatching on Figure 12-a). This is especially true around and to the west of Greenland and the Bering Strait, where blocking is most frequent in DJF (Figure 4). In Western Europe and Scandinavia, by contrast, ocean-to-land warm advection results from the subtropical high shifting polewards, in a positive North Atlantic Oscillation-like situation (not shown). Blocking also seems relevant for regions 4 and 10 in North Africa and the Eastern Mediterranean (Figure 3-c). Warm spells in these two regions indeed occur in conjunction with strong blocking over the Labrador Sea (not shown), which may possibly cause a ridge over North Africa by triggering a downstream wave train (Figure A6).

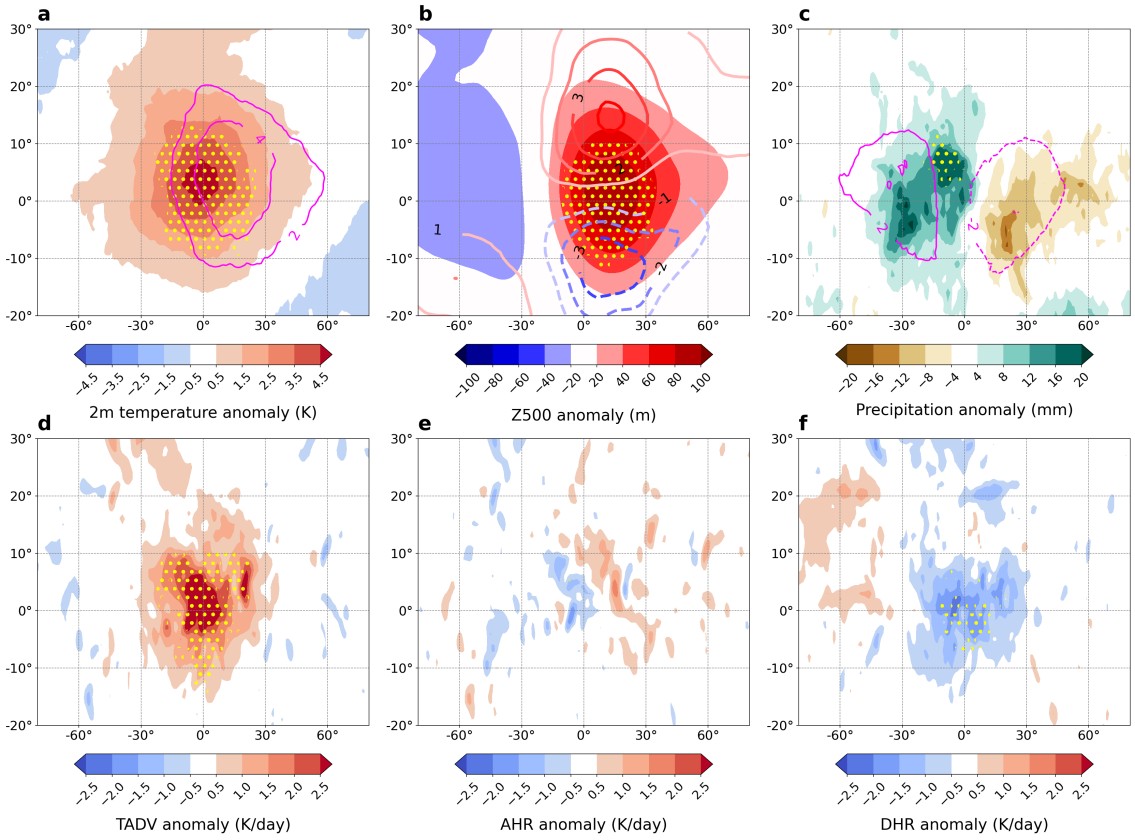

**Figure 9.** Same as Figure 5, but for all the DJF warm spell regions (region 1 excluded; see Figure 3-c).

## 4.4 JJA warm spells

In JJA, persistent warm spells occur for all regions directly below an anomalous ridge (Figure 11-a,b). Almost everywhere above 45°N, this ridge is often identified as a block (Figure 12-d). A jet streak is present polewards of the ridge (consistent with a poleward shift of the mid-latitude jet; Figure 11-b), though in a handful of cases a split jet is present (regions 7, 10, 14 and 21 over Scandinavia, and regions 1, 6, 18 and 23 over Siberia; see Figure 3-d). Dry anomalies with suppressed cyclone activity prevail over most of the warm region (but mostly east and south; Figure 11-c), while precipitation is on average slightly enhanced upstream of the ridge (a signal that is not robust across regions, however). Warm anomalies at the surface come from a mix of temperature advection (essentially from lower latitudes; Figure 10-b), and adiabatic warming resulting from large-scale subsidence (Figure 11-d,e). Advection dominates, especially in the western half of the region, while adiabatic warming is at least as important in the eastern half, where large-scale descent should be strongest. For some regions, especially at lower latitudes (40-45°N), adiabatic warming is even negligible. At 850 hPa, however, adiabatic warming largely exceeds advection (Figure A7-d,e). Note also that in a handful of regions, surface advection is slightly negative during warm spells, and warming results exclusively from the adiabatic term (*e.g.*, Pacific Northwest and Alaska; Figure 10-b). The residual diabatic effects are overall

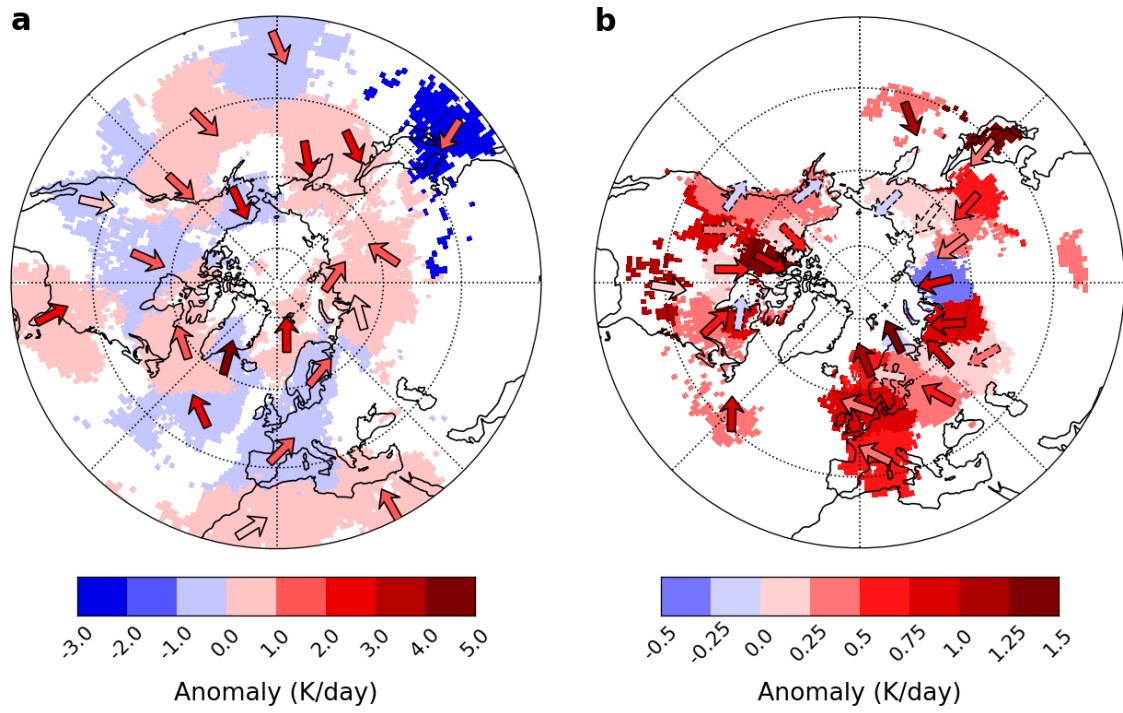

**Figure 10.** Near-surface adiabatic warming rate (shaded) and temperature advection during persistent warm spells in (a) DJF and (b) JJA, averaged over each region. Arrows indicate the approximate direction of temperature advection $\left(-u\frac{\partial T}{\partial x}, -v\frac{\partial T}{\partial y}\right)$ and their color the corresponding anomaly (in K/day). Dashed arrows indicate regions where temperature advection is smaller than the adiabatic warming term.

robustly negative (Figure 11-f), implying that enhanced sensible heating of the lower troposphere from (shortwave) radiative warming of the ground is largely balanced by increased (longwave) radiative cooling. The diabatic term is more negative than in winter (Figure 9-f), when ground temperatures are lower and enhanced cloud cover to the west of the regions result in longwave warming of the surface.

## 4.5 Synoptic drivers of persistent warm and cold spells

Persistent summertime temperature extremes are almost always associated with a meridionally oriented circulation (wavy jet) which favours deep equatorward intrusions of cold air or poleward intrusions of warm air (Figures 5-9 and 11). In winter, the situation is more complex, since temperature also strongly varies with longitude due to land sea contrasts. DJF heatwaves in coastal areas are, for instance, associated with persistent zonal flows (Figure 10-a). Advection dominates the temperature budget during cold spells in winter and summer (Figures 5-8, panels d) and during warm spells in winter (Figure 9-d). Warm advection

also contributes significantly to positive temperature tendencies in summer (especially at the surface and in the lowest-latitude regions; Miralles et al. (2014); Pfahl (2014)) but is sometimes exceeded by adiabatic warming (Figures 10-b and 11-d,e).

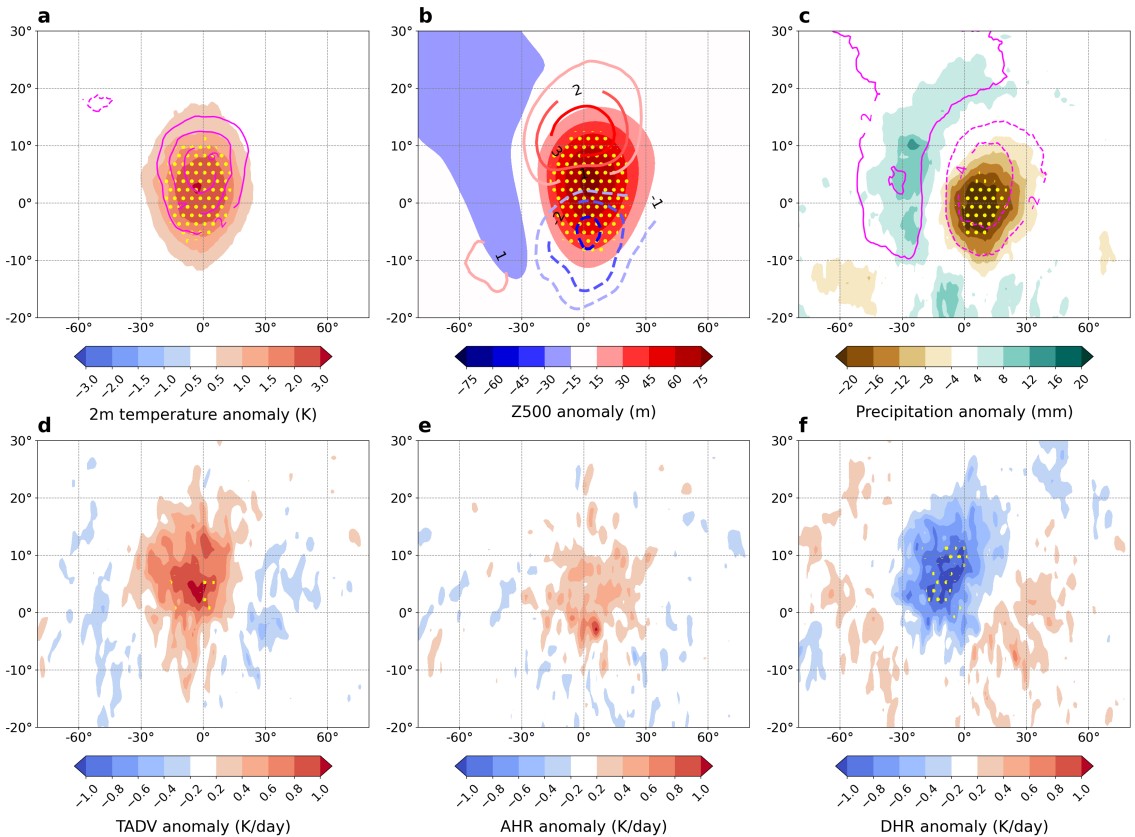

**Figure 11.** Same as Figure 5, but for all JJA warm spell regions (shown on Figure 3-d).

Our analysis of synoptic conditions and temperature budget terms during persistent warm and cold spells highlights several key ingredients for the persistence of temperature anomalies, which we now discuss in turn.

### 4.5.1 Atmospheric blocking

First, atmospheric blocking plays an important role in the persistence of both warm and cold spells in summer and winter. Our hemispheric perspective highlights the various influences of blocking that have been previously discussed (e.g., Carrera et al., 2004; Buehler et al., 2011; Pfahl and Wernli, 2012; Whan et al., 2016; Xie et al., 2017; Brunner et al., 2018; Jeong et al., 2021; Kautz et al., 2022). In most regions, cold spells occur most frequently downstream of persistent blocking anomalies, which drive strong cold advection through the meridional amplification of the jet. In other regions, cold spells are linked

to downstream blocking through enhanced Rossby wave breaking/meridional flow amplification upstream of the block (e.g., Takaya and Nakamura, 2005). This mechanism seems relevant on the eastern side of continents, namely between 20-40°N in eastern North America and in eastern Asia. Both areas are upstream of the two main blocking regions in the North Atlantic and North Pacific oceans (Figure 4). Our results also show that cold advection downstream of a block is a frequent pattern during

persistent summer cold spells (Figure 8). In summer, however, blocks have a weaker and more local effect on temperatures downstream. This is likely related to the weaker meridional temperature gradient and higher wavenumber flow in summer. Blocks also tend to occur at higher latitudes in summer (Steinfeld and Pfahl, 2019).

Blocking is also critical for the persistence of winter and summer warm spells. In summer, the classic picture of a block overlying the warm surface anomalies applies to most of the identified regions, especially above 45°N (Figure 12-d) (Perkins, 2015; Röthlisberger and Martius, 2019). Around 35-45°N, however, persistent highs are more relevant than blocks for summer warm spells. Though this result may be affected by the choice of blocking identification algorithm (one based on Z500 gradients rather than PV anomalies would likely yield higher blocking frequencies in the mid-latitudes), there is good evidence that persistent subtropical ridges rather than blocks are responsible for warm spells around 35-45°N (Della-Marta et al., 2007; Perkins, 2015). Blocks also matter for winter warm spells (Figure 9). Their associated circulation anomalies can result in the warm surface advection that seems essential to the onset and persistence of winter warm spells (Figure 9-d). The role of blocks in winter seems, however, mainly limited to the Northwestern Atlantic and Northwestern Pacific oceans (Figure 12-a).

Atmospheric blocks are by definition persistent features (Schwierz et al., 2004), but their typical lifetime is usually 5-10 days, and longer blocks are extremely rare (e.g., Nabizadeh et al., 2019). The persistence of individual blocks is known to be fueled by upstream latent heating (Steinfeld and Pfahl, 2019), but recurrent blocks are likely also key to the persistence of cold or warm spells over sub-seasonal timescales. For instance, the month-long heatwave over the Baltic in June-July 2022 (Tuel et al., 2022) and the 2010 Russian heatwave (Drouard and Woollings, 2018) were both related to several successive blocking systems.

### 4.5.2 Recurrent Rossby wave packets

The persistent circulation patterns during warm and cold spells can also be part of recurrent Rossby wave packets (RRWPs). In these synoptic-scale wave packets, individual troughs and ridges amplify repeatedly at the same longitudes, leading to persistent circulation anomalies frequently associated with extreme surface weather (Röthlisberger et al., 2019; Ali et al., 2022). To highlight the role of RRWPs, for each region, we look for statistically significant positive R-metric anomalies during event analysis windows in a longitude interval covering the region and up to 30°W of the region's westernmost points (to identify potential upstream RRWPs). Results show that persistent spells in a large number of regions are associated with significant RRWP activity (Figure 12), most consistently across seasons and spell types over central and southern Europe. Our results for winter cold spells and JJA warm spells are highly consistent with those of Röthlisberger et al. (2019).

We find strong relationships between RRWPs and persistent summer warm spells in Western North America, the North Atlantic Ocean, Western Europe, and northwards of the Black and Caspian Seas (Figure 12-d). Interestingly, several of these regions exhibit no significant blocking anomalies during JJA warm spells. A combination of blocking directly overhead and RRWPs could therefore explain much of the circulation persistence across regions. RRWPs, however, sometimes are related to up- or downstream blocking (Röthlisberger et al., 2019).

The link between RRWPs and DJF cold spells is evident in the North Pacific, the Caribbean and the Mediterranean Basin (Figure 12-a). JJA cold spells are associated with enhanced RRWP frequency in Western Europe only (Figure 12-b), but we find a significant link of DJF warm spells to RRWPs in Western North America, the North Atlantic and the Eastern Mediterranean

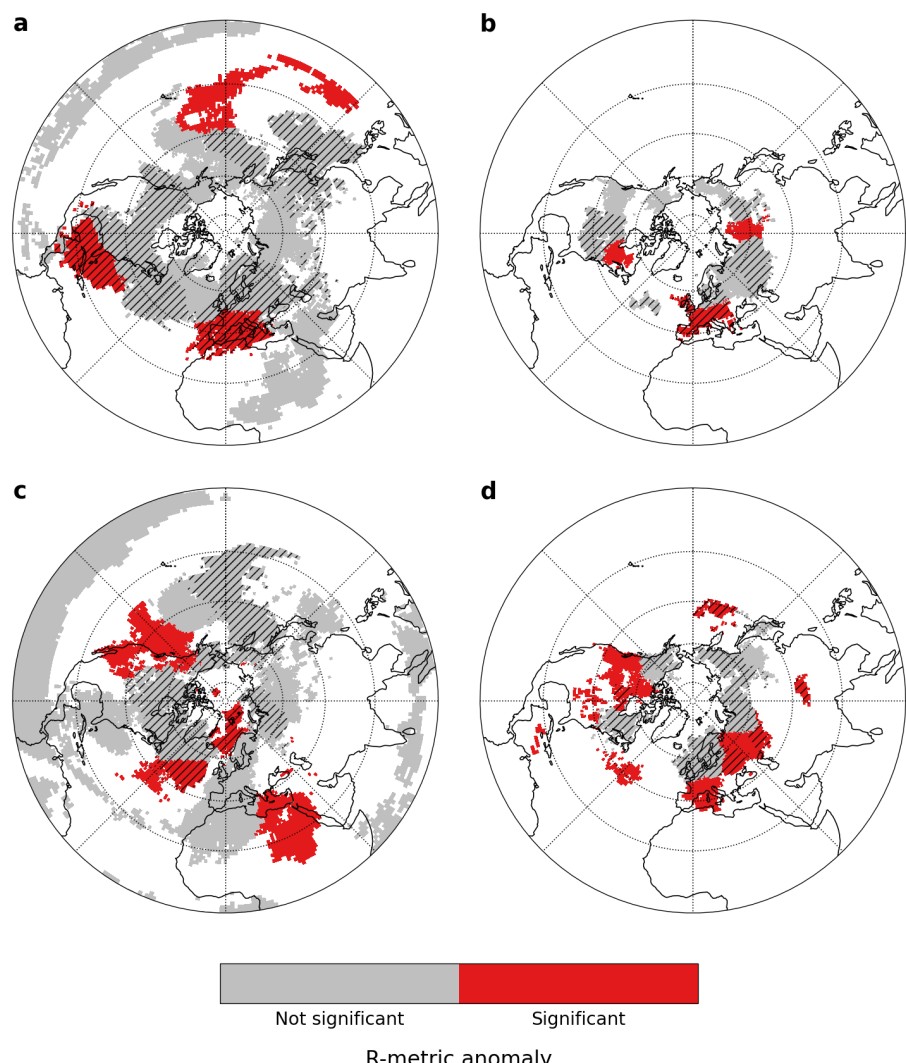

**Figure 12.** Significance of R-metric and blocking anomalies during (a) DJF cold, (b) JJA cold, (c) DJF warm and (d) JJA warm spells. Red shading indicates regions with statistically significant positive R-metric anomalies during the corresponding events. For each region we look for R-metric anomalies above the region and up to 30°W of its westernmost points. Hatching indicates significant positive blocking frequencies up- or downstream of the region (a-c) or above the region (d) (not restricted to 30°W).

(Figure 12-c). In the Eastern Mediterranean, these RRWPs may develop downstream of the Greenland blocking that occurs during warm spells (Berkovic and Raveh-Rubin, 2022). In Western North America and the North Atlantic (region 15 on Figure 3-c, one of the few non spatially-coherent ones), RRWPs may be part of a persistent wave train associated with the so-called "North American winter temperature dipole" (Singh et al., 2016).

### 4.5.3 Land-atmosphere feedbacks

It has long been argued that land-atmosphere feedbacks can be essential for the persistence of summer warm spells (e.g., Lorenz et al., 2010; Mueller and Seneviratne, 2012; Perkins, 2015; Bartusek et al., 2021; Martius et al., 2021). Dry soils increase sensible heat fluxes to the detriment of latent heat fluxes, leading to lower tropospheric warming and drying, which further dries the soils and stabilises the atmospheric column (Seneviratne et al., 2010; Miralles et al., 2019). This is a well-known mechanism whose role has been highlighted in several recent persistent heatwaves (e.g., García-Herrera et al., 2010; Dirmeyer et al., 2021). All our regions experience precipitation deficits during JJA warm spells, most of which are statistically significant (Figures 11 and A8-a). Over continents, these precipitation deficits translate into increased sensible heat fluxes upwards from the surface (Figure A8-b), which enhance the positive temperature anomalies and partly balance the radiative cooling seen in the negative diabatic tendencies (Figure 11-f). Regions with particularly high precipitation deficits (Western North America, Western Europe, Kazakhstan and Eastern Siberia) also tend to exhibit the largest sensible heat flux anomalies (Figure A8). Land-atmosphere feedbacks thus seem important for the persistence of summer warm spells, especially in transition regions where soils are neither too dry or too wet (e.g., Western Europe or Scandinavia), where they have already been show to play a role (e.g., Dirmeyer et al., 2021). It remains unclear, however, to what extent dry soils can trigger persistent heatwaves on their own, or whether a dynamical forcing is required (e.g., Domeisen et al., 2023). In our case, we do not find significant precipitation deficits before the onset phase the persistent warm spells (not shown).

### 4.5.4 Subpolar troughs

Finally, DJF cold spells in several high-latitude regions are linked to persistent subpolar troughs that stretch across a wide zonal band (Figures 4 and 7). Our analysis cannot determine to what extent these troughs are part of Rossby wave trains or consist of isolated vortices, which have been documented in the Arctic where the beta effect is weak (Cavallo and Hakim, 2010). They could also occur in the framework of annular modes of variability (Arctic Oscillation). Still, Rossby wave behaviour at high latitudes supports the persistence of subpolar troughs (Woollings et al., 2022). Indeed, at high latitudes, phase and group velocities are much closer than in the mid-latitudes. Rossby wave dispersion is consequently limited. Additionally, the troughs could be part of locally triggered waves, or of waves triggered remotely from the tropics (by deep convection) or mid-latitudes (through baroclinicity). Among the waves initiated at lower latitudes, only the longest ones, with slower phase speed, can reach the Arctic (Hoskins and Ambrizzi, 1993). That and the weak eddy diffusion would favour long lifetimes for subpolar cyclones. The most extreme cold spells, especially in high-latitude Eurasia, could also be related to sudden stratospheric warming events, which impact surface conditions for weeks in a row (e.g., Baldwin et al., 2021). Finally, Röthlisberger and Papritz (2023a) found that at high latitudes, slow, near-surface diabatic cooling was the dominant cause of cold extremes. We find by contrast that, in the Eulerian budget, near the surface and averaged over the multi-week time-scales, diabatic cooling does not contribute significantly to persistent cold spells at high latitudes. Reasons for this discrepancy may be that our regions do not exactly capture those with strong diabatic influence in Röthlisberger and Papritz (2023a) (northeastern Siberia and North America; see

Figure 3-a), and that the spatial and temporal averaging weaken the diabatic contribution (Röthlisberger and Papritz (2023a)
considered daily extremes at grid-point scale).

## 5 Discussion

### 5.1 Model goodness-of-fit

As seen on Figure 3, model goodness-of-fit is unequally distributed across space and time. High $DR$ values are largely concentrated in the mid- to high latitudes (polewards of 30°N in DJF and of 40°N in JJA), where Z500 variability is considerably larger than in the tropics. Our selected Z500 EOFs thus primarily capture extratropical circulation variability. Our choice of Z500 as a covariate for the quantile regression is consequently not well-suited for the tropics, where the streamfunction or normalised Z500 fields would yield better results. A separate model for the tropical band would also be preferable, as the tropics account for a much larger area than the extratropics, and therefore streamfunction/normalised Z500 EOFs may be biased in favour of tropical variability. Relatively small $DR$ values above the polar circle may similarly be related to the small area these regions occupy (relative to the mid-latitudes), which weighs against them in the EOF analysis. To avoid such issues, it is conceivable to create a regionalisation based on a distance that does not rely on covariates (like the Jaccard distance or an edit-type distance; Banerjee et al. (2022)). Such distances however use much less of the data than our quantile regression (only the extremes of interest) and may result in less stable and interpretable results.

The regression model also performs much better in DJF than in JJA, especially over oceans. The continent-ocean contrast is particularly sharp in JJA, when $DR$ values over oceans mostly remain below 0.4 (Figure 3-b,d). This is consistent with oceans being mostly cooler than the overlying air in summer, and resulting in frequent low-level inversions, which create a more stable boundary layer than over land, and make 2-meter temperatures over oceans less sensitive to variations in the large-scale circulation. The large thermal inertia of the oceans similarly weakens the dependence of surface temperatures to subseasonal-to-seasonal variability in the large-scale circulation. Over land, surface temperatures respond much faster to forcing from the circulation. Results on Figure 3-d are in line with those of Pfahl and Wernli (2012), who found that blocking was much less relevant for summer warm spells over oceans, except over the Northwestern Pacific and central North Atlantic where our model skill is also higher.

Other spatial contrasts on Figure 3 are not so easy to explain. It is notably unclear why differences in model skill between warm and cold spells exist. The ability of Z500 EOFs to capture the relevant atmospheric processes may differ regionally between warm and cold spells (due to the size, location or persistence of relevant circulation anomalies, for instance). Over land, model skill during warm spells may also be limited by the local influence of land-atmosphere feedbacks, which may not necessarily impact the large-scale circulation. Finally, topography and, in winter, snow cover, also probably influence the results. We see for example that model skill is low over the Rocky mountains in DJF. Snow cover perturbs the surface energy balance, while topography may block or steer the large-scale flow. Surface temperatures at high elevations also likely suffer from biases in ERA5.

## 5.2 Regionalisation

While our goal is not to discuss individual regions in detail, some general remarks can be made about the regionalisation results shown on Figure 3.

First, it is important to note that, as with any regionalisation approach, our results depend on the number of regions we select. The distance metric we use implies that regions (clusters) are areas where warm/cold spells are related to similar large-scale circulation patterns, and therefore where spells tend to occur at approximately the same time. The subsequent synoptic analysis showed that our choice of regions seems consistent with the corresponding physical drivers. Too few regions would likely have blurred the significant signals in e.g. blocking location. We also find that neighbouring regions generally experience warm/cold spells at different times. In most cases, events in neighbouring regions overlap (in time) by less than 20%. Still, our proposed regions are not necessarily the most relevant from an impacts perspective, notably because (i) the temperature threshold we consider are relative, not absolute, and (ii) the extremeness of the temperature anomalies is softened by averaging over large scales.

Second, the difference in the spatial footprint and mean zonal extent of regions between DJF and JJA, while certainly impacted by the choice of region number, is consistent with planetary waves having longer wavelengths in the cold season. This leads to larger temperature anomalies at the surface, e.g., the well-known North American dipole (Singh et al., 2016), which may correspond to regions 11 in Figure 3-a and 15 in Figure 3-c. Another important feature of the winter season is that temperature gradients are not only meridional, but also zonal. This is much less the case in summer when meridional gradients dominate. Zonal circulation can therefore lead to strong cold advection in winter. Our temperature budget analysis shows that cold spells are primarily advection-driven, in both winter and summer. The cold air in summer essentially comes from the high latitudes, and a strong meridional circulation is required to bring it equatorwards, consistent with rather narrow regions in Figure 3-b. By contrast, in winter, land/ocean temperature contrasts are steep, and zonal circulations can bring about strong cold (or warm) advection, consistent with the large zonal extent of some regions like e.g., region 13 on Figure 3-a. Locally, some particularly zonally elongated regions like regions 6 and 23 over Europe in Figure 3-a, are also likely related to strong zonal circulations (in this case the North Atlantic Oscillation) that can lead to persistent surface temperature patterns. Further differences in the regionalisation between winter and summer may be due to the seasonality of blocking (Steinfeld and Pfahl, 2019) or RRWPs (Röthlisberger et al., 2019) (in terms of location, extent and intensity).

Finally, it is not straightforward to compare our regionalisation results to previous findings, because (i) we take a hemispheric perspective, (ii) we look at long (3-week) cold and warm spells and (iii) we limit ourselves to regions where the quantile regression model skill is high. Most previous studies are indeed limited to specific areas (Europe in particular) and focus on shorter-term events, chiefly heatwaves. Figure A1 shows that while there is certainly a lot of overlap between our 21-day spells and more traditional 3-day extremes, the two are not the same, notably during summer (Figure A1-b,d), which makes a direct comparison difficult. Still, over Europe during summer, our results are in general agreement with those of Carril et al. (2008) who, from EOFs of monthly temperature anomalies, found a clear tripole structure between Northwestern Europe, the Euro-Mediterranean and Eastern Europe/Western Russia similar to regions 10, 15 and 21 on Figure 3-d. Stefanon et al.

(2012) and Pyrina and Domeisen (2023) likewise both found 6 spatial clusters across Europe based on the simultaneity of hot summer days, including a North Sea cluster comparable to region 21 on Figure 3-d, a Western European cluster (region 15), a Scandinavian cluster (region 7), an Eastern European cluster (region 10) and a Russian cluster (region 11). Some of the typical European heatwave patterns of Felsche et al. (2023) are also similar to our Figure 3-d results. Regarding cold spells, Xie et al. (2017) detected three main patterns of winter cold waves in North America (East, Northwest and Centre), two of which bear

some resemblance to regions 11 and 13 on Figure 3-a. Region 11 is also similar to the North American region of cold outbreaks triggered by sudden stratospheric warmings identified by Kretschmer et al. (2018).

The analysis of circulation anomalies during persistent warm and cold spells supports the statistical and physical relevance of our regionalisation. Identified regions are clearly different when it comes to the physical mechanisms responsible for persistent temperature anomalies. For instance, warm spells in the Euro-Mediterranean are linked to a persistent subtropical ridge, while

spells in Northwestern Europe are connected to blocking at higher latitudes (Figure 10-b; Carril et al. (2008); Sousa et al. (2018)). Noticeable similarities between our results and those of Röthlisberger et al. (2019) on the influence of RRWPs also suggest that our regionalisation is physically meaningful. We note finally that the size of many obtained regions is comparable to that of typical atmospheric blocks (Croci-Maspoli et al., 2007; Nabizadeh et al., 2019) and associated temperature anomalies (e.g., Carrera et al., 2004; Buehler et al., 2011; Whan et al., 2016; Schaller et al., 2018; Sousa et al., 2018). Given how relevant

blocking is for the persistence of extreme temperatures, this suggests that our choice of optimal region number is justified.

### 5.3 Other potential drivers of persistent warm and cold spells

We focused here on the regional, synoptic-scale patterns associated with persistent warm and cold spells in the extratropics. In particular, we did not discuss potential remote influences on the extratropical circulation that could modulate the likelihood and persistence of surface temperature extremes. Notable among these are sea-surface temperatures (SSTs) and the Madden-

520 Julian Oscillation (MJO). Both tropical and extratropical SSTs impact the extratropical circulation at sub-seasonal to seasonal timescales relevant for our study. Tropical SST anomalies (and the MJO) modulate deep convection and the resulting Rossby wave trains that propagate polewards (e.g., Lin and Brunet, 2018). Temperature extremes in the mid-latitudes are for instance known to be influenced by ENSO (e.g., Arblaster and Alexander, 2012; Li et al., 2017; Dai and Tan, 2019). The MJO also influences cold air outbreaks in North America (Moon et al., 2012) and East Asia (Abdillah et al., 2018). In the extratropics, SST

anomalies can also lead to stormtrack shifts or substantial regional circulation anomalies that impact surface weather (Hoskins and Karoly, 1981; Brayshaw et al., 2008). For example, North Atlantic SST anomalies matter for European summer heatwaves (Duchez et al., 2016; Mecking et al., 2019), and North Pacific SSTs modulate cold spells in North America (Li et al., 2017). Our regionalisation could therefore be used to identify the potential SST or tropical circulation anomalies associated with persistent warm and cold spells and relate them to the circulation anomalies around the warm/cold region. As it considerably reduces the

dimension of the problem, our regionalisation also makes it possible to adopt more time-intensive methods like causal analysis, and to include more than just classical modes of SST or tropical variability as potential covariates.

## 5.4 Limitations of the regional averaging

While the goal of our paper is to highlight the similarities in terms of synoptic-scale anomalies during persistent regional warm/cold spells across the northern hemisphere, there are some limitations relative to the averaging that must be mentioned.

First, whereas summer regions are generally confined to land (Figure 3-b,d), in winter this is not the case (Figure 3-a,c). As a consequence, averaged anomaly maps shown in Figures 5, 6, 7 and 9 mix together land and ocean grid points. We choose not to treat land and ocean separately, but one should remember that it may affect the results in the aforementioned figures. Since cold spells are advection-driven, land/ocean contrasts probably play a limited role, but for warm spells, surface heat fluxes from the ocean surface may be significant.

Second, averaging over regions of different shapes and sizes, and located at different latitudes or different points of the storm tracks, also impacts the results. There is variability in synoptic anomalies, both among different events for the same region, and among different regions, even ones that we grouped together (e.g., Figure 4). For instance, the location of a block can differ across events or regions (which may even impact the persistence of the corresponding spell). This we chose not to treat, opting instead for a simplified approach which relies on the regionalisation to reduce the problem dimension and easily highlight the similarities across different locations. The analysis of individual regions will be the subject of future work.

Last, we note that ERA5 is now available back to 1940 (83 years vs. the 42-year dataset used in this study). Considering this back-extension would allow to identify many more warm and cold spells and draw more robust conclusions.

## 6 Conclusion

This study introduces a regionalisation for Northern Hemisphere persistent warm and cold spells in winter and summer. The regionalisation is based on the association of persistent temperature extremes with large-scale circulation variability, which we assess with quantile regression. Between 16 and 26 coherent regions are identified for summer and winter warm and cold spells across the Northern Hemisphere. Our analysis highlights the key similarities and the diversity of processes responsible for persistent temperature extremes across regions and seasons. Cold spells systematically result from northerly cold advection. Warm spells are caused either by adiabatic warming (in summer) or warm advection (in winter). Persistent cold anomalies in winter are associated with blocking upstream (Europe, west coast continental US and eastern Asia), blocking downstream (east coasts of the continents) or polar troughs (Siberia, Alaska and Greenland). Persistent cold surface anomalies are associated with dry conditions in winter but wet in summer, and persistent warm anomalies with wet conditions in winter and dry conditions in summer. We also discuss some key mechanisms responsible for the persistence in temperature extremes: blocking and recurrent Rossby wave packets are important in all seasons, while land-atmosphere feedbacks matter for summer warm spells only. Our Eulerian perspective is useful to highlight synoptic conditions during persistent warm and cold spells, and thus to better understand the role of the atmospheric circulation. It would be interesting to extend our results with a Lagrangian analysis to investigate parcel origin and evolution during persistent spells.

|  | Cold | Warm |
|---|---|---|
| **DJF** | 10-16 (12) | 8-23 (12) |
| **JJA** | 5-13 (9) | 8-24 (12) |

**Table A1.** Analysis window length (in days) for 2-week warm and cold spells: inter-region range and median.

|  | Cold | Warm |
|---|---|---|
| **DJF** | 12-20 (16.5) | 11-25 (18) |
| **JJA** | 8-17 (13) | 12-27 (17) |

**Table A2.** Analysis window length (in days) for 4-week warm and cold spells: inter-region range and median.

*Code and data availability.* ERA5 reanalysis data can be downloaded from https://doi.org/10.24381/cds.adbb2d47. The code for blocking identification is available from https://doi.org/10.5281/ZENODO.4765560 (Steinfeld, 2021), and the code to calculate the R metric is available on GitHub (https://doi.org/10.5281/zenodo.5742810; Ali, S. M. (2021)). The code to reproduce our results and the regionalisation (at a 1° resolution) is available at https://github.com/Quriosity129/warm_cold_spells.

|  | Cold | Warm |
|---|---|---|
| **DJF** | 11-22 (16) | 9-23 (17) |
|  | 1.4-4.4 (2.9) | 1.2-3.4 (2.3) |
| **JJA** | 14-26 (19) | 11-24 (17) |
|  | 0.6-1.8 (1.0) | 1.0 - 3.3 (1.9) |

**Table A3.** Number of identified persistent spells (black), and ratio of number of 3-day events to number of persistent spells (blue), for each season and type of spell (warm and cold). For each case, we indicate the range across clusters and in brackets the corresponding median.

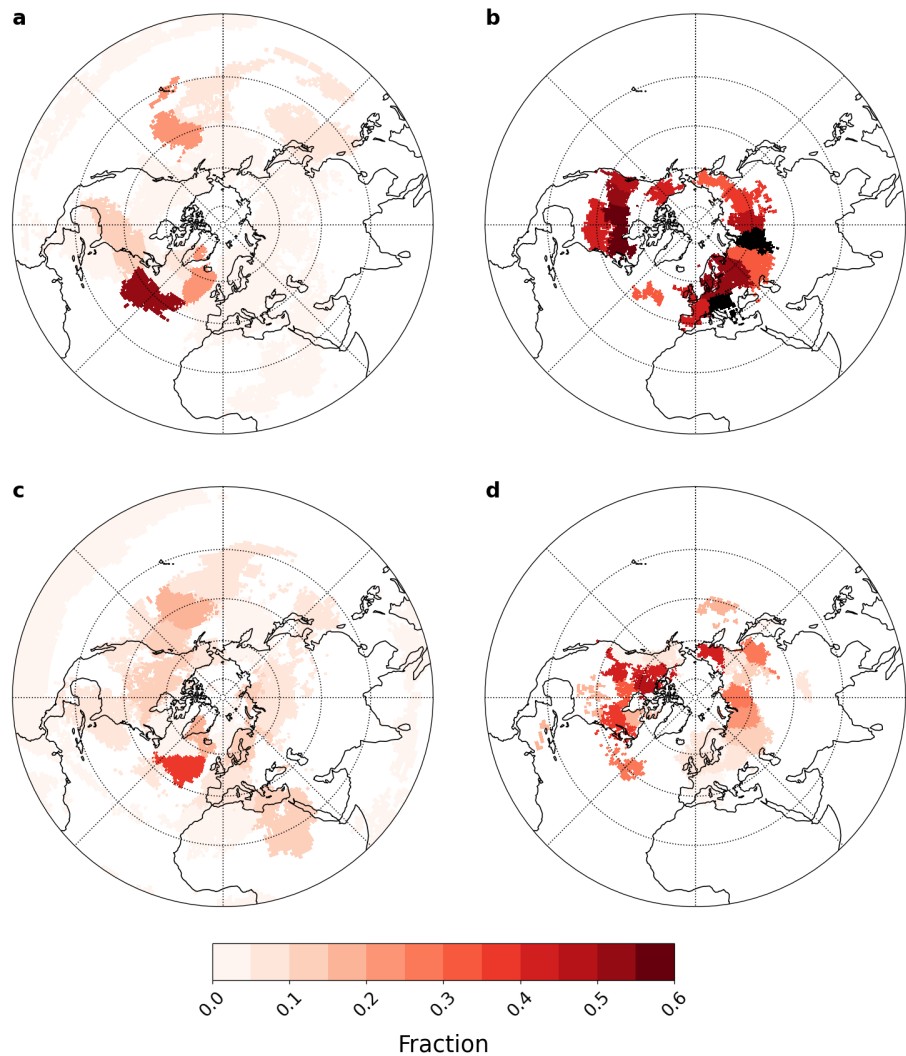

**Figure A1.** Fraction of 21-day persistent warm/cold spells which do not include a 3-day extreme warm/cold period, for (a) DJF cold spells, (b) JJA cold spells, (c) DJF warm spells and (d) JJA warm spells. 3-day extreme periods are defined based on continuous exceedances of the daily $5^{th}$/$95^{th}$ temperature percentiles in each region. In order to make sense of the values shown, we provide in Table A3 an idea of the numbers of persistent spells and 3-day events.

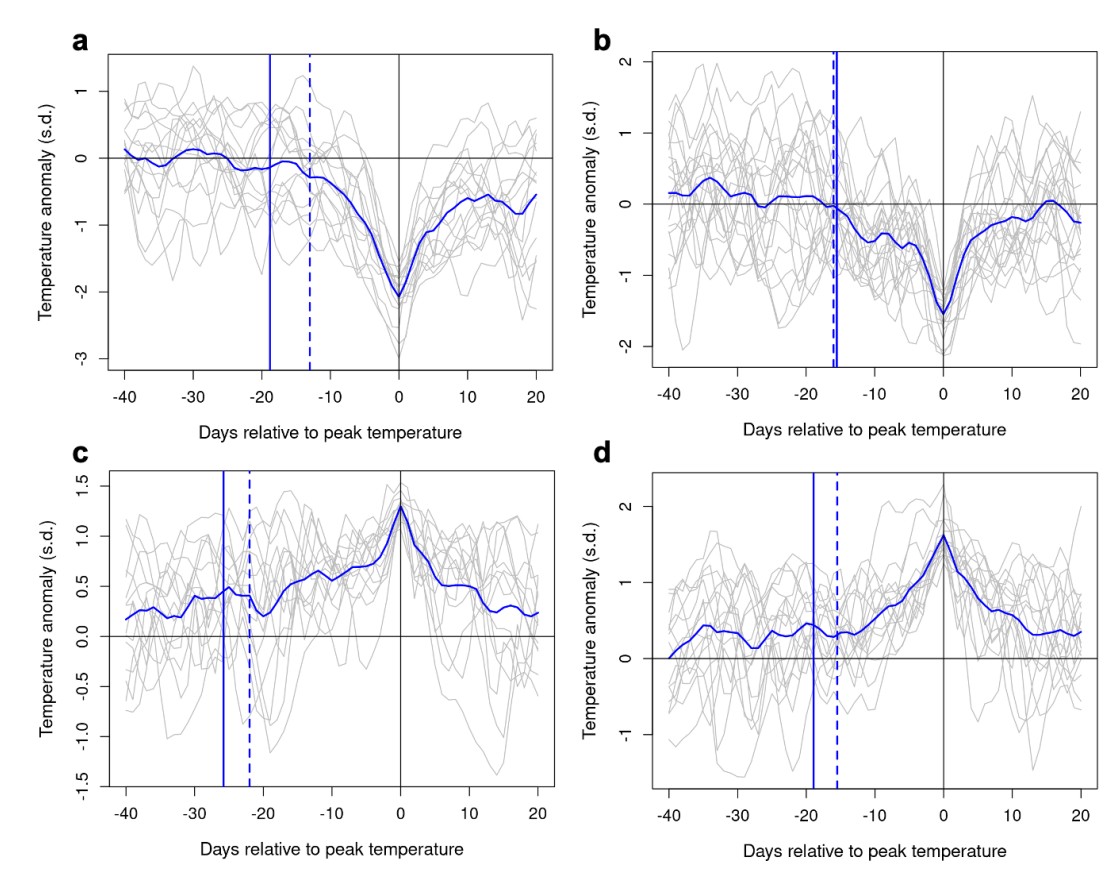

**Figure A2.** Time series of region-average daily normalised temperature anomalies (unitless) around 3-week warm and cold spells, centred on the day of peak temperature anomaly, for (a) DJF cold spells in region 19 (Northeastern Europe), (b) JJA cold spells in region 6 (Western Russia), (c) DJF warm spells in region 16 (Scandinavia) and (d) JJA warm spells in region 15 (Southwestern Europe). Gray lines show the temperature evolution for individual events, and the thick blue lines show the mean. Solid (resp. dashed) blue vertical lines indicate the mean (resp. median) start time of the analysis windows. See Figure 3 for the correspondence between region numbers and their locations.

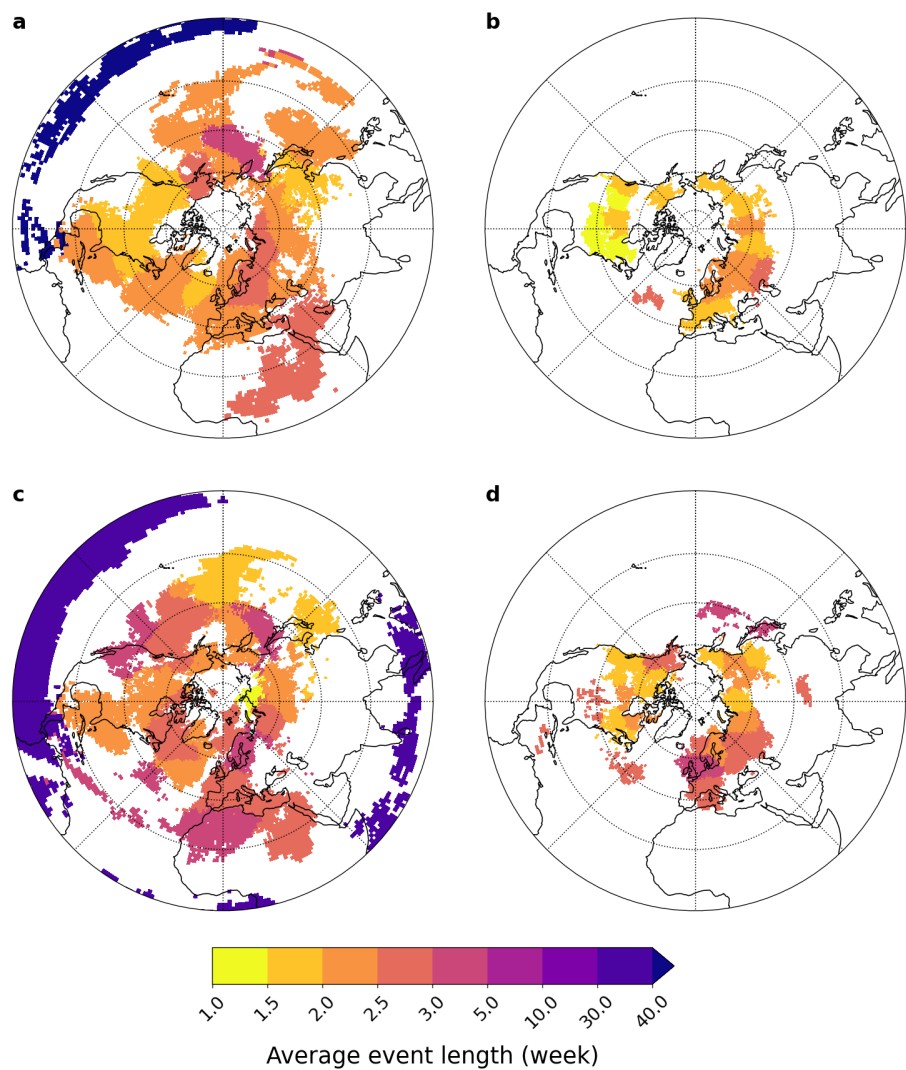

**Figure A3.** Median window length between the last positive (respectively negative) temperature anomaly before 3-week cold (respectively warm) spells and the minimum (respectively maximum) temperature anomaly in daily region-average temperature series, for (a) DJF cold spells, (b) JJA cold spells, (c) DJF warm spells and (d) JJA warm spells.

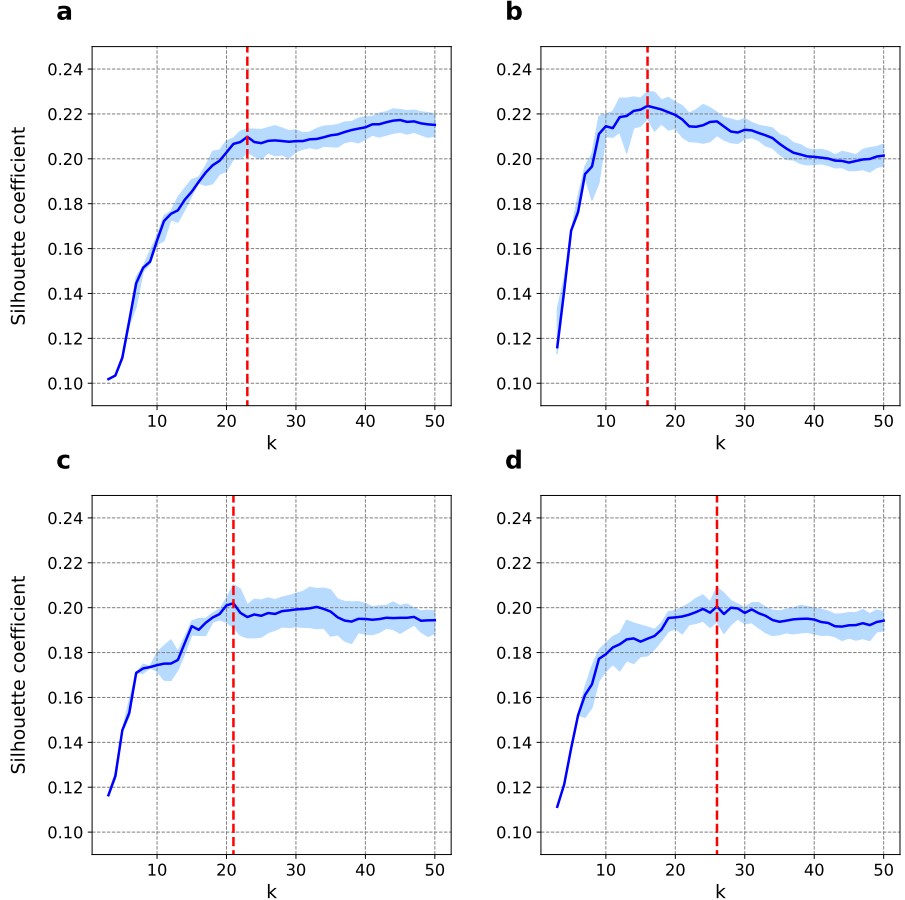

**Figure A4.** Silhouette coefficient as a function of region number $k$ in an ensemble of 100 PAM regionalisation results, for (a) DJF cold spells, (b) JJA cold spells, (c) DJF warm spells and (d) JJA warm spells. Shown are the ensemble mean (solid blue line) and range (blue shading), and selected number of regions (red vertical lines): (a) $k = 23$, (b) $k = 16$, (c) $k = 21$ and (d) $k = 26$.

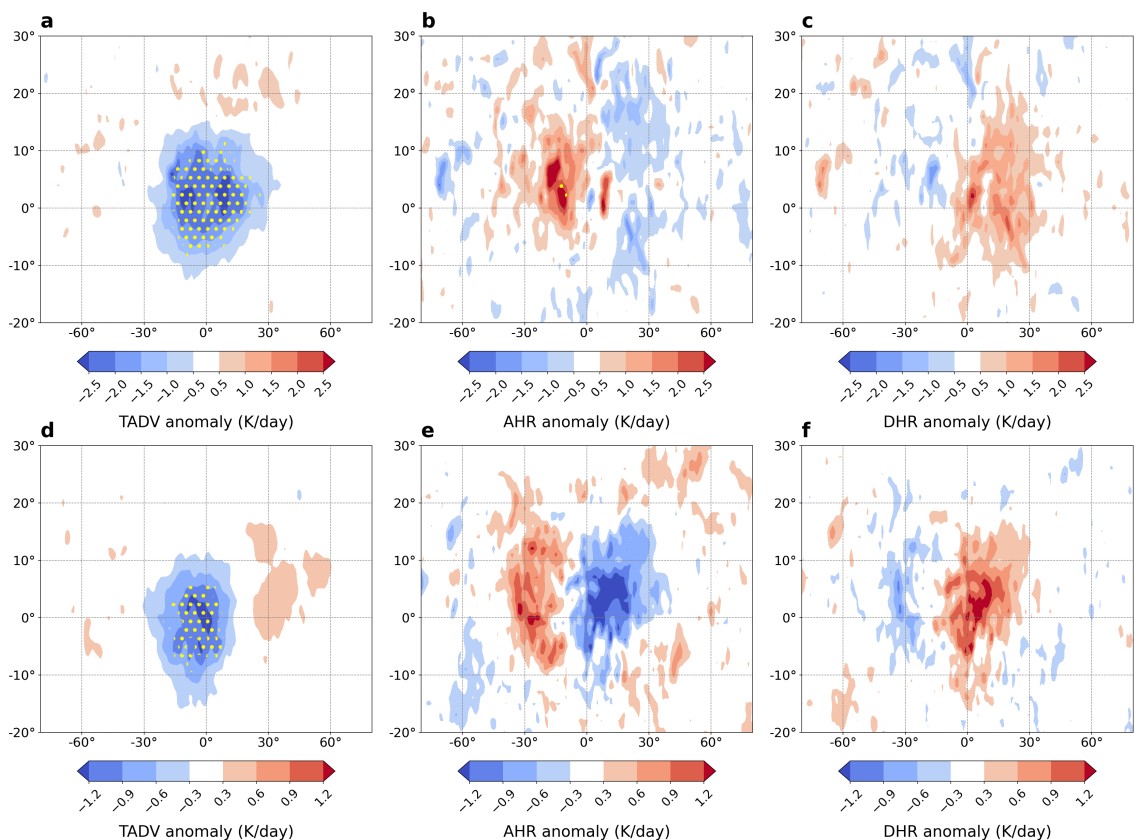

**Figure A5.** Average temperature budget term anomalies at 850 hPa for all (a-c) DJF cold spell regions (outside the tropics; see Figure 4) and (d-f) JJA cold spell regions: (a,d) horizontal temperature advection, (b,e) adiabatic heating rate and (c,f) diabatic heating rate. Yellow hatching in all panels indicates areas where anomalies are of the same sign and statistically significant for more than two-thirds of the regions.

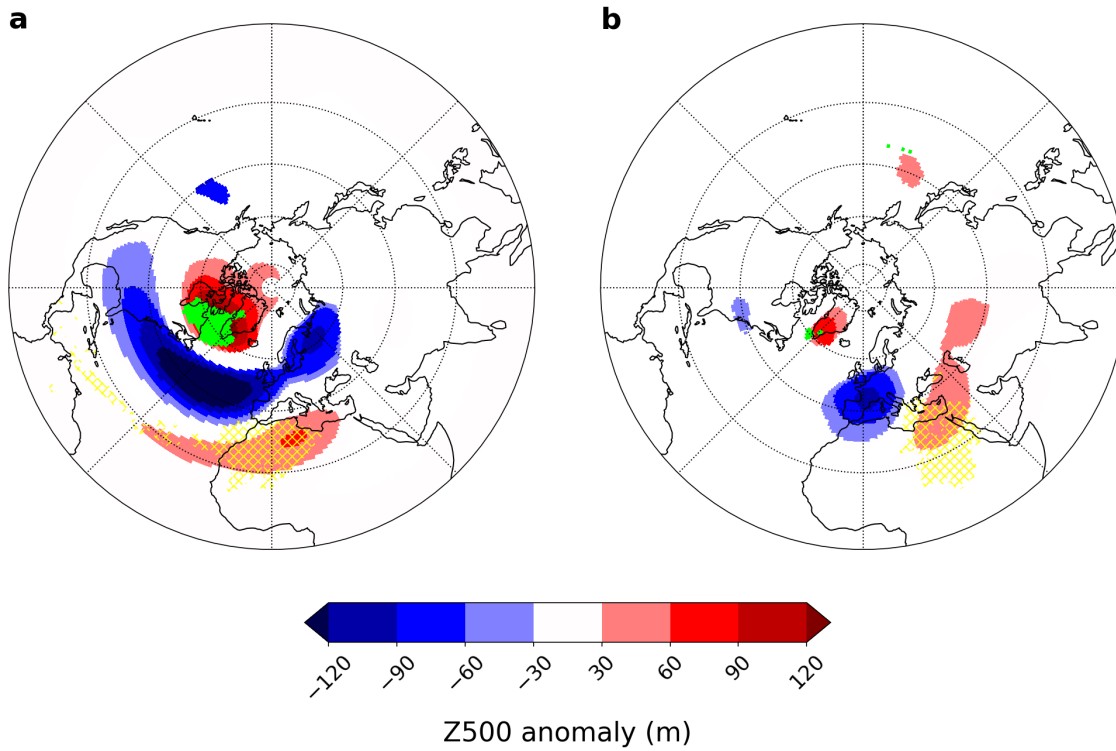

**Figure A6.** 500 hPa geopotential height anomalies (blue-red shading) and mean blocking frequency (green shading) during persistent DJF warm spells in regions (a) 4 and (b) 10 (yellow hatching; Figure 3-c). Only anomalies significant at the 10% level are plotted.

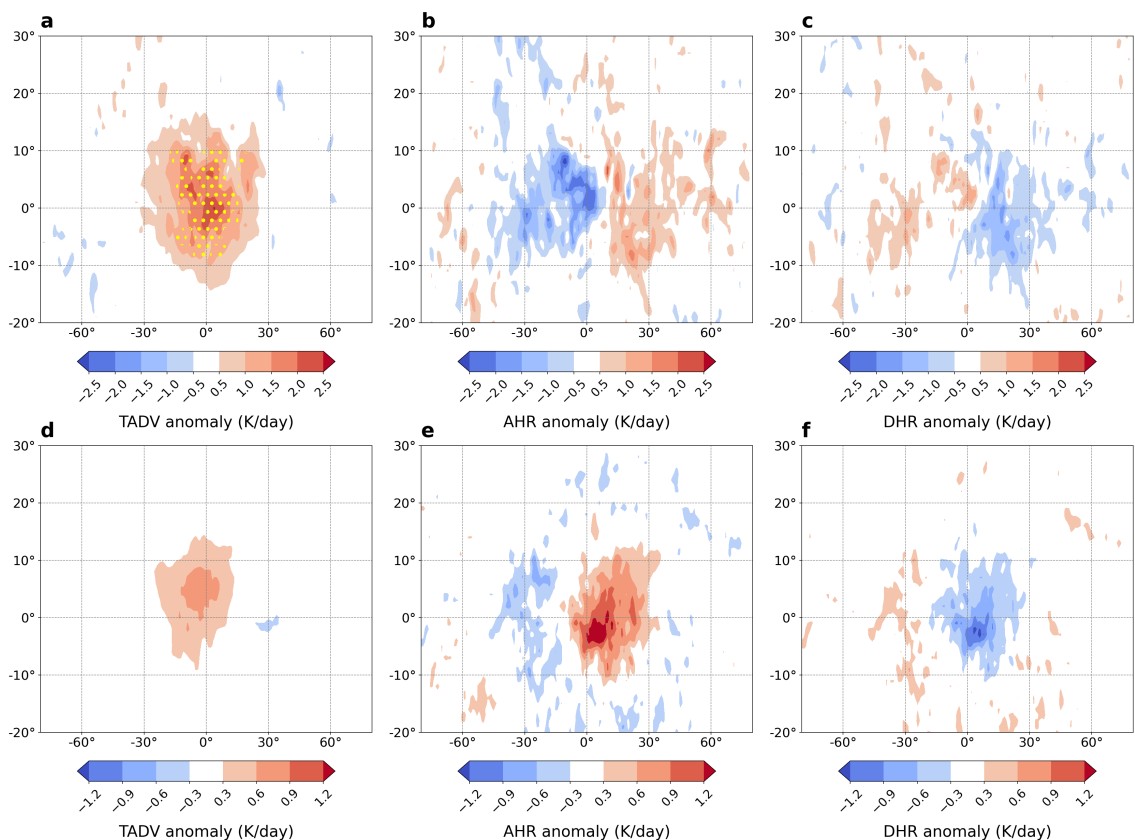

**Figure A7.** Average temperature budget term anomalies at 850 hPa for all (a-c) DJF warm spell regions (except region 1; see Figure 3-c) and (d-f) JJA warm spell regions: (a,d) horizontal temperature advection, (b,e) adiabatic heating rate and (c,f) diabatic heating rate. Green hatching in all panels indicates areas where anomalies are of the same sign and statistically significant for more than two-thirds of the regions.

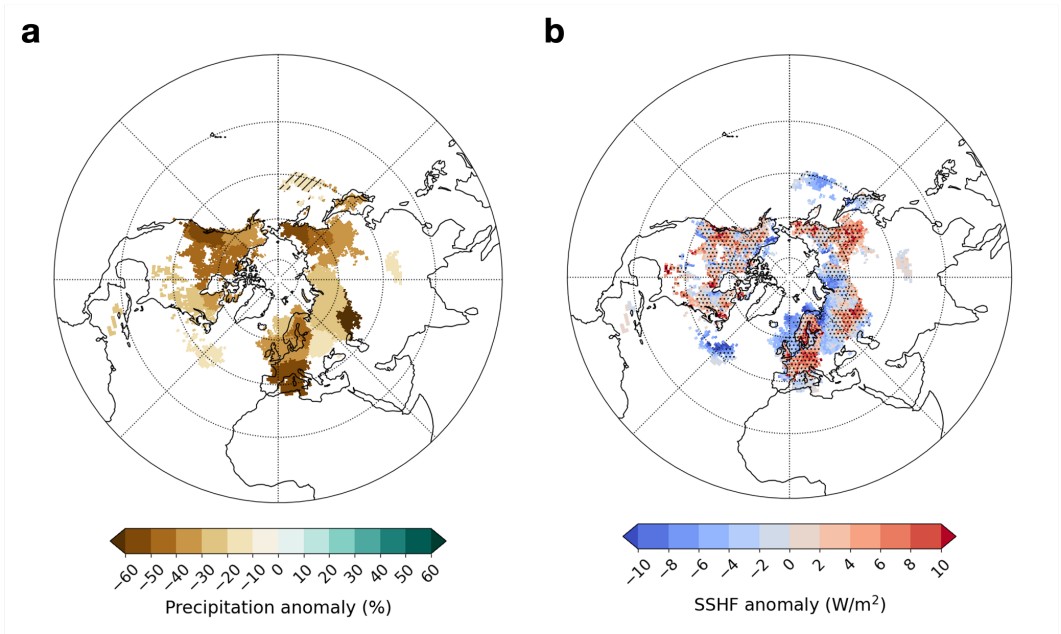

**Figure A8.** (a) Precipitation deficits (in %) and (b) surface sensible heat flux anomalies during persistent JJA warm spells (data from ERA5). Hatching in (a) indicates the lack of statistical significance of the region-average anomaly at the 95% confidence level. Stippling in (b) indicates statistical significance of the region-average anomaly at the 95% confidence level. In (b), anomalies are calculated for each $1 \times 1°$ grid cell during persistent warm spells of the corresponding region to highlight the strong land-ocean contrasts.

*Author contributions.* A.T. and O.M. conceived the study. A.T. performed all analyses and wrote the first draft with input from O.M. A.T. and O.M. interpreted the results.

*Competing interests.* The authors declare that they have no competing interests.

*Acknowledgements.* O.M. acknowledges support from the Swiss Science Foundation (SNSF) grant number 178751. The authors gratefully acknowledge the help of S. Mubashshir Ali, who calculated the PV data; of Marco Rohrer, for providing the cyclone frequency indices; and of Daniel Steinfeld, for providing the blocking frequency indices. They also thank Heini Wernli and Matthias Röthlisberger for helpful discussions.

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
