# Peer review of "Persistent warm and cold spells in the Northern Hemisphere extratropics: regionalisation, synoptic-scale dynamics, and temperature budget."

_EGUsphere, 2022_

## Author Comment (AC1)

**Response to reviewer comments**
* * *
**Comment 1.1** *I find the premises and motivation for the paper unconvincing. The authors argue that one of the most interesting aspects of temperature extremes is their persistence, as this is closely linked to their impacts and thus needs to be studied more (ll. 13-27). They further identify two key gaps in the literature: (i) the focus has been on short-lived temperature extremes but it is necessary to look at temperature extremes over longer periods (ll. 28 and following); and (ii) in the past, work has been conducted mostly on arbitrary regions (ll. 42 and following). I understand these to be the main limitations this study seeks to address. Concerning point (i), what I find to be missing is any evidence that looking at 3-week temperature extremes as opposed to e.g. 5-day temperature extremes actually provides a better picture of the impacts (or even of the meteorological drivers) of the temperature extremes. The 5-day or similar threshold generally captures past high-impact events and allows to identify coherent sets of meteorological drivers, and I assume that these are amongst the reasons why it has enjoyed such popularity in the literature. Can the authors actually identify a set of high-impact events that is overlooked by a 5-day minimum persistence criterion but captured by their own definition? Similarly, can they make a case for the fact that using a 5-day minimum persistence criterion confounds the meteorological drivers of longer-lasting events? After all, the main conclusions on the dynamical drivers of the temperature extremes that the authors find in many cases seem to support previous findings in the literature. A further point that I detail further in one of my other comments is that a lower 5-day threshold on heatwaves does not prevent including much longer-lived events in the analysis. Additionally, in many cases, impacts are related to duration of temperature extremes in a non-intuitive fashion (e.g. Xu et al., 2016), which again seems to go against the argument of the authors for looking at 3-week periods.*

**Answer**: Thank you for this important input. We realise that the motivation needs a clarification and we will rework our introduction and motivation for looking at persistent temperature extremes. We choose to define warm and cold spells to look at longer time windows than is classically done in the literature for the following reasons:

1. From an observational perspective, high or low temperature conditions sometimes persist for weeks (e.g., the October 2022 and the mid-December 2022 to mid-January 2023 warm spells in Western Europe).

2. Short windows windows tend to focus on the period of most extreme temperature within warm and cold spells. Admittedly, a 3-week warm spell would likely be detected with a shorter (say, 5-day) window, but we want to look at such events in their entirety, specifically their build-up and persistence over periods of potentially several weeks. This allows us to highlight some mechanisms that are maybe less obvious for short events, like recurrent Rossby waves.

3. While most 3-week extremes do include short periods of very extreme temperatures, only about half of 5-day extreme events occur within 3-week warm/cold spells (see Figure R1). The two approaches are thus not exactly interchangeable and with a longer window we are more confident that we capture really persistent events.

4. 5-day windows make the regionalisation more challenging. There is mechanically less synchronicity between extreme events in different locations over 5-day periods than over

3-week periods. This leads to a much more complex regionalisation (many more regions) while we want to reduce the dimension of the problem to provide a simple, physically-meaningful regionalisation.

We should also note that the fact that our results generally agree with the existing literature isn't a sufficient reason to discard them. We do not argue that we discovered new mechanisms (on that point we should certainly reformulate certain passages of the manuscript that suggest the contrary), but that we provide a comprehensive, hemispheric-wide perspective on the distribution and drivers of persistent temperature extremes. The regionalisation is an important output of our study (something which has previously not been analysed in this extent in the literature, to our knowledge).

Finally, there is evidence that choosing a longer time window to define events does provide a better representation of impacts. It is true that when it comes to human health, most studies focused on short periods (roughly 3-7 days, cf. Xu et al., 2016). But the impact of warm and cold spells on the energy sector and vegetation, for instance, clearly scales with their duration in a non-linear way (see e.g., Añel et al. 2017; doi:10.3390/atmos8110209, or the many studies looking at the impacts of long summer heatwaves on vegetation). There is hence interest in S2S prediction of warm and cold spells on these time-scales (see e.g., Van Straten et al. 2022 https://doi.org/10.1175/MWR-D-21-0201.1 who focus on four week warm periods). Regarding meteorological drivers, it is less clear that selecting a 3-week time scale is necessarily more relevant than selecting a 5-day time scale – after all, our results are in agreement with the literature (the opposite would be surprising). Still, the longer time window provides for more robustness in the regionalisation (compared to a 5-day window, see point 4 above) and thus helps associate more robust drivers to specific regions.

**Comment 1.2** *Concerning point (ii), there are several papers that have proposed regional partitions of temperature extremes based on somewhat objective meteorological criteria, some of which are cited by the authors later in the paper (e.g. Stefanon et al., 2012 and others in Sect. 6.2). Moreover, there are several studies that have chosen specific regions motivated by non-meteorological but perfectly sensible criteria, such as taking an impacts perspective (e.g. Lowe et al., 2015), maximizing data availability (e.g. Hirschi et al., 2011) or favouring ease of comparison with previous work (notably SREX regions, e.g. Perkins-Kirkpatrick and Lewis, 2020). The authors do mention that arbitrary regions may make sense from an impact perspective; I would argue that defining the regions from an impact perspective makes them distinctly non-arbitrary. I do see a value in defining regions on a hemispheric rather than continental level, as the authors do here, but I find the statement on l. 45 to be a gross misrepresentation of the literature.*

**Answer**: We agree that "arbitrary" is the wrong word here – we used it from a physical driver perspective (i.e., regions chosen based on impacts instead of based on the coherence of physical drivers). In a revised manuscript, we should reformulate by saying that previous studies have mainly looked at regions based on impacts or observed extremes, while we are interested in a purely meteorologically-driven regionalisation. We will make sure to include additional papers when discussing our motivation such as the ones you suggest.

**Comment 1.3** *In Sect. 2.2.2 the authors define their analysis window. I understand the logic*

[Figure]

Figure R1: Fraction of 5-day warm and cold spells occurring within 21-day warm or cold spells, for (a) DJF cold, (b) JJA cold, (c) DJF warm and (d) JJA warm spells. In each case spells are defined relative to the 95th or 5th percentiles of the corresponding 5- or 21-day temperature distribution.

*of wanting to study the onset phase until the peak of the warm/cold spells, but defining the first day as the one where the regional temperature anomaly changes sign may introduce an unrealistically long build-up phase for some events. For example, I am not sure that I buy the argument of a wintertime warm spell taking on average more than 2.5 weeks to build up, as suggested by table 1.*
*I also think this makes the reasoning on ll. 91-92 somewhat circular. The authors define a*

*very generous window to analyse the build-up of temperature anomalies, and then argue for the importance of analyzing temperature deviations over a long time period based on their own definition of the build-up period.*

*I would be more convinced if some analysis could show the typical evolution of regional temperatures from the first day of the analysis window and make the case that this reflects a consistent, continuous build-up as opposed to e.g. a period of oscillations around weak anomalies of the same sign and then a rapid increase of absolute anomalies closer to the peak date. Presumably, both from a physical and a more impacts-based perspective, the interest lies in the build-up and persistence of somewhat large temperature anomalies, not in small anomalies, even if persistently of the same sign.*

**Answer**: The boundaries of warm and cold spells could be defined in several ways. Here, we followed Röthlisberger and Papritz (https://doi.org/10.1038/s41561-023-01126-1) in defining the beginning of a spell as the last time before the spell when the daily temperature anomaly remained of the same sign (positive for warm spells and negative for cold spells). While for single events, this may not always be the most relevant choice, it certainly makes sense from a statistical perspective: on average, we want to capture the period during which temperatures consistently depart from their climatological average. Again, there are variations across events, but the goal is to capture their average behaviour. We show on Figure R2 the example of four different regions in Western Eurasia, which illustrate the fact that temperature anomalies, on average, take several weeks to build up, persistently deviating from their climatological mean. Note that we do not argue that the 21-day window is necessarily relevant for all regions or all events. We only take a simplified perspective in order to draw robust inferences across regions and events.

One important reason why we define spell boundaries the way we do is that we are looking at S2S timescales, and we are specifically interested in the drivers behind the onset and build-up of the extreme events, not just their extreme part. This is why it makes sense to go back to the last time when the daily temperature anomaly changed sign.

As to whether the reasoning is circular, our initial formulation was clearly misleading. We do not argue that finding analysis windows of about 3 weeks makes the case for looking at 3-week temperature anomalies (since the analysis window is constrained by the length of the event we consider). Instead, the analysis window lengths we find are consistent with mechanisms developing over S2S timescales, rather than very short-term drivers that would suddenly swing temperatures towards extreme anomalies. In that sense, the reasoning is not circular, and we will make sure to clarify it in a revised version.
* * *
**Comment 1.4** *A separate issue I have with the methodology is that the authors weigh differently each event, and could be giving a disproportionately large weight to events with a long build-up time, regardless of whether these are particularly extreme events or not. What is the range of the analysis windows for individual events and is there a correlation between event severity and duration of the analysis windows within the single regions?*

**Answer**: We are not sure what you mean by "weigh differently each event". Do you mean to say that longer events would get more weight in the anomaly calculations? There is certainly a large variability in event duration for a single region. The coefficient of variation (standard deviation divided by mean) of event durations is around 0.5 on average. Longer events, however, do not get more weight in the anomaly calculations since anomaly maps are calculated for each event separately before being averaged (this will be made clear in a revised version). And again,

[Figure]

Figure R2: Time series of region-average daily normalised temperature anomalies (unitless) around 3-week warm and cold spells, centred on the day of peak temperature anomaly, for (a) DJF cold spells in region 19 (Northeastern Europe), (b) JJA cold spells in region 6 (Western Russia), (c) DJF warm spells in region 16 (Scandinavia) and (d) JJA warm spells in region 15 (Southwestern Europe). Gray lines represent individual events, and the thick blue lines are the mean. See manuscript Figure 3 for the correspondence between region numbers and their locations.

the point is not to look at specific events, but to highlight the robust signals across events. Finally, we do not find a systematic link between the length of the analysis window and the event severity (measured by either the peak event anomaly or the 3-week event average temperature anomaly).
* * *
**Comment 1.5** *Further, I am confused by the description of the data preprocessing. On l. 65 the authors mention normalization of daily temperature using the mean and standard deviation. I assume that the former part entails subtracting the mean. On ll. 137 and following they mention subtracting the long-term average. Is the procedure such that the authors first normalize the data, including subtracting a mean value, then average it regionally and then subtract again a mean value?*

**Answer**: Thank your for pointing this out we will clarify this point in a revised version. We do normalise daily temperature series (l. 65) by removing the mean and dividing by the standard deviation, both mean and standard deviation being calculated on 30-day, 7-year moving windows

(to remove both seasonality effects and long-term trends). However, this normalisation is only used for the modeling part. In the event anomaly maps (l. 137), we proceed differently: we calculate event anomalies by removing, for each event, the long-term average calculated for the same period of the year as the observed event. For instance, if we observe a warm spell during the period of September 1-15, 2000, the anomaly maps in a given variable for this events are calculated by removing the long-term average of the variable for all September 1-15 periods (1979-2020). You might argue that for temperature or Z500 this does not take long-term trends into account. That's certainly true, however we did not find very significant differences if we removed trends from temperature and Z500 series beforehand when calculating event anomalies. We will make sure to make this more explicit in the revised version (also related to your next comment).
* * *
**Comment 1.6** *Finally, as far as I can tell the way statistical significance is computed is never explained in detail. For example, in Fig. 5 and following how is the 90% confidence level computed, and do the authors account for multiple testing?*

**Answer**: Thank you for pointing this out. We mention at l. 140 that we assess statistical significance based on a bootstrap analysis, but we certainly need to give more details about the procedure. For each region, season and type of spell, we randomly generate 1000 sets of events with the same distribution, in terms of duration and timing (i.e., starting day of the year) as observed events. We then calculate anomaly fields for these random sets of events, and determine from these empirical p-values for the anomaly fields associated with the actual events. All the significance maps include a correction for multiple testing (we apply the false discovery rate correction of Wilks (2016).
* * *
**Comment 1.7** *While some of the regions defined in the paper are relatively intuitive, others look puzzling to say the least. For example, is it really the case that cold spells in the middle-east are part of a coherent region with cold spells in the south-western Sahel (region 2 (or 3, I can't really tell the colours apart) in Fig. 2a)? Similarly, Fig. 2d seems to suggest that heatwaves in Eastern Europe/Western Russia actually belong to three (or even four) separate clusters – something that I do not recall ever having seen in the literature. I am in general not against introducing new definitions to the literature; indeed, it is part of scientific progress. However, when new definitions are presented – in this case of temperature extreme regions – which appear at odds with the "conventional" ones from the previous literature, some more robust justification and contextualization would appear necessary. I find Sect. 6.2 somewhat dismissive, by providing references to three previous regional definitions of temperature anomalies but not going any further in this direction.*

**Answer**: Our regionalisation results are of course dependent on the number of regions we select, and the results should be interpreted as showing regions where warm or cold spells are related to similar large-scale circulation patterns. Our analysis focuses on the extratropics, not the tropics, where regionalisation results (region 2 in Figure 3a which you refer to in your comment) should be taken with a grain of salt.
Our results are instead meant to highlight spatial connectivity in a systematic way (specific events may behave differently) to best analyse the physical drivers. In practice, we find that neighbouring regions generally experience warm/cold spells at different times. There is certainly some overlap (regions 3 and 8 in Figure 4-d, for instance, share about 40% of their warm spells,

[Figure]

Figure R3: Average probability of exceeding the 95th percentile of 3-week mean temperature anomalies during JJA warm spells in 5 different regions (shown by the blue contours).

which is the highest percentage between any pair of regions in either season), but that is to be expected (see Figure R3 for the example of Europe during summer). In most cases, events in neighbouring regions overlap by less than 20%. The point of the regionalisation is to group together similar locations, but as with any clustering methods, categorising points at the edges and determining the optimal number of clusters are difficult things to do.

Second, the point of our analysis is to provide a different, data-driven and physically interpretable perspective on warm and cold spells, and not to stick to "conventional" regions (though our results are consistent with other regional-scale analyses). The way that we identify regions is also rather different from what is usually done, as we explain in the introduction. However, we should spend more time discussing the consistency of our results with previous analyses. We need to search for more studies, and would be grateful if you have any specific ones in mind to point out to us.
* * *
***Comment 1.8*** *A separate comment on the regionalization results is the conflation of land and sea regions. I assume that the composites shown in Figs. 5 and following are centred on the centroid of each region. The composites then show an odd combination of averages over land and sea regions. Variables such as cyclone frequency may be largely governed by specific regions spanning the storm tracks, and much less relevant for others. I suspect that more coherent results may be obtained if, especially for surface and near-surface variables, the authors tried to composite separately land and sea points.*

**Answer**: You are right that in Figures 5, 6, 7 and 9, we do mix land and ocean points. However, (i) the whole point of the analysis in this paper is to highlight similarities and differences in warm/cold spell dynamics across space. We purposefully do not discuss the behaviour of specific regions. This would require an additional paper. While the differences you mention (e.g., relative location to storm tracks) are certainly important locally, this is not what we focus on here. (ii) Compositing land and ocean points separately is not practicable (because the location of e.g.

ocean grid points relative to the rest of the region is not consistent across regions). (iii) While the magnitude of anomalies in e.g. cyclone frequency may be very different between ocean and land regions, our analysis would nevertheless highlight areas where these are of the same sign (which is what we are interested in, after all. (iv) For cold spells the difference between land and ocean is not especially relevant, because cold spells are advection-driven. Differences are then instead to be found in the magnitude of the anomalies (temperature, advection, etc.) but not in their pattern. Regarding warm spells in winter, this could admittedly be a limitation, which we should discuss in a revised version.
* * *
**Comment 1.9** *Sect. 5 is an odd section, which seems to be midway between a results and a discussion section. Much of the section is speculative and imprecise, and generally seems to make strong statements without a clear support from the analysis presented in the study. An example is the subsection on Recurrent Rossby Wave Packets. The authors dedicate a subsection to this without ever explaining clearly whether they define in some objective way RRWPs. It is unclear to me how the reference they make to Fig. 12 would support their claims. The subsection is largely based on speculation and analogies with previous results in the literature, rather than an objective analysis. Similarly, to support the statement made on ll. 349-350, one would need to at least show precipitation anomalies in Fig. A6.*

**Answer**: In section 5, we do not claim to quantify the role of all the spell drivers we identified. This may be why you find the section to be "imprecise". A complete and quantitative analysis would go beyond the scope of our paper (which centers on the regionalisation and the common features between warm and cold spells across regions), and will be the subject of future research. That being said, we should certainly expand some of the points in this section. As noted by the other reviewer, we clearly need to clarify the definition of recurrent Rossby wave packets that we use, to better support the results shown on Figure 12. We meant to include the precipitation deficit figure, but forgot to put it next to the sensible heat flux anomaly map (Fig. A6). This is why a figure S5 was incorrectly referenced at l.348 of the manuscript. We include it here in this response (see Figure R4).
* * *
**Comment 1.10** *Finally, in the author's intention this section should discuss persistence of temperature extremes, but no definition or quantification of persistence is ever presented. The only place where the authors do define a timescale is in choosing the averaging period for the temperature anomalies and computing the "analysis window length". However, neither the persistence of the warm/cold spells (which may or may not be similar to the analysis window length if one is concerned here with the persistence of somewhat large anomalies) nor the persistence of the atmospheric circulation features associated with these is ever explicitly addressed. I struggle to see how one may make statements on the circulation features explaining the persistence of a given surface feature when persistence is never defined, nor is the duration of either feature quantified.*

**Answer**: We agree that we need to provide more details on our definition of persistence. Here, we take a fixed timescale to define "persistent" warm and cold spells. The choice of timescale, as we explained above, is motivated (i) by impacts considerations and (ii) by observed warm and cold spells during which high or low temperatures indeed persisted for several weeks in a row. What we could show however to make a better case about the perspectice we take, and make it clearer to readers, is some metric of daily temperature persistence during these warm and cold

[Figure]

Figure R4: Region-averaged precipitation anomalies (in %) during JJA warm spells. Hatching indicates the absence of significance at 5%.

events. For instance the number of days above some high (1 standard deviation) or extreme (90th percentile) threshold to show that the large temperature anomalies are indeed "persistent". By the way, we just published as a preprint a review manuscript on the various definitions of persistence in the literature, which may be of interest (see https://doi.org/10.5194/egusphere-2023-111).
* * *
***Comment 1.11*** *I find the contextualisation of the findings lacking. While in Sect. 6.2 the authors attempt to make a link to previous work which has looked at coherent regions of extreme surface temperature occurrences, what is completely missing is a discussion section making a link to the wealth of literature looking at drivers of different regional cold or warm spells. Admittedly, there is some of this in Sect. 5. However, discussing in more detail the literature related to the mechanisms is a key point needed to contextualise the current analysis relative to the literature, and I would have expected the bulk of the discussion section to be dedicated to this.*

**Answer**: We are sorry if you find the discussion of previous results insufficient. We did make an extensive literature search and referenced many studies that have looked at (persistent) warm and cold extremes, and in fact cited dozens of papers on this topic. It is true that we could expand section 5 a bit more (it is already 2 full pages long, so the discussion of previous studies is arguably not "completely missing"). However, as we argued above, the details of the regionalisation itself are not the main point of this manuscript – we do not discuss individual regions in detail, focusing instead on the similarities in terms of synoptic-scale anomalies and

drivers across regions and seasons. We will expand section 6.2 as well to provide more context for our regionalisation results in light of previously identified dynamical drivers.
* * *
**Comment 1.12** *ll. 14-15 Perhaps the authors could explain what spatially and temporally compounding events would be in the context of temperature extremes. Presumably concurrent, geographically remote heatwaves or cold spells or successions of heat waves or cold spells at the same location within a given season? Moreover, compound events are never mentioned or discussed in the analysis presented in the paper, so it seems largely irrelevant to discuss them in the introduction.*

**Answer**: This sentence is indeed irrelevant to the paper and should be removed in a revised version.
* * *
**Comment 1.13** *ll. 36-37 I find this statement misleading. First of all, there are indeed papers that have treated temperature extremes on multi-week to monthly timescales (I can think of Galfi and Lucarini (2021) off the top of my head, but a careful literature search would likely bring up others). Second, while many papers do impose a 5-day or similar threshold for heatwave duration, most heatwaves end up being much more persistent than that. It is thus misleading to state that these papers "focused on short-lived events, on the order of a few days only". Indeed, using a few days in duration as a lower threshold can easily lead to identifying multi-week heatwaves, see e.g. Vogel et al. (2020) or Fig. 10 in Grotjahn et al. (2016). This also links to my previous major comment on the motivation for looking at 3-week temperature deviations.*

**Answer**: It is true that a few papers have looked at multi-week warm or cold spells explicitly. You are also right that multi-week events can also often be detected by looking at short windows (like 5 days) (while the converse it not true, cf. our reply to your first comment). We shall reformulate accordingly.
* * *
**Comment 1.14** *ll. 63 Since the authors themselves later in the paper mention the challenges of working with 42 years of data, an obvious question is why they have chosen not to take advantage of the ERA5 back-extension, which has been available in preliminary form for over two years and in its final form since mid-2022. The major issues with the preliminary version were tropical cyclones, which is not something that would have affected the analysis presented here.*

**Answer**: It is true that the ERA5 back extension represents a wealth of new data that upcoming studies should make the most of. However, when the final version came out, we were already very advanced in our analysis, and we decided to stick to the 42-year data.
* * *
**Comment 1.15** *ll. 108 In Fig. 1 many of the patches with DR>0.4 are fragmented. How does the PAM behave when given data with scattered "holes" in it? Does this affect the robustness of the final clustering?*

**Answer**: The clustering algorithm does not take any geographical information as input, so whether the DR map is fragmented or not does not matter. It is true that some regions end up including some fragments (i.e., a few lone points scattered away from the region's main group of points) but these represent at most a few percent of a region's points and do not impact the

|      | Cold        | Warm       |
| ---- | ----------- | ---------- |
| **DJF** | 10-16 (12) | 8-23 (12) |
| **JJA** | 5-13 (9)   | 8-24 (12) |

Table 1: Analysis window length (in days) for 2-week warm and cold spells: inter-region range and median.

|      | Cold         | Warm        |
| ---- | ------------ | ----------- |
| **DJF** | 12-20 (16.5) | 11-25 (18) |
| **JJA** | 8-17 (13)    | 12-27 (17) |

Table 2: Analysis window length (in days) for 4-week warm and cold spells: inter-region range and median.

results. Such fragments are unavoidable in our regionalisation since by choosing a rather small number of regions forces the algorithm to add isolated points to one of the regions. We explain at ll. 117-121 how we remove these isolated points to better visualise the regions.
* * *
**Comment 1.16** *ll. 190-191 It would be interesting for the readers to see these numbers for 2 and 4 week averaging times, e.g as an Appendix table.*

**Answer**: Here are the numbers for 2 and 4 week events:
* * *
**Comment 1.17** *Sect. 4.1 How do the authors determine this grouping? Is it through some objective criterion or through a subjective analysis of the individual regions?*

**Answer**: We obtained this grouping by subjective analysis based on the anomaly maps, and we should make it explicit in the revision. Note however that a simple clustering applied to the Z500 anomaly maps yields a similar partition.
* * *
**Comment 1.18** *A few typos (although I appreciated how well-written the paper generally was):*

*l. 211 near-surface diabatic*
*ll. 272, 328 Incorrect figure reference?*
*l. 328 (difference in intensity/spatial extent?)*
*ll. 347-348 (Figures 11) and S5)*

**Answer**: Thanks, we will correct these typos in the revised manuscript.
* * *
**Comment 1.19** *ll. 296-297 I am not sure I agree with this statement, and indeed the following two sentences seem to counter it. I would suggest adding a "summertime" to the first sentence to avoid confusion.*

**Answer**: You are right (zonal configurations are important in winter) and we will modify the sentence accordingly.
* * *
**Comment 1.20** *l. 310 and following. The current formulation seems to suggest that there is plenty of work on the role of upstream blocking but that the authors are the first to highlight the*

[Figure]

Figure R5: Modified Figure 5 (see manuscript).

*role of downstream blocking/processes for the occurrence of upstream temperature extremes. While upstream blocking has certainly received more attention, there are studies which also consider the role of downstream features (e.g. Takaya and Nakamura, 2005; or Lehman and Coumou, 2015, who discuss the role of downstream storm track anomalies in the context of eastern North American extremes).*

**Answer**: You are correct to point out that the initial formulation was misleading, and we do not claim to be the first to highlight the role of downstream blocking. Our point is that our regionalisation captures the varied influences of blocking (up- or downstream) in a coherent way and provides a useful hemispheric perspective on the role of blocking. Takaya and Nakamura discuss a nice example for Eastern Asia which we will include as reference. As far as we can see, however, Lehman and Coumou only perform pixel-wise regressions and do not discuss teleconnections between surface extremes and circulation features.

**Comment 1.21** *Fig. 5 and following: using green dots on a red background is not ideal for many readers (including this reviewer).*

**Answer**: Thank you for pointing it out. It is hard to find a color that stands out well on red, blue, green and brown. We could use yellow (see Figure R5).

**Comment 1.22** *Fig. 12 If the authors only look upstream of the regions (30 W of the westernmost point as stated in the figure caption) how can they diagnose downstream blocking as again stated in the caption?*

**Answer**: We only look 30°W upstream for the R-metric anomalies, not for the blocking (the caption states that "Hatching indicates significant positive blocking frequencies up- or downstream of the region". We will clarify the caption accordingly.
* * *
**Comment 1.23** *On a separate point, it may be worth explaining in the methods what the R-metric is and how it is being used here.*

**Answer**: You are right; we realise now that we should have spent more time explaining the R-metric and the corresponding analysis. We will do so if invited to submit a revised version.

---

## Author Comment (AC2)

**Response to reviewer comments**
* * *
**Comment 1.1** *The manuscript is well-written and the figures are of high quality. However, I have significant concerns about the methods, the implication of the results, and the comparison with the existing literature. In my opinion, the study does not address the research question stated in the title.*

**Answer**: Thank you for your comment. We agree that our initial title was misleading: we do not analyse the persistence of warm and cold spells per se, but the characteristics of persistent warm and cold spells (which is different). We would modify the title accordingly if invited to submit a revision. Our replies below should address your concerns as to the methodology and comparison with existing literature.
* * *
**Comment 1.2** *In the Introduction, the authors identify two gaps in the literature on the processes responsible for warm and cold spells: (i) persistence and (ii) spatial dependence. With respect to the first gap (persistence), what can we learn meteorologically from longer lasting events that we don't already know from the existing literature? There is already a lot known on driving factors for heat and cold waves. I am not arguing that it is not interesting to look at the persistence, but the authors insufficiently motivate why it could be interesting to look at persistent events (apart from impacts). In addition, a minimum duration of three days does not exclude long-lasting events, e.g. in Zschenderlein et al. (2019), a heat wave with a duration of more than 40 days was included in the analysis. It is therefore not true that previous work generally focused on short-lived events only, as the authors claim.*

**Answer**: We should certainly rework our introduction and motivation for looking at persistent temperature extremes. We choose to define warm and cold spells to look at longer time windows than is classically done in the literature for the following reasons:

1. From an observational perspective, high or low temperature conditions sometimes persist for weeks (e.g., the October 2022 and the mid-December 2022 to mid-January 2023 warm spells in Western Europe).

2. Short windows windows tend to focus on the period of most extreme temperature within warm and cold spells. Admittedly, a 3-week warm spell would likely be detected with a shorter (say, 5-day) window, but we want to look at such events in their entirety, specifically their build-up and persistence over periods of potentially several weeks. This allows us to highlight some mechanisms that are maybe less obvious for short events, like recurrent Rossby waves.

3. While most 3-week extremes do include short periods of very extreme temperatures, only about half of 5-day extreme events occur within 3-week warm/cold spells (see Figure R1). The two approaches are thus not exactly interchangeable and at least with a longer window we are more confident that we capture really persistent events.

4. 5-day windows make the regionalisation more challenging. There is mechanically less synchronicity between extreme events in different locations over 5-day periods than over 3-week periods. This leads to a much more complex regionalisation (many more regions) while we want to reduce the dimension of the problem to provide a simple, physically-meaningful regionalisation.

Additionally, there is evidence that choosing a longer time window to define events does provide a better representation of impacts. It is true that when it comes to human health, most studies focused on short periods (roughly 3-7 days, cf. Xu et al., 2016). But the impact of warm and cold spells on the energy sector and vegetation, for instance, clearly scales with their duration in a non-linear way (see e.g., Añel et al. 2017; doi:10.3390/atmos8110209, or the many studies looking at the impacts of long summer heatwaves on vegetation).

Regarding meteorological drivers, it is less clear that selecting a 3-week time scale is necessarily more relevant than selecting a 5-day time scale – after all, our results are in agreement with the literature (the opposite would be surprising). Still, the longer time window provides for more robustness in the regionalisation (compared to a 5-day window, see point 4 above) and thus helps associate more robust drivers to specific regions.

As to what we can learn from longer-lasting extremes, we openly admit that our results are in full agreement with the previous literature. It would have been surprising to have found something completely new! (on that point we should certainly reformulate certain passages of the manuscript that may suggest the contrary) But this fact isn't a sufficient reason to say that analysing longer events is useless. We argue however that our results provide a comprehensive, hemispheric-wide perspective on the distribution and drivers of persistent temperature extremes. The regionalisation is an important output of our study (something which has previously not been analysed to this extent in the literature, to our knowledge). The long timescale is also important to obtain a robust regionalisation.
* * *
**Comment 1.3** *With respect to the second gap (spatial dependence), the authors argue that the recent literature looked at arbitrary regions and mixed up areas where warm or cold spells are shaped by different physical processes. They are by far not arbitrary in the recent literature, either regions are motivated by impacts (which is perfectly fine and definitely not arbitrary) or by different climates (e.g. humid vs. dry climate). There are also studies looking at the different physical processes in a region, e.g. Sousa et al. (2018, Fig. 5). It is therefore not overlooked that in one region the processes can differ regionally.*

**Answer**: We agree that "arbitrary" was a very wrong choice of word. We interpreted it from a physical driver perspective – i.e., regions chosen based on impacts instead of based on the coherence of physical drivers. In a revised manuscript, we should reformulate by saying that previous studies have mainly (but not exclusively) looked at regions based on impacts or observed extremes, while we are interested in a purely meteorologically-driven regionalisation (which admittedly has not been done, at least on a hemispheric scale).
* * *
**Comment 1.4** *The study promises a lot to provide new insights into the persistence and spatial dependence of warm and cold spells. However, both aspects are not really quantified in the study. I detail my comments below:*

*1. Persistence: Sections 4 and 5 are believed to address this research question. However, these sections rather analyse synoptic conditions during warm and cold spells (which is nicely done) and identify three atmospheric drivers (upstream blocking, downstream blocking, subpolar troughs). Many studies already investigated the position of the block relative to the high/low temperature extreme at the surface (e.g. Pfahl and Wernli 2012, Pfahl 2014, Bieli et al., 2015, Santos et al., 2015, Sousa et al. 2018, Zschenderlein et al., 2019). The authors should therefore make the new findings more clear. While I find the synoptic analysis very nice, I don't understand what this has to do with persistence. And this is the main aspect of the paper (at least it is stated*

[Figure]

Figure R1: Fraction of 5-day warm and cold spells occurring within 21-day warm or cold spells, for (a) DJF cold, (b) JJA cold, (c) DJF warm and (d) JJA warm spells. In each case spells are defined relative to the 95th or 5th percentiles of the corresponding 5- or 21-day temperature distribution.

*in the title). For example, how important is the position of the block relative to the warm/cold spell with respect to the persistence in comparison to a heat/cold spell of about 5 days? I think the authors try to discuss the persistence aspect in Section 5, but not really successful. They are rather speculating about the role of blocks, RRWPs, land-atmosphere feedback, and subpolar troughs (with results from the existing literature and not their own) and not quantifying the importance of these processes with respect to persistence.*

**Answer**: We realise now that the title we chose was misleading since it led readers into thinking that we would be analysing persistence of warm and cold spells in itself rather than the drivers during persistent events and their possible role in the persistence of the warm and cold spells (which is not exactly the same thing). A detailed analysis of persistence would indeed require investigating warm and cold spells across different timescales to understand what determines their length. We will therefore make sure to reformulate our title and introduction so as to avoid misleading readers as to the content of the manuscript.
* * *
**Comment 1.5** *2. Spatial dependence: I think the authors want to address this research question with the regionalisation approach (Fig. 3). But in the end, they summarise all regions into three categories (and they don't describe how they do it, see also my minor comments). What is the added value of the regionalisation? What can we learn from it? What do the different colors in Fig. 3 imply meteorologically? In my view, Figure 3 is insufficienty discussed.*

**Answer**: The regionalisation certainly helps identify distinct regions, which could be analysed separately in detail, but this is not the purpose of our manuscript (this will be the topic of future work, hopefully also by others who will make use of our regionalisation results). Instead, in this paper we wish to highlight the common features during warm or cold spells across different regions, so as to identify robust signals, while still analysing the different seasons separately and retaining some spatial variability (for instance when it comes to blocking or recurrent Rossby waves).

We understand that it may be frustrating not to see the case of specific regions be discussed in detail, and many regionalisation papers do take that path (we did just that in a previous paper on the regionalisation of temporal clustering of precipitation extremes; https://doi.org/10.1175/JCLI-D-21-0562.1). With a hemispheric perspective, such a paper would however be too long and we chose instead to highlight similarities across regions. In this sense, the regionalisation is extremely helpful to reduce the dimension of the problem: instead of conducting an analysis for each grid point separately, the regionalisation allows to first reduce the problem to a small, manageable set of regions. In addition, taking this regional approach allows us to shed light on the structure of regional-scale anomalies (e.g., eastern and western halves of regions sometimes behaving differently).

We could nevertheless discuss Figure 3 a bit more (e.g., the distribution of region area, a more in-depth comparison to previous regionalisation attempts, differences/similarities across seasons, etc.)
* * *
**Comment 1.6** *I am not convinced by the method for the identification of persistent warm and cold spells. While I understand that you incorporate the days prior to the spells to investigate the onset, I do not understand why your analysis window ends with the peak temperature anomaly. What if this anomaly is reached already very early during the warm/cold spell and high/low temperatures still persist after that peak?*

**Answer**: It is true that the way we define the analysis window is not perfect. The way we do it here emphasises the build-up part of the spell, because we wanted to avoid including the decay part (when the drivers are likely weakening). Additionally, as long as the drivers are in place, temperature anomalies are likely to increase in magnitude (towards either extreme warmth or cold), even weakly, rather than to remain stable. We don't think that going a few days beyond the peak anomaly would make much difference, but in a revised version we will mention this

point, and possibly compare results with an analysis window that extends a bit further (e.g., as long as the anomaly does not drop "too much" back towards the climatology)
* * *
**Comment 1.7** *And for the quantile regression I would exclude grid points in the tropics beforehand, because a low Z500 variability in this area is no surprise.*

**Answer**: The quantile regression is applied to each grid point separately, so there is no influence of e.g. the tropics onto the mid-latitudes. As to the regionalisation, we remove grid points with low model skill (deviance ratio below 0.4) beforehand, which includes the vast majority of the tropics.
We could have restricted our analysis to, say, polewards of $20°$ or $30°$N. Still, since some of the regions extend slightly more to the south, it is just as well to keep the whole Northern Hemisphere (and in any case, tropical grid points are largely excluded, as explained above).
* * *
**Comment 1.8** *The results of this study are not set into context with the existing literature. An attempt is made in Sections 5 and 6.2, but it is not clear what new knowledge is introduced by the study.*

**Answer**: We are sorry if you find the discussion of previous results insufficient. We did make an extensive literature search and referenced many studies that have looked at (persistent) warm and cold extremes, and in fact cited dozens of papers on this topic. It is true that we could expand section 5 a bit more. We will also expand section 6.2 to provide more context for our regionalisation results in light of previously identified dynamical drivers.
Additionally, we will make it clearer what are the new results brought by this study. A main result is the regionalisation itself. Though we do not discuss the case of individual regions separately (this will be the topic of future research), we highlight the coherence of the regionalisation and underline the many similarities and differences in terms of synoptic-scale anomalies and drivers of warm and cold spells across the Northern hemisphere extratropics. Admittedly we do not claim to have identified new, previously unknown drivers of warm and cold spells, and we also need to make clearer that we do not discuss persistence per se but rather the characteristics of persistent warm and cold spells (our initial title was misleading in that regard).
* * *
**Comment 1.9** *L5-10 (Abstract): All listed processes (blocks important precursors of warm and cold spells, location of the blocks, recurrent Rossby waves, land-atmosphere feedbacks) are already described in many papers (see list in major comment 2) or in a new review paper by Domeisen et al. (2022, "Atmospheric processes" section; the paper was probably not yet published at the time the authors submitted to WCD).*

**Answer**: Our findings are indeed consistent with the literature (the opposite would be surprising). We will make sure to include the recent reference of Domeisen et al. which was indeed not yet published when we wrote our manuscript.
* * *
**Comment 1.10** *L65: Does this mean that you first subtract the mean and divide by the standard deviation?*

**Answer**: Yes, indeed. We will make it explicit in the revision.
* * *
**Comment 1.11** *L66-68: Why do you use different methods for the trend removal of T2M and Z500?*

**Answer**: The main reason is that we did not want to normalise the Z500 data; otherwise, tropical variability would have been dominant and we would not have captured the mid-latitude signals. So we simply removed the long-term trend in the Z500 mean.
* * *
**Comment 1.12** *L70: Is the persistence criterion of 4 days valid for an individual grid point or for the whole cyclone mask?*

**Answer**: It applies to the cyclone as a system (this is a Eulerian persistence criterion).
* * *
**Comment 1.13** *L70-80: Should be moved to the methods section.*

**Answer**: These lines are indeed a mix between data and methods, but we would prefer to focus the methods section on the core methods for this paper (modeling and analysis) rather than on the data pre-processing.
* * *
**Comment 1.14** *L84: Is the percentile based on daily or 3-weekly values? And please do not use different words for the same method (normalise, rescaled)*

**Answer**: On 3-week values. The details are given just a bit later in section 2.2.1 so to avoid any confusion we will remove the reference to percentiles at l. 84.
* * *
**Comment 1.15** *L89: Why non-overlapping 3-week intervals?*

**Answer**: We want the intervals to be non-overlapping to avoid as much dependency between model input data points as possible.
* * *
**Comment 1.16** *L105: How sensitive are your results with respect to the DR threshold?*

**Answer**: We chose this specific threshold (0.4) to capture about as much of the extratropics in all seasons as possible. A lower threshold would lead to more tropical grid points retained in the regionalisation (extratropical regions remaining essentially the same in the regionalisation algorithm). A higher threshold removes points from the extratropics and quickly tends to decrease the "optimal" number of regions.
* * *
**Comment 1.17** *L112: What is a silhouette coefficient?*

**Answer**: We should define this in a revised version. The silhouette coefficient is a common metric used to identify an "optimal" cluster number. It essentially compares the mean intra- and inter-cluster distances, with higher values indicating a better clustering (high inter-cluster distance and low intra-cluster distance).
* * *
**Comment 1.18** *L129: Why not using the back-extension of ERA5?*

**Answer**: This comment was also made by the other reviewer. It is true that the ERA5 back extension represents a wealth of new data that upcoming studies should make the most of.

However, when the final version came out, we were already very advanced in our analysis, and we decided to stick to the 42-year data.
* * *
**Comment 1.19** *L135: Is the decay phase of the event not interesting?*

**Answer**: It certainly is, but not in the context of our paper which focuses on the mechanisms responsible for the onset and maintenance of the warm and cold spells. We can expand the analysis window a bit, but we should avoid including too much of the decay phase since there is no reason why the same drivers should still be in place.
* * *
**Comment 1.20** *L143: median point – do you mean centroid?*

**Answer**: No, here we take the actual (geographical) median point, and not the cluster centroid (which in PAM are called medoids, and minimise the distances to all other cluster points).
* * *
**Comment 1.21** *Figure 1: I don't understand the values on the y-axis. Is it normalised with the 95th percentile? I also do not understand why the grey shaded area goes from approximately day 11 to 31.*

**Answer**: Figure 1 illustrates an example of a warm spell by showing the corresponding daily normalised temperature series. This is why the y-axis of Figure 1 is in multiples of the standard deviation of daily temperature anomalies (cf. normalisation process in section 2.1). As to the grey shaded area, it corresponds to the identified 3-week warm spell (identification based on a rolling 3-week average of daily temperature anomalies; section 2.2.2). Based on this event (which occurs from day 11 to day 31 here), we calculate an analysis window between the last day with a negative temperature anomaly (day 6, in this case) and the day of peak temperature anomaly during the 3-week event (day 28, in this case). This figure is just an illustration of the method and the actual values of the days is meaningless (the x-axis could start at -11, for instance). We will change the caption to clarify these points.
* * *
**Comment 1.22** *Section 4.1: How have you identified the three main groups?*

**Answer**: This is something we should have specified in the manuscript. We identified the main groups visually, after removing the tropical regions.
* * *
**Comment 1.23** *Figure 5 (and others): Mark the centroid of the regions. Add wind directions or Z500 mean values.*

**Answer**: By construction, the centroid of the regions is at the origin in these figures. The mean Z500 and wind fields are not as meaningful as the anomaly fields here since we average across regions located at different latitudes/longitudes.
* * *
**Comment 1.24** *General remark to the composites: I personally think that compositing can be quite dangerous here, because you are averaging over areas with different sizes and shapes. Can you comment on that?*

**Answer**: You are right and this is certainly a limitation that should be explicited. However, the whole point of the compositing is to highlight the similarities in terms of synoptic-scale

anomalies between different regions. In some cases (e.g., blocking or recurrent Rossby waves) we were careful to analyse each region separately (to obtain Figures 4, 10 and 12). Region-specific details would have to be discussed in a different analysis.
* * *
**Comment 1.25** *L225: Be careful because you are mixing land and ocean grid points (see red colours in Fig. 4).*

**Answer**: This is true (and the point was also made by the other reviewer. There are certainly limitations to our analysis due to the mixing together of land and ocean grid points, which we should make explicit. However, for cold spells the difference between land and ocean is less of a problem, because cold spells are advection-driven. Differences are to be found in the magnitude of the anomalies (temperature, advection, etc.) but not in their pattern. Additionally, the point of our analysis is to highlight similarities and differences in warm/cold spell dynamics across space.
* * *
**Comment 1.26** *Figure 6: The scale in Fig. 6c differs from Fig. 5c!*

**Answer**: We will make sure to use the same scale, thanks.
* * *
**Comment 1.27** *L250: These results are in line with Zschenderlein and Wernli (2022).*

**Answer**: We will add the citation here, thanks.
* * *
**Comment 1.28** *Section 4.4 and Figure 12: I am not able to follow because you haven't described the R-metric sufficiently.*

**Answer**: This point was also raised by the other reviewer, and we will make sure to add details about the R-metric in a revised version. The current information in section 2.1 is rather sparse. All details are in Röthlisberger et al. (2019), but we should expand a bit for readers unfamiliar with this work. The R-metric is calculated as the magnitude of the envelope of meridionally-averaged 35-65°N 250 hPa 6-hour meridional winds, which were first averaged using a moving 14-day window, and then filtered to retain wavenumbers 4-15 only (the ones relevant for synoptic Rossby waves). A high R-metric thus indicates a strong wavenumber 4-15 component in the series, and to multiple Rossby waves because of the 14-day running mean.
* * *
**Comment 1.29** *L290: remove the second "is"*

**Answer**: We will correct that, thanks.
* * *
**Comment 1.30** *L310: This agrees with a host ... → ok, but where is the new knowledge regarding the persistence?*

**Answer**: See our answer to your Comment 1.8.
* * *
**Comment 1.31** *L312-313: This is probably also true for events on the shorter time scales.*

**Answer**: Yes, certainly.
* * *
**Comment 1.32** *L318: remove half sentence in brackets at the end of the line.*

**Answer**: Corrected, thanks.
* * *
**Comment 1.33** *L348-350: This is not clear to me. Increasing surface heat fluxes would imply diabatic heating, but we see diabatic cooling in Fig. 11f. Radiative cooling is typically dominant in the mid- to upper-troposphere.*

**Answer**: In Figure 11f, the diabatic heating term is calculated as the residual of the energy balance equation, and is not restricted to radiative cooling. Still, what probably happens is that the radiative cooling term outbalances surface sensible heat fluxes, since it has to balance not only sensible heat fluxes, but also adiabatic warming and horizontal advection. Note that our analysis also covers day and night, and that sensible heat fluxes peak during the day but are weaker at night, while radiative cooling can remain large across the clock. We would however make sure to mention this point in a revised version.
* * *
**Comment 1.34** *L351-352: What can we learn from this sentence?*

**Answer**: We mentioned this because a number of previous studies have argued that extreme warm spells were often associated with antecedent precipitation deficits. However, in our case we probably do not look at event that are extreme enough to see the connection.
* * *
**Comment 1.35** *Section 5.4: You discuss how a subpolar trough develops. But I don't see how you discuss subpolar troughs and their connection to persistence of warm and cold spells.*

**Answer**: You are right that we do not discuss the development of subpolar troughs, but rather point out that persistent wintertime cold spells at high latitudes are likely associated with such systems (the analysis of synoptic-scale anomalies points in that direction). They connect to persistent cold spells because of the associated circulation anomalies, but also because such systems have a general tendency to persistence, as we discuss in the manuscript (cf. also Woolings et al. (2023); https://doi.org/10.5194/wcd-4-61-2023).
* * *
**Comment 1.36** *Section 6.1: A matter of taste, but the discussion of Fig. 3 appears quite late in the study. I would prefer to discuss the results earlier.*

**Answer**: We agree. One possibility would be to merge current sections 5 and 6 into a larger discussion section, and to move section 6.1 to the beginning, before discussing the physical drivers.
* * *
**Comment 1.37** *L383: Acronym S2S not introduced*

**Answer**: Corrected, thanks (S2S = "subseasonal-to-seasonal")

---

## Author Response (AR1)

**Response to reviewer comments**
* * *
**Comment 1.1** *I find the premises and motivation for the paper unconvincing. The authors argue that one of the most interesting aspects of temperature extremes is their persistence, as this is closely linked to their impacts and thus needs to be studied more (ll. 13-27). They further identify two key gaps in the literature: (i) the focus has been on short-lived temperature extremes but it is necessary to look at temperature extremes over longer periods (ll. 28 and following); and (ii) in the past, work has been conducted mostly on arbitrary regions (ll. 42 and following). I understand these to be the main limitations this study seeks to address. Concerning point (i), what I find to be missing is any evidence that looking at 3-week temperature extremes as opposed to e.g. 5-day temperature extremes actually provides a better picture of the impacts (or even of the meteorological drivers) of the temperature extremes. The 5-day or similar threshold generally captures past high-impact events and allows to identify coherent sets of meteorological drivers, and I assume that these are amongst the reasons why it has enjoyed such popularity in the literature. Can the authors actually identify a set of high-impact events that is overlooked by a 5-day minimum persistence criterion but captured by their own definition? Similarly, can they make a case for the fact that using a 5-day minimum persistence criterion confounds the meteorological drivers of longer-lasting events? After all, the main conclusions on the dynamical drivers of the temperature extremes that the authors find in many cases seem to support previous findings in the literature. A further point that I detail further in one of my other comments is that a lower 5-day threshold on heatwaves does not prevent including much longer-lived events in the analysis. Additionally, in many cases, impacts are related to duration of temperature extremes in a non-intuitive fashion (e.g. Xu et al., 2016), which again seems to go against the argument of the authors for looking at 3-week periods.*

**Answer**: Thank you for this important input. In response to our earlier comment, we modified the introduction at length to better motivate our approach and improve the literature discussion. Please see our reworked introduction. In particular, we added a paragraph to justify our choice of time window: *"However, previous works have generally used short time windows to define warm or cold spells, on the order of 3-5 days (e.g., Schaller et al., 2018; Plavcová and Kyselý, 2019; Jeong et al., 2021; Jiménez-Esteve and Domeisen, 2022). Heatwaves, for instance, are frequently defined as sequences of at least 3 or 5 extreme warm days, with the longest events usually lasting about a week (e.g., Perkins and Alexander, 2013; Plavcová and Kyselý, 2019). While such windows may also identify multi-week spells, they focus on the part of the spells with the most extreme temperatures. In addition, while persistent spells frequently include short periods of extreme temperature anomalies, the opposite is not true: many short-lived, extreme temperature anomalies are not part of multi-week warm or cold spells. Finally, choosing a longer window is useful to identify regions where spells tend to co-occur, as it allows to capture the trajectory of the synoptic-scale systems responsible for the temperature anomalies across their lifetime."*
* * *
**Comment 1.2** *Concerning point (ii), there are several papers that have proposed regional partitions of temperature extremes based on somewhat objective meteorological criteria, some of which are cited by the authors later in the paper (e.g. Stefanon et al., 2012 and others in Sect. 6.2). Moreover, there are several studies that have chosen specific regions motivated by non-meteorological but perfectly sensible criteria, such as taking an impacts perspective (e.g.*

*Lowe et al., 2015), maximizing data availability (e.g. Hirschi et al., 2011) or favouring ease of comparison with previous work (notably SREX regions, e.g. Perkins-Kirkpatrick and Lewis, 2020). The authors do mention that arbitrary regions may make sense from an impact perspective; I would argue that defining the regions from an impact perspective makes them distinctly non-arbitrary. I do see a value in defining regions on a hemispheric rather than continental level, as the authors do here, but I find the statement on l. 45 to be a gross misrepresentation of the literature.*

**Answer**: "Arbitrary" was a poor choice of word here and we agree that previous region selection can make sense in a perspective different from ours. Here, we are interested in a purely meteorologically-driven regionalisation to highlight (i) the spatial co-occurrence of persistent spells and (ii) the influence of different drivers. We modified the introduction to highlight this argument: "*Still, much remains unknown about the geography of persistent warm and cold spells, their spatial dependence, and how the role of various identified drivers varies in space. Such information is essential for risk assessment and to improve forecasts (Perkins, 2015; van Straaten et al., 2022). Persistent spells indeed typically occur over large spatial scales, from hundreds to thousands of square kilometers, with potentially complex spatial footprints (Lyon et al., 2019; Vogel et al., 2020). Yet, most previous work on heat waves, for instance, has focused at the grid-point scale or over regions based on impacts, administrative boundaries, data availability or past observed events (e.g., Hirschi et al., 2011; Bieli et al., 2015; Lowe et al., 2015; Plavcová and Kyselý, 2019; Zschenderlein et al., 2019; Hartig et al., 2022). Case studies provide useful insights into the dynamics of persistent warm and cold spells, but their results cannot necessarily be generalised. There is consequently value in a more systematic regionalisation of persistent warm and cold spells, as has been attempted in a few studies at the regional scale (e.g. Carril et al., 2008; Stefanon et al., 2012; Sousa et al., 2018).*".
* * *
**Comment 1.3** *In Sect. 2.2.2 the authors define their analysis window. I understand the logic of wanting to study the onset phase until the peak of the warm/cold spells, but defining the first day as the one where the regional temperature anomaly changes sign may introduce an unrealistically long build-up phase for some events. For example, I am not sure that I buy the argument of a wintertime warm spell taking on average more than 2.5 weeks to build up, as suggested by table 1.*

*I also think this makes the reasoning on ll. 91-92 somewhat circular. The authors define a very generous window to analyse the build-up of temperature anomalies, and then argue for the importance of analyzing temperature deviations over a long time period based on their own definition of the build-up period.*

*I would be more convinced if some analysis could show the typical evolution of regional temperatures from the first day of the analysis window and make the case that this reflects a consistent, continuous build-up as opposed to e.g. a period of oscillations around weak anomalies of the same sign and then a rapid increase of absolute anomalies closer to the peak date. Presumably, both from a physical and a more impacts-based perspective, the interest lies in the build-up and persistence of somewhat large temperature anomalies, not in small anomalies, even if persistently of the same sign.*

**Answer**: Following our earlier reply to this comment, we modified the manuscript by expanding the motivation for our choice of analysis window and deleting the comment on ll. 91-92. Please see the following addition to the methods section:

"*For each event, we define an analysis window (during which to calculate circulation and temperature budget anomalies) as the period between the last day before the event on which the region-mean daily temperature anomaly was negative (for warm spells) or positive (for cold spells) and the day during the event on which the daily-mean temperature anomaly peaks (maximum for warm spells, minimum for cold spells; Figure 1). This definition follows Röthlisberger and Papritz (2023a) and Röthlisberger and Papritz (2023b). The reason why we do this, instead of using the original three-week periods that define the spells, is to capture the onset of the persistent warm or cold spells at a daily resolution, and avoid the decay part of the spells when synoptic drivers are likely disappearing. Note that with this definition the analysis window is different for each spell. By looking back at the last day when temperature anomalies changed signs, we can capture the onset and build-up period of the spells, not just their peak. Admittedly, this choice may not always be the most relevant for individual events. However, on average, it allows to capture the period during which temperatures consistently depart from the climatology. One downside is that this definition may favour unrealistically long build-up times and weaken the significance of the detected synoptic anomalies. Another limitation of this choice is that it excludes the days following the peak temperature anomaly, which can still be relevant, for instance in terms of impacts. Still, we show in Figure A2 that spells do tend to develop over multi-week time scales, slowly and persistently deviating from their climatology, which supports our choice of analysis window.*"
* * *
**Comment 1.4** *A separate issue I have with the methodology is that the authors weigh differently each event, and could be giving a disproportionately large weight to events with a long build-up time, regardless of whether these are particularly extreme events or not. What is the range of the analysis windows for individual events and is there a correlation between event severity and duration of the analysis windows within the single regions?*

**Answer**: We added the following lines at the beginning of section 4: "*There is however a large variability in analysis window length from one event to the other (coefficient of variation approx. 0.5 for most regions). Note that because we calculate synoptic anomaly maps for each single event separately (section 2.2.3), longer events do not get more weight than shorter ones in the results.*".
* * *
**Comment 1.5** *Further, I am confused by the description of the data preprocessing. On l. 65 the authors mention normalization of daily temperature using the mean and standard deviation. I assume that the former part entails subtracting the mean. On ll. 137 and following they mention subtracting the long-term average. Is the procedure such that the authors first normalize the data, including subtracting a mean value, then average it regionally and then subtract again a mean value?*

**Answer**: Thank your for pointing this out, we clarified this point in the revision. We added the following paragraph to the data section: "*For the statistical modeling and regionalisation of warm and cold spells, we further normalise daily temperature data on each day and at each grid point by subtracting the mean and dividing by the standard deviation, both being estimated on a 30-day, 7-year moving window (as in Pfleiderer et al. (2019)). This allows us to remove the seasonality and long-term trends so as to focus exclusively on sub-seasonal variability. We only use this normalisation in the quantile regression (section 2.2.1). For anomaly maps during warm and cold spells (section 2.2.3), we use the non-normalised temperature data..*"
* * *
**Comment 1.6** *Finally, as far as I can tell the way statistical significance is computed is never explained in detail. For example, in Fig. 5 and following how is the 90% confidence level computed, and do the authors account for multiple testing?*

**Answer**: Thank you for this comment. We added the following details to section 2.2.3:
"*We then calculate anomaly maps of selected fields (section 2.1) averaged over the analysis windows. For each season, region and event, we average the different fields during the corresponding time window, and subtract the long-term average calculated over the same calendar days across the whole ERA5 record. Anomaly maps for each season and region are then averaged across events for warm and cold spells separately. We assess their statistical significance of anomalies by calculating at each grid point their rank among a sample of 1000 anomalies obtained from randomly generated sets of events with the same distribution of duration and timing of occurrence (calendar days) as observed events. From this rank an empirical p-value is obtained, which we correct for the false discovery rate (Wilks, 2016).*"
* * *
**Comment 1.7** *While some of the regions defined in the paper are relatively intuitive, others look puzzling to say the least. For example, is it really the case that cold spells in the middle-east are part of a coherent region with cold spells in the south-western Sahel (region 2 (or 3, I can't really tell the colours apart) in Fig. 2a)? Similarly, Fig. 2d seems to suggest that heatwaves in Eastern Europe/Western Russia actually belong to three (or even four) separate clusters – something that I do not recall ever having seen in the literature. I am in general not against introducing new definitions to the literature; indeed, it is part of scientific progress. However, when new definitions are presented – in this case of temperature extreme regions – which appear at odds with the "conventional" ones from the previous literature, some more robust justification and contextualization would appear necessary. I find Sect. 6.2 somewhat dismissive, by providing references to three previous regional definitions of temperature anomalies but not going any further in this direction.*

**Answer**: The point of our regionalisation approach is to let the data define the regions themselves. Our results are meant to highlight the spatial connectivity of warm/cold spells in a systematic way (specific events may behave differently) to best analyse the corresponding physical drivers. In practice, we find that neighbouring regions generally experience warm/cold spells at different times. There is certainly some overlap, as we discussed in our previous reply, but in most cases, events in neighbouring regions overlap by less than 20%.

We also expanded section 6.2 by adding some recent references: "*It is not straightforward to compare our regionalisation results to previous findings, because (i) we take a hemispheric perspective, (ii) we look at long (3-week) cold and warm spells and (iii) we limit ourselves to regions where the quantile regression model skill is high. Most previous studies are indeed limited to specific areas (Europe in particular) and focus on shorter-term events, chiefly heatwaves. Still, over Europe during summer, our results are in general agreement with those of Carril et al. (2008) who, from EOFs of monthly temperature anomalies, found a clear tripole structure between Northwestern Europe, the Euro-Mediterranean and Eastern Europe/Western Russia similar to regions 10, 15 and 21 on Figure 3-d. Stefanon et al. (2012) and Pyrina and Domeisen (2023) likewise both found 6 spatial clusters across Europe based on the simultaneity of hot summer days, including a North Sea cluster comparable to region 21 on Figure 3-d, a Western European cluster (region 15), a Scandinavian cluster (region 7), an Eastern European cluster (region 10)*

*and a Russian cluster (region 11). Some of the typical European heatwave patterns of Felsche et al. (2023) are also similar to our Figure 3-d results. Regarding cold spells, Xie et al. (2017) detected three main patterns of winter cold waves in North America (East, Northwest and Centre), two of which bear some resemblance to regions 11 and 13 on Figure 3-a. Region 11 is also similar to the North American region of cold outbreaks triggered by sudden stratospheric warmings identified by Kretschmer et al. (2018).*"
* * *
**Comment 1.8** *A separate comment on the regionalization results is the conflation of land and sea regions. I assume that the composites shown in Figs. 5 and following are centred on the centroid of each region. The composites then show an odd combination of averages over land and sea regions. Variables such as cyclone frequency may be largely governed by specific regions spanning the storm tracks, and much less relevant for others. I suspect that more coherent results may be obtained if, especially for surface and near-surface variables, the authors tried to composite separately land and sea points.*

**Answer**: You are correct and these are limitations which we need to make explicit in the manuscript. Consequently, we added the following paragraph: "*While the goal of our paper is to highlight the similarities in terms of synoptic-scale anomalies during persistent warm/cold spells across the northern hemisphere, there are some limitations relative to the averaging that must be mentioned. First, whereas summer regions are generally confined to land (Figure3-b,d), in winter this is not the case (Figure 3-a,c). As a consequence, averaged anomaly maps shown in Figures 5, 6, 7 and 9 mix together land and ocean grid points. As the particulars of individual regions is not the main focus of the present study, we choose not to treat land and ocean separately, but one should remember that it may affect the results in the aforementioned figures. Since cold spells are advection-driven, land/ocean contrasts probably play a limited role, but for warm spells, surface heat fluxes from the ocean surface may be significant. Second, averaging over regions of different shapes and sizes, and located at different latitudes or different points of the storm tracks, also impacts the results. The analysis of individual regions will be the subject of future work.*".
* * *
**Comment 1.9** *Sect. 5 is an odd section, which seems to be midway between a results and a discussion section. Much of the section is speculative and imprecise, and generally seems to make strong statements without a clear support from the analysis presented in the study. An example is the subsection on Recurrent Rossby Wave Packets. The authors dedicate a subsection to this without ever explaining clearly whether they define in some objective way RRWPs. It is unclear to me how the reference they make to Fig. 12 would support their claims. The subsection is largely based on speculation and analogies with previous results in the literature, rather than an objective analysis. Similarly, to support the statement made on ll. 349-350, one would need to at least show precipitation anomalies in Fig. A6.*

**Answer**: We agree that the content of section 5 should rather be placed in a results section, and therefore we moved it to the end of section 4 ("Synoptic conditions during persistent warm and cold spells"). We also rephrased some of the paragraphs to better explain the analyses that we did (especially the RRWP analysis and the precipitation deficit figure, which we had forgotten to include).

For the precipitation analysis, we added the corresponding panel on Figure A8, as well as the following sentence: "*Regions with particularly high precipitation deficits (Western North*

*America, Western Europe, Kazakhstan and Eastern Siberia) also tend to exhibit the largest sensible heat flux anomalies (Figure A8).".*

For RRWPs we included the following: *"The persistent circulation patterns during warm and cold spells can also be part of recurrent Rossby wave packets (RRWPs). In these synoptic-scale wave packets, individual troughs and ridges amplify repeatedly at the same longitudes, leading to persistent circulation anomalies frequently associated with extreme surface weather (Röthlisberger et al., 2019; Ali et al., 2022). To highlight the role of RRWPs, for each region, we look for statistically significant positive R-metric anomalies during event analysis windows in a longitude interval covering the region and up to 30∘W of the region's westernmost points (to identify potential upstream RRWPs). Results show that persistent spells in a large number of regions are associated with significant RRWP activity (Figure 12),"*

Note that we do not claim to precisely quantify the role of all the physical drivers we identified. In this part we list and discuss the major drivers highlighted by our analysis (and which also concur with the literature). A complete and quantitative analysis would go beyond the scope of our paper.
* * *
**Comment 1.10** *Finally, in the author's intention this section should discuss persistence of temperature extremes, but no definition or quantification of persistence is ever presented. The only place where the authors do define a timescale is in choosing the averaging period for the temperature anomalies and computing the "analysis window length". However, neither the persistence of the warm/cold spells (which may or may not be similar to the analysis window length if one is concerned here with the persistence of somewhat large anomalies) nor the persistence of the atmospheric circulation features associated with these is ever explicitly addressed. I struggle to see how one may make statements on the circulation features explaining the persistence of a given surface feature when persistence is never defined, nor is the duration of either feature quantified.*

**Answer**: Thank you for this comment. As we discussed in a recent preprint, persistence can be defined in many ways, and we clearly need to explicitly detail which one we choose. As you wrote, we consider a fixed timescale to define persistent warm and cold spells. The 3-week time scale is meant to capture persistence over sub-seasonal time scales. We added the following paragraph to the beginning of the methods section (2.2) in the revision:

*"Our proposed regionalisation method brings together grid points where persistent warm or cold spells share a similar dependence on the large-scale circulation. There are many ways to characterise a time series, such as temperature, as "persistent" (Tuel and Martius, 2023). What we consider as "persistent" spells here are periods of time that stretch over sub-seasonal time scales (week to month) and during which temperature anomalies tend to remain either much above or much below zero. This approach corresponds to the "episodic" persistence in the typology introduced by Tuel and Martius (2023). To identify such periods, we consider a fixed sub-seasonal time scale of 3 weeks, and define sub-seasonal warm and cold spells as 3-week periods during which the average temperature is above its 3-week 95th percentile or below its 5th percentile."*

In addition, in response to your Comment 1.3, we added supplementary figure A2 to illustrate the development of the warm/cold spells over sub-seasonal periods.
* * *
**Comment 1.11** *I find the contextualisation of the findings lacking. While in Sect. 6.2 the authors attempt to make a link to previous work which has looked at coherent regions of extreme*

*surface temperature occurrences, what is completely missing is a discussion section making a link to the wealth of literature looking at drivers of different regional cold or warm spells. Admittedly, there is some of this in Sect. 5. However, discussing in more detail the literature related to the mechanisms is a key point needed to contextualise the current analysis relative to the literature, and I would have expected the bulk of the discussion section to be dedicated to this.*

**Answer**: The discussion in section 4.5 (previously section 5) already provides many references to the literature looking at the drivers of warm and cold spells, and we cannot expand it too much for the sake of length. However, some aspects indeed deserve to be further detailed, notably the link of the various mechanisms to the persistence of extreme temperature spells. Consequently, we expanded section 4.5, in particular:

- Section 4.5.1 (on blocking), by adding the following: "*Atmospheric blocks are by definition persistent features (Schwierz et al., 2004), but their typical lifetime is usually on the order of 5-10 days, and longer blocks are extremely rare (e.g., Nabizadeh et al., 2019). The persistence of individual blocks is known to be fueled by upstream latent heating (Steinfeld and Pfahl, 2019), but recurrent blocks are likely also key to the persistence of cold or warm spells over sub-seasonal timescales. For instance, the month-long heatwave over the Baltic in June-July 2022 (Tuel et al., 2022) and the 2010 Russian heatwave (Drouard and Woollings, 2018) were both related to several successive blocking systems.*"

- Section 4.5.2 (on RRWPs), see our answer to your Comment 1.9.

- Section 4.5.1 (on land-atmosphere feedbacks), which we reformulated as follows: "*It has long been argued that land-atmosphere feedbacks are essential for the persistence of summer warm spells (e.g., Lorenz et al., 2010; Mueller and Seneviratne, 2012; Perkins, 2015; Bartusek et al., 2021; Martius et al., 2021). Dry soils indeed increase sensible heat fluxes to the detriment of latent heat fluxes, leading to lower tropospheric warming and drying which further dries the soils and stabilises the atmospheric column (Miralles et al., 2019). This is a well-known mechanism whose role has been highlighted in several recent persistent heatwaves (e.g., García-Herrera et al., 2010; Dirmeyer et al., 2021). All our regions experience precipitation deficits during JJA warm spells, most of which are statistically significant (Figures 11 and A8-a). Over continents, these precipitation deficits translate into increased sensible heat fluxes upwards from the surface (Figure A8-b), which enhance the positive temperature anomalies and partly balance the radiative cooling seen in the negative diabatic tendencies (Figure 11-f). Regions with particularly high precipitation deficits (Western North America, Western Europe, Kazakhstan and Eastern Siberia) also tend to exhibit the largest sensible heat flux anomalies (Figure A8). Land-atmosphere feedbacks thus seem important for the persistence of summer warm spells, especially in Europe and Scandinavia, where they have already been show to play a role (e.g., Dirmeyer et al., 2021). It remains however unclear to what extent dry soils can trigger persistent heatwaves on their own, or whether a dynamical forcing is required (e.g., Domeisen et al., 2023). In our case, we do not find significant precipitation deficits before the start of the persistent warm spells (not shown). A pre-conditioning by dry soils therefore does not appear to be required for persistent warm spells, though they may still play a role for the most extreme warm spells (Hirschi et al., 2011; Stefanon et al., 2012).*"

**Comment 1.12** *ll. 14-15 Perhaps the authors could explain what spatially and temporally compounding events would be in the context of temperature extremes. Presumably concurrent, geographically remote heatwaves or cold spells or successions of heat waves or cold spells at the same location within a given season? Moreover, compound events are never mentioned or discussed in the analysis presented in the paper, so it seems largely irrelevant to discuss them in the introduction.*

**Answer**: This sentence was indeed irrelevant and we removed it.
* * *
**Comment 1.13** *ll. 36-37 I find this statement misleading. First of all, there are indeed papers that have treated temperature extremes on multi-week to monthly timescales (I can think of Galfi and Lucarini (2021) off the top of my head, but a careful literature search would likely bring up others). Second, while many papers do impose a 5-day or similar threshold for heatwave duration, most heatwaves end up being much more persistent than that. It is thus misleading to state that these papers "focused on short-lived events, on the order of a few days only". Indeed, using a few days in duration as a lower threshold can easily lead to identifying multi-week heatwaves, see e.g. Vogel et al. (2020) or Fig. 10 in Grotjahn et al. (2016). This also links to my previous major comment on the motivation for looking at 3-week temperature deviations.*

**Answer**: Thank you for this comment; we modified the introduction to better reflect the literature by adding the following:
"*A handful of studies have looked specifically at prolonged periods of heat and cold, especially over sub-seasonal timescales. Galfi and Lucarini (2021) modeled the marginal probabilities of persistent warm and cold spells of various lengths to describe their climatology across several regions with large deviation theory. Carril et al. (2008) analysed the circulation patterns associated with extreme warm summer months in Europe and the Mediterranean, and Li et al. (2017) did the same for persistent cold spells of at least 10 days in North America. More recently, van Straaten et al. (2022) attempted to discover sources of predictability for monthly warm anomalies in Central Europe.*"
and:
"*[...] previous works have generally used short time windows to define warm or cold spells, on the order of 3-5 days (e.g., Schaller et al., 2018; Plavcová and Kyselý, 2019; Jeong et al., 2021; Jiménez-Esteve and Domeisen, 2022). Heatwaves, for instance, are frequently defined as sequences of at least 3 or 5 extreme warm days, with the longest events usually lasting about a week (e.g., Perkins and Alexander, 2013; Plavcová and Kyselý, 2019). While such windows may also identify multi-week spells, they focus on the part of the spells with the most extreme temperatures. In addition, while persistent spells frequently include short periods of extreme temperature anomalies, the opposite is not necessarily true, i.e. some short-lived, extreme temperature anomalies are not part of multi-week warm or cold spells (Figure A1). Finally, choosing a longer window is useful to identify regions where spells tend to co-occur, as it allows to capture the trajectory of the synoptic-scale systems responsible for the temperature anomalies across their lifetime.*"
* * *
**Comment 1.14** *ll. 63 Since the authors themselves later in the paper mention the challenges of working with 42 years of data, an obvious question is why they have chosen not to take advantage of the ERA5 back-extension, which has been available in preliminary form for over two years and in its final form since mid-2022. The major issues with the preliminary version*

|      | Cold       | Warm      |
| ---- | ---------- | --------- |
| **DJF** | 10-16 (12) | 8-23 (12) |
| **JJA** | 5-13 (9)   | 8-24 (12) |

Table 1: Analysis window length (in days) for 2-week warm and cold spells: inter-region range and median.

|      | Cold         | Warm       |
| ---- | ------------ | ---------- |
| **DJF** | 12-20 (16.5) | 11-25 (18) |
| **JJA** | 8-17 (13)    | 12-27 (17) |

Table 2: Analysis window length (in days) for 4-week warm and cold spells: inter-region range and median.

*were tropical cyclones, which is not something that would have affected the analysis presented here.*

**Answer**: It is true that the ERA5 back extension represents a wealth of new data that upcoming studies should make the most of. However, when the final version came out, we were already very advanced in our analysis, and we decided to stick to the 42-year data.
* * *
**Comment 1.15** *ll. 108 In Fig. 1 many of the patches with DR>0.4 are fragmented. How does the PAM behave when given data with scattered "holes" in it? Does this affect the robustness of the final clustering?*

**Answer**: The clustering algorithm does not take any geographical information as input, so whether the DR map is fragmented or not does not matter. It is true that some regions end up including some fragments (i.e., a few lone points scattered away from the region's main group of points) but these represent at most a few percent of a region's points and do not impact the results. Such fragments are unavoidable in our regionalisation since by choosing a rather small number of regions forces the algorithm to add isolated points to one of the regions. We explain at ll. 117-121 how we remove these isolated points to better visualise the regions.
* * *
**Comment 1.16** *ll. 190-191 It would be interesting for the readers to see these numbers for 2 and 4 week averaging times, e.g as an Appendix table.*

**Answer**: We added the numbers for 2 and 4 week events as appendix tables.
* * *
**Comment 1.17** *Sect. 4.1 How do the authors determine this grouping? Is it through some objective criterion or through a subjective analysis of the individual regions?*

**Answer**: We obtained this grouping by subjective analysis based on the anomaly maps. We made it clear in the revision: "*Comparing circulation anomaly maps across regions, we visually identify three main groups of regions with similar circulation anomalies during persistent DJF cold spells...*"
* * *
**Comment 1.18** *A few typos (although I appreciated how well-written the paper generally was): l. 211 near-surface diabatic*

*ll. 272, 328 Incorrect figure reference?*
*l. 328 (difference in intensity/spatial extent?)*
*ll. 347-348 (Figures 11) and S5)*

**Answer**: Thanks, we corrected these typos in the revision.
* * *
**Comment 1.19** *ll. 296-297 I am not sure I agree with this statement, and indeed the following two sentences seem to counter it. I would suggest adding a "summertime" to the first sentence to avoid confusion.*

**Answer**: We followed your suggestion, thanks.
* * *
**Comment 1.20** *l. 310 and following. The current formulation seems to suggest that there is plenty of work on the role of upstream blocking but that the authors are the first to highlight the role of downstream blocking/processes for the occurrence of upstream temperature extremes. While upstream blocking has certainly received more attention, there are studies which also consider the role of downstream features (e.g. Takaya and Nakamura, 2005; or Lehman and Coumou, 2015, who discuss the role of downstream storm track anomalies in the context of eastern North American extremes).*

**Answer**: Thank you for this comment. We refomulated the sentence to avoid misleading readers om the previous literature ("*First, atmospheric blocking plays an important role in the persistence of both warm and cold spells in summer and winter. Our hemispheric perspective highlights the various influences of blocking that have been discussed in a host of previous studies...*") and added the Takaya and Nakamura citation.
* * *
**Comment 1.21** *Fig. 5 and following: using green dots on a red background is not ideal for many readers (including this reviewer).*

**Answer**: Thank you for pointing it out. We changed to yellow dots in all relevant figures.
* * *
**Comment 1.22** *Fig. 12 If the authors only look upstream of the regions (30 W of the westernmost point as stated in the figure caption) how can they diagnose downstream blocking as again stated in the caption?*

**Answer**: We only look 30°W upstream for the R-metric anomalies, not for the blocking (the caption states that "Hatching indicates significant positive blocking frequencies up- or downstream of the region". We added "*not restricted to 30° W*" to the caption to make this more explicit.
* * *
**Comment 1.23** *On a separate point, it may be worth explaining in the methods what the R-metric is and how it is being used here.*

**Answer**: We added the following paragraph to the data section:
"*To assess the potential role of recurrent synoptic-scale Rossby wave patterns, we use the R-metric developed by Röthlisberger et al. (2019). The R-metric is calculated from conventional Hovmoller diagrams of 6-hourly 35–65∘N averaged 250 hPa meridional wind. We first apply apply a 14-day running mean to the series, and remove contributions outside the synoptic wavenumber range k = 4–15. The R-metric is then defined as the absolute value of the time- and wavenumber-filtered signal. Large R-metric values therefore tend to indicate the presence of several successive synoptic-scale wave packets, in other words recurrent Rossby waves.*"

**Response to reviewer comments**
* * *
**Comment 1.1** *The manuscript is well-written and the figures are of high quality. However, I have significant concerns about the methods, the implication of the results, and the comparison with the existing literature. In my opinion, the study does not address the research question stated in the title.*

**Answer**: Thank you for your comment. We agree that our initial title was misleading: we do not analyse the persistence of warm and cold spells per se, but the characteristics of persistent warm and cold spells (which is different). We modified the title accordingly. We hope our replies below address your concerns as to the methodology and comparison with existing literature.
* * *
**Comment 1.2** *In the Introduction, the authors identify two gaps in the literature on the processes responsible for warm and cold spells: (i) persistence and (ii) spatial dependence. With respect to the first gap (persistence), what can we learn meteorologically from longer lasting events that we don't already know from the existing literature? There is already a lot known on driving factors for heat and cold waves. I am not arguing that it is not interesting to look at the persistence, but the authors insufficiently motivate why it could be interesting to look at persistent events (apart from impacts). In addition, a minimum duration of three days does not exclude long-lasting events, e.g. in Zschenderlein et al. (2019), a heat wave with a duration of more than 40 days was included in the analysis. It is therefore not true that previous work generally focused on short-lived events only, as the authors claim.*

**Answer**: Thank you for your comments. We reworked our introduction to further motivate our work and discuss the choice of the 3-week time scale. Specifically, we expanded the second paragraph as follows:

*"There already is substantial literature on extratropical warm and cold extremes and their driving factors (e.g., Domeisen et al., 2023). Among these, atmospheric blocking has long been recognized as a key driver of temperature extremes in the mid- to high-latitudes, for both summer warm spells and winter cold spells (Buehler et al., 2011; Pfahl and Wernli, 2012; Schaller et al., 2018; Kautz et al., 2022). Topography (Jiménez-Esteve and Domeisen, 2022) and land-atmosphere interactions (Bieli et al., 2015; Miralles et al., 2019; Wehrli et al., 2019) also play important roles, as does polar vortex variability for cold air outbreaks in mid- and high-latitudes (Kolstad et al., 2010; Biernat et al., 2021; Huang et al., 2021). However, previous works have generally used short time windows to define warm or cold spells, on the order of 3-5 days (e.g., Schaller et al., 2018; Plavcová and Kyselý, 2019; Jeong et al., 2021; Jiménez-Esteve and Domeisen, 2022). Heatwaves, for instance, are frequently defined as sequences of at least 3 or 5 extreme warm days, with the longest events usually lasting about a week (e.g., Perkins and Alexander, 2013; Plavcová and Kyselý, 2019). While such windows may also identify multi-week spells, they focus on the part of the spells with the most extreme temperatures. In addition, while persistent spells frequently include short periods of extreme temperature anomalies, the opposite is not necessarily true, i.e. some short-lived, extreme temperature anomalies are not part of multi-week warm or cold spells (Figure A1). Finally, choosing a longer window is useful to identify regions where spells tend to co-occur, as it allows to capture the trajectory of the synoptic-scale systems responsible for the temperature anomalies across their lifetime."*
* * *
**Comment 1.3** *With respect to the second gap (spatial dependence), the authors argue that the recent literature looked at arbitrary regions and mixed up areas where warm or cold spells are shaped by different physical processes. They are by far not arbitrary in the recent literature, either regions are motivated by impacts (which is perfectly fine and definitely not arbitrary) or by different climates (e.g. humid vs. dry climate). There are also studies looking at the different physical processes in a region, e.g. Sousa et al. (2018, Fig. 5). It is therefore not overlooked that in one region the processes can differ regionally.*

**Answer**: We agree that "arbitrary" was a very wrong choice of word. Here, we are interested in a purely meteorologically-driven regionalisation to highlight (i) the spatial co-occurrence of persistent spells and (ii) the influence of different drivers. We modified the introduction to highlight this argument: "*Still, much remains unknown about the geography of persistent warm and cold spells, their spatial dependence, and how the role of various identified drivers varies in space. Such information is essential for risk assessment and to improve forecasts (Perkins, 2015; van Straaten et al., 2022). Persistent spells indeed typically occur over large spatial scales, from hundreds to thousands of square kilometers, with potentially complex spatial footprints (Lyon et al., 2019; Vogel et al., 2020). Yet, most previous work on heat waves, for instance, has focused at the grid-point scale or over regions based on impacts, administrative boundaries, data availability or past observed events (e.g., Hirschi et al., 2011; Bieli et al., 2015; Lowe et al., 2015; Plavcová and Kyselý, 2019; Zschenderlein et al., 2019; Hartig et al., 2022). Case studies provide useful insights into the dynamics of persistent warm and cold spells, but their results cannot necessarily be generalised. There is consequently value in a more systematic regionalisation of persistent warm and cold spells, as has been attempted in a few studies at the regional scale (e.g. Carril et al., 2008; Stefanon et al., 2012; Sousa et al., 2018).*".
* * *
**Comment 1.4** *The study promises a lot to provide new insights into the persistence and spatial dependence of warm and cold spells. However, both aspects are not really quantified in the study. I detail my comments below:*
*1. Persistence: Sections 4 and 5 are believed to address this research question. However, these sections rather analyse synoptic conditions during warm and cold spells (which is nicely done) and identify three atmospheric drivers (upstream blocking, downstream blocking, subpolar troughs). Many studies already investigated the position of the block relative to the high/low temperature extreme at the surface (e.g. Pfahl and Wernli 2012, Pfahl 2014, Bieli et al., 2015, Santos et al., 2015, Sousa et al. 2018, Zschenderlein et al., 2019). The authors should therefore make the new findings more clear. While I find the synoptic analysis very nice, I don't understand what this has to do with persistence. And this is the main aspect of the paper (at least it is stated in the title). For example, how important is the position of the block relative to the warm/cold spell with respect to the persistence in comparison to a heat/cold spell of about 5 days? I think the authors try to discuss the persistence aspect in Section 5, but not really successful. They are rather speculating about the role of blocks, RRWPs, land-atmosphere feedback, and subpolar troughs (with results from the existing literature and not their own) and not quantifying the importance of these processes with respect to persistence.*

**Answer**: Thank you for your comments. The choice of title was clearly misleading, and we modified it accordingly to "*Persistent warm and cold spells in the Northern Hemisphere extratropics: regionalisation, synoptic-scale dynamics, and temperature budget*", since we do not explicitly analyse the mechanisms causing spells to persist or not. Such a detailed analysis would

indeed require investigating warm and cold spells across different timescales to understand what determines their length, and is beyond the scope of the present manuscript (but will be the subject of future work). We hope that our analysis and discussion are now more in line with the title. In particular, an important outcome of our study is the regionalisation, as well as the comparison between regions it allows to make in terms of driving mechanisms. We consequently expanded the corresponding discussion (please see our answer to your Comment 1.5.).
* * *
**Comment 1.5** *2. Spatial dependence: I think the authors want to address this research question with the regionalisation approach (Fig. 3). But in the end, they summarise all regions into three categories (and they don't describe how they do it, see also my minor comments). What is the added value of the regionalisation? What can we learn from it? What do the different colors in Fig. 3 imply meteorologically? In my view, Figure 3 is insufficienty discussed.*

**Answer**: The regionalisation approach is indeed meant to highlight the spatial connectivity of persistent warm and cold spells. This is something we should have stressed more and which we therefore developed in the revised manuscript. Discussing what the regionalisation implies meteorogically is difficult to do in detail at it would require analysing each region separately, which cannot be done in a single paper, but we expanded the discussion of Figure 3 in section 5.2 by adding the following:

*"While our goal is not to discuss individual regions in detail, some general remarks can be made about the regionalisation results shown on Figure 3. First, it is important to note that, as with any regionalisation approach, our results are dependent on the number of regions we select. The distance metric we use implies that regions should be understood as areas where warm/cold spells are related to similar large-scale circulation patterns, and therefore where spells tend to occur at approximately the same time. The subsequent synoptic analysis showed that our choice of regions seems consistent with the corresponding physical drivers. Too few regions would likely have blurred the significant signals in e.g. blocking location. In practice, we also find that neighbouring regions generally experience warm/cold spells at different times. While there is some overlap, in most cases, events in neighbouring regions overlap (in time) by less than 20%. Still, our proposed regions are not necessarily the most relevant from an impacts perspective, notably because (i) the temperature threshold we consider are relative, not absolute, and (ii) the extremeness of the temperature anomalies is softened by averaging over large scales.*

*Second, the difference in the spatial footprint and mean zonal extent of regions between DJF and JJA, while certainly impacted by the choice of region number, is consistent with planetary waves having longer wavelengths in the cold season. This leads to more extent temperature anomalies at the surface, e.g., the well-known North American dipole (Singh et al., 2016) which may correspond to regions 11 in Figure 3-a and 15 in Figure 3-c. Another important feature of the winter season is that temperature gradients are not only meridional, but also zonal. This is much less the case in summer when meridional gradients dominate. Zonal circulations can therefore lead to strong cold advection in winter. Our temperature budget analysis shows that cold spells are primarily advection-driven, in both winter and summer. The cold air in summer essentially comes from the high latitudes, and a strongly meridional circulation is required to bring it equatorwards, consistent with rather narrow regions in Figure 3-b. By contrast, in winter, land/ocean temperature contrasts are steep, and zonal circulations can bring about strong cold (or warm) advection, consistent with the large zonal extent of some regions like e.g., region 13 on Figure 3-a. Locally, some particularly zonally elongated regions like regions 6 and 23 over Europe in Figure 3-a, are also likely related to strong zonal circulations (in this case the*

*North Atlantic Oscillation) that can lead to persistent surface temperature patterns. Further differences in the regionalisation between winter and summer may be due to the seasonality of blocking (Steinfeld and Pfahl, 2019) or RRWPs (Röthlisberger et al., 2019) (in terms of location, extent and intensity).".*
* * *
**Comment 1.6** *I am not convinced by the method for the identification of persistent warm and cold spells. While I understand that you incorporate the days prior to the spells to investigate the onset, I do not understand why your analysis window ends with the peak temperature anomaly. What if this anomaly is reached already very early during the warm/cold spell and high/low temperatures still persist after that peak?*

**Answer**: There are certainly limitations to our choice of analysis window, as we detailed in our initial reply. Our choice however allows to capture, on average, the onset and build-up phase of the spells. Individual events will behave differently from the average behaviour, but we are precisely interested in the average. We modified/expanded the corresponding paragraph of the methods section to further justify our choice and state its limitations:

"*For each event, we define an analysis window (during which to calculate circulation and temperature budget anomalies) as the period between the last day before the event on which the region-mean daily temperature anomaly was negative (for warm spells) or positive (for cold spells) and the day during the event on which the daily-mean temperature anomaly peaks (maximum for warm spells, minimum for cold spells; Figure 1). This definition follows Röthlisberger and Papritz (2023a) and Röthlisberger and Papritz (2023b). The reason why we do this, instead of using the original three-week periods that define the spells, is to capture the onset of the persistent warm or cold spells at a daily resolution, and avoid the decay part of the spells when synoptic drivers are likely disappearing. Note that with this definition the analysis window is different for each spell. By looking back at the last day when temperature anomalies changed signs, we can capture the onset and build-up period of the spells, not just their peak. Admittedly, this choice may not always be the most relevant for individual events. However, on average, it allows to capture the period during which temperatures consistently depart from the climatology. One downside is that this definition may favour unrealistically long build-up times and weaken the significance of the detected synoptic anomalies. Another limitation of this choice is that it excludes the days following the peak temperature anomaly, which can still be relevant, for instance in terms of impacts. Still, we show in Figure A2 that spells do tend to develop over multi-week time scales, slowly and persistently deviating from their climatology, which supports our choice of analysis window.*"
* * *
**Comment 1.7** *And for the quantile regression I would exclude grid points in the tropics beforehand, because a low Z500 variability in this area is no surprise.*

**Answer**: The quantile regression is applied to each grid point separately, so there is no influence of e.g. the tropics onto the mid-latitudes. As to the regionalisation, we remove grid points with low model skill (deviance ratio below 0.4) beforehand, which includes the vast majority of the tropics.
* * *
**Comment 1.8** *The results of this study are not set into context with the existing literature. An attempt is made in Sections 5 and 6.2, but it is not clear what new knowledge is introduced by the study.*

**Answer**: Please see our answer to your Comment 1.4.
* * *
**Comment 1.9** *L5-10 (Abstract): All listed processes (blocks important precursors of warm and cold spells, location of the blocks, recurrent Rossby waves, land-atmosphere feedbacks) are already described in many papers (see list in major comment 2) or in a new review paper by Domeisen et al. (2022, "Atmospheric processes" section; the paper was probably not yet published at the time the authors submitted to WCD).*

**Answer**: Our findings are indeed consistent with the literature. Please see our reply to your Comment 1.4. We included the recent reference of Domeisen et al. to the manuscript.
* * *
**Comment 1.10** *L65: Does this mean that you first subtract the mean and divide by the standard deviation?*

**Answer**: Yes, please see our next answer.
* * *
**Comment 1.11** *L66-68: Why do you use different methods for the trend removal of T2M and Z500?*

**Answer**: We reformulated this paragraph to make our processing clearer. T2M is treated differently for the quantile regression analysis only. We added the following: "*For the statistical modeling and regionalisation of warm and cold spells, we further normalise daily temperature data on each day and at each grid point by subtracting the mean and dividing by the standard deviation, both being estimated on a 30-day, 7-year moving window (as in Pfleiderer et al. (2019)). This allows us to remove the seasonality and long-term trends so as to focus exclusively on sub-seasonal variability. We only use this normalisation in the quantile regression (section 2.2.1). For anomaly maps during warm and cold spells (section 2.2.3), we use the non-normalised temperature data..*"
* * *
**Comment 1.12** *L70: Is the persistence criterion of 4 days valid for an individual grid point or for the whole cyclone mask?*

**Answer**: It applies to the cyclone as a system (this is a Eulerian persistence criterion).
* * *
**Comment 1.13** *L70-80: Should be moved to the methods section.*

**Answer**: These lines are indeed a mix between data and methods, but we would prefer to focus the methods section on the core methods for this paper (modeling and analysis) rather than on the data pre-processing.
* * *
**Comment 1.14** *L84: Is the percentile based on daily or 3-weekly values? And please do not use different words for the same method (normalise, rescaled)*

**Answer**: Thank you for this comment. The percentile is indeed calculated on 3-week averages. We slighlty expanded the sentence to make this clear: "*Warm and cold spells are defined as 3-week periods during which the average temperature is above its 3-week 95th percentile or below its 5th percentile. Temperature is first normalised (section 2.1) to remove seasonal and long-term trends.*"

**Comment 1.15** *L89: Why non-overlapping 3-week intervals?*

**Answer**: We want the intervals to be non-overlapping to avoid as much dependency between model input data points as possible.
* * *
**Comment 1.16** *L105: How sensitive are your results with respect to the DR threshold?*

**Answer**: We chose this specific threshold (0.4) to capture about as much of the extratropics in all seasons as possible. A lower threshold would lead to more tropical grid points retained in the regionalisation (extratropical regions remaining essentially the same in the regionalisation algorithm). A higher threshold removes points from the extratropics and quickly tends to decrease the "optimal" number of regions.
* * *
**Comment 1.17** *L112: What is a silhouette coefficient?*

**Answer**: We added the following to the manuscript: "*The silhouette coefficient is a common metric used to identify an "optimal" cluster number. It essentially compares the mean intra- and inter-cluster distances, with higher values indicating a better clustering (high inter-cluster distance and low intra-cluster distance)*".
* * *
**Comment 1.18** *L129: Why not using the back-extension of ERA5?*

**Answer**: It is true that the ERA5 back extension represents a wealth of new data that upcoming studies should make the most of. However, when the final version came out, we were already very advanced in our analysis, and we decided to stick to the 42-year data.
* * *
**Comment 1.19** *L135: Is the decay phase of the event not interesting?*

**Answer**: It certainly is, but not in the context of our paper which focuses on the mechanisms responsible for the onset and maintenance of the warm and cold spells. See our reply to your Comment 1.6.
* * *
**Comment 1.20** *L143: median point – do you mean centroid?*

**Answer**: No, here we take the actual (geographical) median point, and not the cluster centroid (which in PAM are called medoids, and minimise the distances to all other cluster points). We specified "geographical" in the text to make this clear.
* * *
**Comment 1.21** *Figure 1: I don't understand the values on the y-axis. Is it normalised with the 95th percentile? I also do not understand why the grey shaded area goes from approximately day 11 to 31.*

**Answer**: Thanks for this comment; we reformulated the caption to make all this clearer: "*From the region-averaged normalised (i.e., unitless; see section 2.1) daily temperature series, a warm spell (gray shading) is identified as an extreme 3-week average anomaly (here, from days 11-31). We then identify the corresponding analysis window as the period between the last day before the event on which the temperature anomaly was negative (day 6) and the day during the event*

*on which the temperature anomaly peaked (day 28). Cold spells are identified in a symmetrical way. Note that the reference day value for this example was artificially set at 0."*
* * *
**Comment 1.22** *Section 4.1: How have you identified the three main groups?*

**Answer**: We identified the main groups visually, after removing the tropical regions. We made this explicit in the revision.
* * *
**Comment 1.23** *Figure 5 (and others): Mark the centroid of the regions. Add wind directions or Z500 mean values.*

**Answer**: By construction, the centroid of the regions is at the origin in these figures. The mean Z500 and wind fields are not as meaningful as the anomaly fields here since we average across regions located at different latitudes/longitudes.
* * *
**Comment 1.24** *General remark to the composites: I personally think that compositing can be quite dangerous here, because you are averaging over areas with different sizes and shapes. Can you comment on that?*

**Answer**: Following our earlier reply, we added a paragraph stating the limitations relative to the averaging over regions of different sizes/shapes/latitudes and over land/ocean grid points: *"While the goal of our paper is to highlight the similarities in terms of synoptic-scale anomalies during persistent warm/cold spells across the northern hemisphere, there are some limitations relative to the averaging that must be mentioned. First, whereas summer regions are generally confined to land (Figure3-b,d), in winter this is not the case (Figure 3-a,c). As a consequence, averaged anomaly maps shown in Figures 5, 6, 7 and 9 mix together land and ocean grid points. As the particulars of individual regions is not the main focus of the present study, we choose not to treat land and ocean separately, but one should remember that it may affect the results in the aforementioned figures. Since cold spells are advection-driven, land/ocean contrasts probably play a limited role, but for warm spells, surface heat fluxes from the ocean surface may be significant. Second, averaging over regions of different shapes and sizes, and located at different latitudes or different points of the storm tracks, also impacts the results. The analysis of individual regions will be the subject of future work."*.
* * *
**Comment 1.25** *L225: Be careful because you are mixing land and ocean grid points (see red colours in Fig. 4).*

**Answer**: Please see our answer to your Comment 1.24.
* * *
**Comment 1.26** *Figure 6: The scale in Fig. 6c differs from Fig. 5c!*

**Answer**: The scale varies depending on the figure, otherwise the panels would not be easily readable.
* * *
**Comment 1.27** *L250: These results are in line with Zschenderlein and Wernli (2022).*

**Answer**: Which study are you referring to? The only one we could find has been withdrawn.

**Comment 1.28** *Section 4.4 and Figure 12: I am not able to follow because you haven't described the R-metric sufficiently.*

**Answer**: We added the following paragraph to describe the R-metric:
"*To assess the potential role of recurrent synoptic-scale Rossby wave patterns, we use the R-metric developed by Röthlisberger et al. (2019). The R-metric is calculated from conventional Hovmoller diagrams of 6-hourly 35–65 N averaged 250 hPa meridional wind. We first apply apply a 14-day running mean to the series, and remove contributions outside the synoptic wavenumber range k = 4–15. The R-metric is then defined as the absolute value of the time- and wavenumber-filtered signal. Large R-metric values therefore tend to indicate the presence of several successive synoptic-scale wave packets, in other words recurrent Rossby waves.*"

**Comment 1.29** *L290: remove the second "is"*

**Answer**: Corrected, thanks.

**Comment 1.30** *L310: This agrees with a host ... → ok, but where is the new knowledge regarding the persistence?*

**Answer**: Please see our answer to your *Comment 1.8*.

**Comment 1.31** *L312-313: This is probably also true for events on the shorter time scales.*

**Answer**: You are right; we added the Takaya and Nakamura (2005) reference which discusses the case of East Asian cold waves.

**Comment 1.32** *L318: remove half sentence in brackets at the end of the line.*

**Answer**: Corrected, thanks.

**Comment 1.33** *L348-350: This is not clear to me. Increasing surface heat fluxes would imply diabatic heating, but we see diabatic cooling in Fig. 11f. Radiative cooling is typically dominant in the mid- to upper-troposphere.*

**Answer**: In Figure 11f, the diabatic heating term is calculated as the residual of the energy balance equation. Therefore it is not restricted to radiative cooling. What likely happens however is that the radiative cooling term outbalances surface sensible heat fluxes, since it has to balance not only sensible heat fluxes, but also adiabatic warming and horizontal advection. Note that our analysis also covers day and night, and that sensible heat fluxes peak during the day but are weaker at night, while radiative cooling can remain large across the clock.

**Comment 1.34** *L351-352: What can we learn from this sentence?*

**Answer**: We mentioned this because a number of previous studies have argued that extreme warm spells were often associated with antecedent precipitation deficits. However, in our case we probably do not look at event that are extreme enough to see the connection.

**Comment 1.35** *Section 5.4: You discuss how a subpolar trough develops. But I don't see how you discuss subpolar troughs and their connection to persistence of warm and cold spells.*

**Answer**: You are right that we do not discuss the development of subpolar troughs, but rather point out that persistent wintertime cold spells at high latitudes are likely associated with such systems (the analysis of synoptic-scale anomalies points in that direction). They connect to persistent cold spells because of the associated circulation anomalies, but also because such systems have a general tendency to persistence, as we discuss in the manuscript (cf. also Woollings et al. (2023); https://doi.org/10.5194/wcd-4-61-2023).
* * *
**Comment 1.36** *Section 6.1: A matter of taste, but the discussion of Fig. 3 appears quite late in the study. I would prefer to discuss the results earlier.*

**Answer**: We expanded the discussion of Figure 3 in the revision, please see our answer to your Comment 1.5.
* * *
**Comment 1.37** *L383: Acronym S2S not introduced*

**Answer**: Corrected, thanks (S2S = "subseasonal-to-seasonal")

---

## Author Response (AR2)

**Response to reviewer comments**
* * *
**Comment 1.1** *The authors have conducted additional analyses and implemented several textual edits. They have responded to a number of my comments, but I find their response to my main comment, namely whether they can demonstrate the relevance of picking a 3-week timescale, very weak. Indeed, several of the citations they have added to the text actually work against their argument (see Comment 1.1b below), and they never replied to my question of whether it is possible to identify a set of high-impact events that is overlooked by a 5-day minimum persistence criterion but captured by their own definition, or whether they could make a clear case for events whose meteorological drivers would be confounded if using a 5-day minimum persistence criterion.*

**Answer**: Thank you for your detailed comments. It appears that your major concern regarding our paper remains our choice of the 3-week timescale. You argue that the events thus identified are likely the same as those which would be identified with a shorter (e.g., 5-day) timescale. We realise our initial response had not been clear enough in that regard. We therefore conducted an additional analysis, namely to calculate the fraction of 3-week events we identify which do not include 3- or 5-day heat/cold events. The latter are defined as in the literature, *i.e.* by looking at consecutive runs of 3 or 5 days with extreme temperatures (extreme being defined on a daily basis). See Figures R1 and R2. Note that requiring strictly consecutive daily extremes or allowing for a 1-day gap in the series makes little difference (Figures R3 and R4).

Results show that a significant fraction of the 21-day spells we identify do not overlap with the more classic 3- or 5-day extremes. This is especially true for summer, especially summer cold spells (Figure R1-b), while in winter, only 0-20% of 21-day spells do not include short periods of extreme temperatures.

There is therefore an undeniable overlap between our persistent spells and the more traditional, short extreme hot/cold spell definition. However, our argument isn't about identifying pointwise events so much as identifying the temporal extent of events (see our answer to your Comment 1.3). We do not really see the point of showing that we capture many events with a three-week timescale that are different from the events that are captured when working with 5-day timescales. The whole point of our argument is that when working with 3- or 5-day timescales, one will not necessarily know if the event lasted longer than 3 or 5 days. And the key here is our focus on the longer-lasting events. This is precisely why we adopt the 3-week timescale. We are not interested in whether the longer-lasting events include short periods of extreme daily temperatures. The question is whether these periods indeed correspond to persistent warm/cold anomalies.

We replaced Figure A1 in the manuscript with the present Figure R1, and expanded the text as follows. First, we replaced the reference to Figure A1 in the introduction by "*In addition, a sizeable fraction of persistent spells do not include short periods of extreme temperature anomalies, especially in summer (Figure A1).*". Second, we added the following sentence to the regionalisation discussion of section 5.2: "*Figure A1 shows that while there is certainly a lot of overlap between our 21-day spells and more traditional 3-day extremes, the two are not the same, notably during summer (Figure A1-b,d), which makes a direct comparison difficult.*"
* * *
**Comment 1.2** *I only partly agree with the authors' argument that "while persistent spells frequently include short periods of extreme temperature anomalies, the opposite is not true".*

[Figure]

Figure R1: Fraction of 21-day persistent warm/cold spells which do not include a 3-day extreme warm/cold period, for (a) DJF cold spells, (b) JJA cold spells, (c) DJF warm spells and (d) JJA warm spells. 3-day extreme periods are defined based on the daily 5th/95th temperature percentiles.

*Assume we call 3-week events group "A", and 5 or 3-day events group "B". If A and B include a similar number of events, then if group A events frequently include group B events, it is a tautology that group B events must frequently be part of group A events. The authors' statement can only be true if one has many more short periods of extreme temperature anomalies (group B) than persistent spells (group A). If I understood correctly, in their example the authors define extremes using 5-day mean values, thus providing something of a self-fulfilling prophecy (as I assume that this makes the number of events in B larger than the number of events in A – I would recommend indicating sample sizes in the caption to Fig. A1). However, the definition typically used in the literature is 5 (or some other similar number) of (near-)consecutive days above a single-day persistence threshold (this is true of all three papers the authors cite on l. 45. For example, Jiménez-Esteve and Domeisen (2022), define heatwaves using consecutive exceedances*

[Figure]

Figure R2: Same as Figure R1, but for 5-day extreme periods.

*of single day 95th percentiles). The authors' example thus does not reflect the conventional definition used in the literature. I am doubtful that if the authors were to use single-day thresholds as in the papers they cite, and say a 5-day consecutive exceedance criterion, the results they show in Fig. A1 and that they describe in the text would still hold.*

**Answer**: We fully agree that our argument only holds when there are more events in group B than in group A. But with our event definition, this is precisely the case (something we hadn't said explicitly, though). Since there are many more independent 5-day periods than 3-week periods, one will necessarily find more extreme warm or cold spells using a 5-day window than with a three-week window. In any case, we removed this figure and replaced it with Figure R1.
* * *
***Comment 1.3*** *Besides this, I still do not see a clear-cut case for why one should be interested in persistent above-average temperatures that may never be extreme in terms of daily values. In my original review, I asked whether the authors could identify a set of high-impact events that*

[Figure]

Figure R3: Same as Figure R1, but where a 1-day gap is allowed when defining the 3-day extreme periods.

*is overlooked by a 5-day minimum persistence criterion but captured by their own definition. Similarly, I asked whether they could make a clear case for events whose meteorological drivers would be confounded if using a 5-day minimum persistence criterion. So far, the only argument concerning these points that the authors have put forth is that choosing a longer window is useful to identify regions where spells tend to co-occur. That may be true, but does it really provide an advantage over a simple lagged co-occurrence analysis as often performed for shorter events? On this same point, several of the citations that the authors have added to the revised text to support their claim of relevance of prolonged spells, actually use short thresholds to define extremes (e.g. Añel et al. (2017) use three days running means to define their case-study cold spell). Similarly, Chapman et al. (2020) chose to highlight in their abstract 5 and 7-day persistence periods, much closer to the conventional 3-5 day definitions than the 3-week definition proposed here. This supports the idea that, even if there is an impact dependence related to event duration, this is successfully captured by conventional shorter definitions of*

[Figure]

Figure R4: Same as Figure R2, but where a 1-day gap is allowed when defining the 5-day extreme periods.

*extreme temperatures. Returning to my question above, it is probably relatively easy to find high-impact persistent warm or cold periods. However, I would expect that it is hard to find high-impact persistent warm or cold periods which would not be detected by conventional 3 or 5 day heatwave/cold spell definitions (these would be "persistent spells which do not include short periods of extreme temperature anomalies" to paraphrase the authors' response).*

**Answer**: First, the fact that persistent warm or cold spells lead to significant impacts is clearly demonstrated in the literature, as supported by the studies we cited. Daily temperatures do not need to be consistently extreme to cause high impacts, if the temperature anomaly lasts long enough. Classic heatwave impacts focus on daily temperature extremes, but at S2S timescales, persistent warm or cold temperatures can still lead to major impacts, not because of their extremeness on a daily basis, but because of their persistence. Note that Añel et al. (2017) do not define cold/heatwaves with a 2-day threshold. They provide a literature review of case

studies, including ones for which the cold/heatwaves lasted several weeks. Chapman et al. (2020) also specifically discuss the case of long-lasting heat events (15 days and longer). It is true that few papers systematically analyse long warm or cold spells and their impacts (hence also the novelty of our study). In any case, whether persistent warm/cold spells lead to high impacts or not (independent of their including short periods of extreme daily temperatures), we are interested in characterising their persistence. Our paper is not focused on demonstrating the impacts of long warm/cold spells with new data. We added a mention in the introduction of this year's persistent heat in North America that led to massive impacts on human and natural systems.

Second, we agree, as stated previously, that many persistent cold or warm spells do include short time periods (2-5 days) with extreme daily temperature. We do not claim to be looking at a completely different category of events. However, we argue that to consider the persistence of warm or cold anomalies, one should look beyond the classic 2-, 3- or 5-day timescale and consider S2S timescales. The question isn't whether the events are also detected with a 3 or 5-day timescale, but what is the timescale that best captures the full extent of the events.
* * *
**Comment 1.4** *I thank the authors for having contextualized their statement. A small addition I would suggest is to explicitly state in the text what they explain in their reply, namely that the focus here is on a purely meteorologically-driven regionalization.*

**Answer**: Thank you for this comment. We rephrased the beginning of the last paragraph of the introduction as follows: "*Here, we introduce a simple, meteorologically-driven regionalisation method for sub-seasonal warm and cold spells.*"
* * *
**Comment 1.5** *Fig. A2 is a very nice complement to the original analysis, and the wording in the text now provides an honest appraisal of possible issues with the chosen definition. I believe it makes the point that indeed the authors' definition does artificially extend the build-up period, but I leave the editor and the authors to evaluate whether this may be an issue or not. Wanting to reflect the physical build-up period when a rapid temperature change occurs, I would by eye set a mean threshold between 5 and 10+ days after the shown mean values (I struggle to interpret the median vertical lines given that only the mean temperature anomaly is shown in the figure).*

**Answer**: You are right to point out that the mean values (solid vertical lines) are less representative of the true beginning of the build-up period as the median (dashed vertical lines). We didn't show the median temperature series because they were virtually the same as the mean series.
* * *
**Comment 1.6** *Thank you for the clarification, this makes more sense now. To avoid any possible confusion you could perhaps reiterate that the compositing is done for every event separately in the first or second sentence of Sect. 2.2.3.*

**Answer**: Thank you, good point. We rephrased as follows: "*Anomalies are calculated for each event within each season and each region separately. We average the different fields during the corresponding event time window, ...*"
* * *
**Comment 1.7** *Since the authors now have a new "caveats" section they could discuss there (or*

*in the conclusions) this point and explicitly mention the ERA5 back extension.*

**Answer**: We added the following to the "Limitations" section: "*Last, we note that ERA5 is now available back to 1940 (83 years vs. the 42-year dataset used in this study). Considering this back-extension would allow to identify many more warm and cold spells and draw more robust conclusions.*"
* * *
**Comment 1.8** *l. 60 Misspelled reference to Röthlisberger, also recurs later in the text.*

**Answer**: We couldn't find where the name was misspelled. This might be a compilation issue with the "ö" sign in latexdiff.
* * *
**Comment 1.9** *Figure A8. Describe the stippling in the figure caption.*

**Answer**: Thanks for noticing. We added the following to the caption: "*Hatching in (a) indicates the lack of statistical significance of the region-average anomaly at the 95% confidence level. Stippling in (b) indicates statistical significance of the region-average anomaly at the 95% confidence level.*"
* * *
**Comment 1.10** *The revised study has two "Table 1"s.*

**Answer**: Fixed, thanks.

---

## Author Response (AR3)

Dear Editor,

Many thanks for your detailed response and constructive comments, as well as for pointing out this recent paper which we added to our references. We hope our answers below and the corresponding revision will bring the nuance the reviewer hoped for, and address all remaining concerns.

Best wishes,

A. Tuel and O. Martius

**Response to reviewer comments**
* * *
**Comment 1.1** *I thank the authors for their further set of replies. I am still not entirely convinced by some of the arguments that they put forward, and I somewhat struggle to grasp the relevance of the analysis. At the same time, the analysis itself is rigorous and appears scientifically robust. At this stage in the revision process I see no point in further delaying the publication of the study, and recommend acceptance subject to a final round of minor revisions.*

**Answer**: Thank you for your final comments and support for the publication of our study. We hope our answers below will address your remaining concerns.
* * *
**Comment 1.2** *I find Fig. R1 valuable, as it provides readers with an indication of how closely one can expect the results in the present paper to relate to studies implementing the more commonly-used 3/5 day temperature extremes definition. Perhaps, in the caption to Fig. A1, the authors could specify that the definition is: "based on continuous exceedances of the daily 5th/95th temperature percentiles".*

**Answer**: Good suggestion, we reformulated as "3-day extreme periods are defined based on continuous exceedances of the daily $5^{\text{th}}/95^{\text{th}}$ temperature percentiles in each region."
* * *
**Comment 1.3** *It would be useful to provide somewhere (in an Appendix table or in conjunction with the discussion of Fig. A1) an indication of how the sample sizes for your 3-week events relate to the 3/5-day events. This would help readers to make more sense of the statements made in the introduction (paragraph starting on l. 39) and of the ratios shown in the new Figure A1. Indeed, interpreting any overlap ratio between two sets of events is difficult without knowing whether the two sets contain a similar number of events or not.*

**Answer**: This is a very good point. We added this information in a supplementary table and made a reference to it in the caption of Figure A1, see Table 1 reproduced below.
* * *
**Comment 1.4** *I still struggle with the authors' reply to my previous comment 1.b. It is true that persistent warm or cold spells lead to significant impacts, yet less clear that such impacts derive*

|  | **Cold** | **Warm** |
|---|---|---|
| **DJF** | 11-22 (16) | 9-23 (17) |
| | 1.4-4.4 (2.9) | 1.2-3.4 (2.3) |
| **JJA** | 14-26 (19) | 11-24 (17) |
| | 0.6-1.8 (1.0) | 1.0 - 3.3 (1.9) |

Table 1: Number of identified persistent spells (black), and ratio of number of 3-day events to number of persistent spells (blue), for each season and type of spell (warm and cold). For each case, we indicate the range across clusters and in brackets the corresponding median.

*from their persistence and not from the shorter period of peak temperature anomalies within the longer period of unusual warmth/cold. In other words, high impacts during a prolonged warm/cold spell do not mean that the long duration was the cause of the impacts. Indeed, not all of the papers cited by the authors actually prove a link between duration and impacts, and some citations are used in a borderline misleading fashion. For example, White et al. (2022) start their paper with the following sentence: "An unprecedented heatwave occurred in the Pacific Northwest (PNW) from 25 June to 2 July 2021" (8 days – much closer to 5 days than to 3 weeks). I would recommend nuancing the framing in the introduction, clarifying that there is actually quite an incomplete knowledge on the relation between impacts and duration of warm/cold spells, and a limited literature explicitly connecting impacts to persistence.*

**Answer**: Thank you for the suggestions. We reformulated the second paragraph of the introduction by adding a couple more references (including von Buttlar et al. (2018) who make the link between heatwave duration – from days to months – and vegetation impacts) and by adding more nuance in the confidence we have about the relationship between spell duration and impacts:

"*The impacts of warm and cold spells are modulated not only by their magnitude, but also by their temporal persistence and their spatial extent. For instance, long summer or winter cold spells can be especially detrimental to vegetation (Chapman et al., 2020). von Buttlar et al. (2018) found that the duration of heat extremes was key in modulating the impacts on vegetation, with multi-week spells being more harmful than short-term events. Long summer warm spells can also lead to droughts or make droughts worse, notably in water-scarce regions (García-Herrera et al., 2010; Vogel et al., 2021). Finally, Polt et al. (2023) argued in the case of Germany that heatwaves were the most impact-relevant at time scales between 2 weeks and 2 months. While our knowledge of the relationship between the persistence of warm and cold spells and their impacts remains incomplete and further studies are needed to improve it, there is therefore some quantitative evidence that persistent warm spells can lead to much stronger35 impacts on human and natural systems.*"
* * *
**Comment 1.5** *A further detail is that the only definition Anel et al. (2017) ever mention is of "at least two consecutive days". Then it is true that they present a number of case studies, but I could not find any explicit discussion of persistence, duration or whether persistence played a role for the impacts. I am therefore still puzzled as to where the authors conclude that the study explicitly evidences a non-linear impact scaling with duration.*

**Answer**: You are correct; we removed this reference from the introduction.
* * *
**Comment 1.6** *This may be a Latex compilation issue, but in the PDF I used on l. 10 I read "events ? co-localized"*

**Answer**: Probably a compilation issue as you suggested. We'll make sure this gets fixed in the final version.
* * *
**Comment 1.7** *ll. 29-30 This statement may need to be updated, as complete climate data for the whole of 2022 is now available.*

**Answer**: Thanks. We changed l.27 to past tense, but the figures on ll. 29-30 are still valid.
* * *
**Comment 1.8** *l. 62 persistence summer heatwaves. –¿ persistent summer heatwaves*

**Answer**: Thanks, corrected.